# Convergence Analysis of Sequential Federated Learning on Heterogeneous Data

**Yipeng Li and Xinchen Lyu** [*]
National Engineering Research Center for Mobile Network Technologies
Beijing University of Posts and Telecommunications
Beijing, 100876, China
{liyipeng, lvxinchen}@bupt.edu.cn

## Abstract

There are two categories of methods in Federated Learning (FL) for joint training across multiple clients: i) parallel FL (PFL), where clients train models in a parallel manner; and ii) sequential FL (SFL), where clients train models in a sequential manner. In contrast to that of PFL, the convergence theory of SFL on heterogeneous data is still lacking. In this paper, we establish the convergence guarantees of SFL for strongly/general/non-convex objectives on heterogeneous data. The convergence guarantees of SFL are better than that of PFL on heterogeneous data with both full and partial client participation. Experimental results validate the counterintuitive analysis result that SFL outperforms PFL on extremely heterogeneous data in cross-device settings.

## 1 Introduction

Federated Learning (FL) (McMahan et al., 2017) is a popular distributed machine learning paradigm, where multiple clients collaborate to train a global model. To preserve data privacy and security, data must be kept in clients locally cannot be shared with others, causing one severe and persistent issue, namely "data heterogeneity". In cross-device FL, where data is generated and kept in massively distributed resource-constrained devices (e.g., IoT devices), the negative impact of data heterogeneity would be further exacerbated (Jhunjhunwala et al., 2023).

There are two categories of methods in FL to enable distributed training across multiple clients (Qu et al., 2022): i) parallel FL (PFL), where models are trained in a parallel manner across clients with synchronization at intervals, e.g., Federated Averaging (`FedAvg`) (McMahan et al., 2017); and ii) sequential FL (SFL), where models are trained in a sequential manner across clients, e.g., Cyclic Weight Transfer (`CWT`) (Chang et al., 2018). However, both categories of methods suffer from the "client drift" (Karimireddy et al., 2020), i.e., the local updates on heterogeneous clients would drift away from the right direction, resulting in performance degradation.

**Motivation.**   Recently, SFL (more generally, the sequential training manner, see Algorithm 1) has attracted much attention in the FL community (Lee et al., 2020). Specifically, SFL demonstrates advantages on training speed (in terms of training rounds) (Zaccone et al., 2022) and small datasets (Kamp et al., 2023), and both are crucial for cross-device FL. Furthermore, the sequential manner has played a great role in Split learning (SL) (Gupta and Raskar, 2018; Thapa et al., 2022), an emerging distributed learning technology at the edge side (Zhou et al., 2019), where the full model is split into client-side and server-side portions to alleviate the excessive computation overhead for resource-constrained devices. Appendix A will show that the convergence theory in this work is also applicable to SL.

---

[*]Xinchen Lyu is the corresponding author.

37th Conference on Neural Information Processing Systems (NeurIPS 2023).

Convergence theory is critical for analyzing the learning performance of algorithms on heterogeneous data in FL. So far, there are numerous works to analyze the convergence of PFL (Li et al., 2019; Khaled et al., 2020; Koloskova et al., 2020) on heterogeneous data. However, the convergence theory of SFL on heterogeneous data, given the complexity of its sequential training manner, has not been well investigated in the literature, with only limited preliminary empirical studies Gao et al. (2020, 2021). This paper aims to establish the convergence guarantees for SFL and compare the convergence results of PFL and SFL.

**Setup.** In the following, we provide some preliminaries about SFL and PFL.

*Problem formulation.* The basic FL problem is to minimize a global objective function:

$$\min_{\mathbf{x} \in \mathbb{R}^d} \left\{ F(\mathbf{x}) := \frac{1}{M} \sum_{m=1}^{M} \left( F_m(\mathbf{x}) := \mathbb{E}_{\xi \sim \mathcal{D}_m}[f_m(\mathbf{x}; \xi)] \right) \right\},$$

where $F_m$, $f_m$ and $\mathcal{D}_m$ denote the local objective function, the loss function and the local dataset of client $m$ ($m \in [M]$), respectively. In particular, when $\mathcal{D}_m$ has finite data samples $\{\xi_m^i : i \in [|\mathcal{D}_m|]\}$, the local objective function can also be written as $F_m(\mathbf{x}) = \frac{1}{|\mathcal{D}_m|} \sum_{i=1}^{|\mathcal{D}_m|} f_m(\mathbf{x}; \xi_m^i)$.

*Update rule of SFL.* At the beginning of each training round, the indices $\pi_1, \pi_2, \dots, \pi_M$ are sampled without replacement from $\{1, 2, \dots, M\}$ randomly as the clients' training order. Within a round, each client i) initializes its model with the latest parameters from its previous client; ii) performs $K$ steps of local updates over its local dataset; and iii) passes the updated parameters to the next client. This process continues until all clients finish their local training. Let $\mathbf{x}_{m,k}^{(r)}$ denote the local parameters of the $m$-th client (i.e., client $\pi_m$) after $k$ local steps in the $r$-th round, and $\mathbf{x}^{(r)}$ denote the global parameter in the $r$-th round. With SGD (Stochastic Gradient Descent) as the local solver, the update rule of SFL is as follows:

$$\text{Local update}: \mathbf{x}_{m,k+1}^{(r)} = \mathbf{x}_{m,k}^{(r)} - \eta \mathbf{g}_{\pi_m,k}^{(r)}, \quad \text{initializing as } \mathbf{x}_{m,0}^{(r)} = \begin{cases} \mathbf{x}^{(r)}, & m = 1 \\ \mathbf{x}_{m-1,K}^{(r)}, & m > 1 \end{cases}$$

$$\text{Global model}: \mathbf{x}^{(r+1)} = \mathbf{x}_{M,K}^{(r)}$$

where $\mathbf{g}_{\pi_m,k}^{(r)} := \nabla f_{\pi_m}(\mathbf{x}_{m,k}^{(r)}; \xi)$ denotes the stochastic gradient of $F_{\pi_m}$ regarding parameters $\mathbf{x}_{m,k}^{(r)}$ and $\eta$ denotes the learning rate. See Algorithm 1. Notations are summarized in Appendix C.1.

*Update rule of PFL.* Within a round, each client i) initializes its model with the global parameters; ii) performs $K$ steps of local updates; and iii) sends the updated parameters to the central server. The server will aggregate the local parameters to generate the global parameters. See Algorithm 2

In this work, unless otherwise stated, we use SFL and PFL to represent the classes of algorithms that share the same update rule as Algorithm 1 and Algorithm 2, respectively.

---

**Algorithm 1:** Sequential FL

**Output:** $\bar{\mathbf{x}}^{(R)}$: weighted average on $\mathbf{x}^{(r)}$
1 **for** *training round* $r = 0, 1, \dots, R-1$ **do**
2     Sample a permutation $\pi_1, \pi_2, \dots, \pi_M$ of $\{1, 2, \dots, M\}$
3     **for** $m = 1, 2, \dots, M$ **in sequence do**
4         $\mathbf{x}_{m,0}^{(r)} = \begin{cases} \mathbf{x}^{(r)}, & m = 1 \\ \mathbf{x}_{m-1,K}^{(r)}, & m > 1 \end{cases}$
5         **for** *local step* $k = 0, \dots, K-1$ **do**
6             $\mathbf{x}_{m,k+1}^{(r)} = \mathbf{x}_{m,k}^{(r)} - \eta \mathbf{g}_{\pi_m,k}^{(r)}$
7     Global model: $\mathbf{x}^{(r+1)} = \mathbf{x}_{M,K}^{(r)}$

**Algorithm 2:** Parallel FL

**Output:** $\bar{\mathbf{x}}^{(R)}$: weighted average on $\mathbf{x}^{(r)}$
1 **for** *training round* $r = 0, 1, \dots, R-1$ **do**
2     **for** $m = 1, 2, \dots, M$ **in parallel do**
3         $\mathbf{x}_{m,0}^{(r)} = \mathbf{x}^{(r)}$
4         **for** *local step* $k = 0, \dots, K-1$ **do**
5             $\mathbf{x}_{m,k+1}^{(r)} = \mathbf{x}_{m,k}^{(r)} - \eta \mathbf{g}_{m,k}^{(r)}$
6     Global model: $\mathbf{x}^{(r+1)} = \frac{1}{M} \sum_{m=1}^{M} \mathbf{x}_{m,K}^{(r)}$

## 2 Contributions

**Brief literature review.** The most relevant work is the convergence of PFL and Random Reshuffling (SGD-RR). There are a wealth of works that have analyzed the convergence of PFL on data heterogeneity (Li et al., 2019; Khaled et al., 2020; Karimireddy et al., 2020; Koloskova et al., 2020; Woodworth et al., 2020b), system heterogeneity (Wang et al., 2020), partial client participation (Li et al., 2019; Yang et al., 2021; Wang and Ji, 2022) and other variants (Karimireddy et al., 2020; Reddi et al., 2021). In this work, we compare the convergence bounds between PFL and SFL (see Subsection 3.3) on heterogeneous data.

SGD-RR (where data samples are sampled without replacement) is deemed to be more practical than SGD (where data samples are sample with replacement), and thus attracts more attention recently. Gürbüzbalaban et al. (2021); Haochen and Sra (2019); Nagaraj et al. (2019); Ahn et al. (2020); Mishchenko et al. (2020) have proved the upper bounds and Safran and Shamir (2020, 2021); Rajput et al. (2020); Cha et al. (2023) have proved the lower bounds of SGD-RR. In particular, the lower bounds in Cha et al. (2023) are shown to match the upper bounds in Mishchenko et al. (2020). In this work, we use the bounds of SGD-RR to exam the tightness of that of SFL (see Subsection 3.2).

Recently, the shuffling-based method has been applied to FL (Mishchenko et al., 2022; Yun et al., 2022; Cho et al., 2023; Malinovsky et al., 2023). The most relevant works are FL with cyclic client participation (Cho et al., 2023) and FL with shuffling client participation (Malinovsky et al., 2023). The detailed comparisons are given in Appendix B.

**Challenges.** The theory of SGD is applicable to SFL on homogeneous data, where SFL can be reduced to SGD. However, the theory of SGD can be no longer applicable to SFL on heterogeneous data. This is because for any pair of indices $m$ and $k$ (except $m = 1$ and $k = 0$) within a round, the stochastic gradient is not an (conditionally) unbiased estimator of the global objective:

$$\mathbb{E}\left[\nabla f_{\pi_m}(\mathbf{x}_{m,k}; \xi) \mid \mathbf{x}\right] \neq \nabla F(\mathbf{x}_{m,k}).$$

In general, the challenges of establishing convergence guarantees of SFL mainly arise from (i) the sequential training manner across clients and (ii) multiple local steps of SGD at each client.

*Sequential training manner across clients (vs. PFL).* In PFL, local model parameters are updated in parallel within each round and synchronized at the end of the round. In this case, the local updates across clients are mutually independent when conditional on all the randomness prior to the round. However, in SFL, client's local updates additionally depend on the randomness of all previous clients. This makes bounding the client drift of SFL more complex than that of PFL.

*Multiple local steps of SGD at each client (vs. SGD-RR).* SGD-RR samples data samples without replacement and then performs one step of gradient descent (GD) on each data sample. Similarly, SFL samples clients without replacement and then performs multiple steps of SGD on each local objective (i.e., at each client). In fact, SGD-RR can be regarded as a special case of SFL. Thus, the derivation of convergence guarantees of SFL is also more complex than that of SGD-RR.

**Contributions.** The main contributions are as follows:

- We derive convergence guarantees of SFL for strongly convex, general convex and non-convex objectives on heterogeneous data with the standard assumptions in FL in Subsection 3.2.
- We compare the convergence guarantees of PFL and SFL, and find a *counterintuitive* comparison result that the guarantee of SFL is better than that of PFL (with both full participation and partial participation) in terms of training rounds on heterogeneous data in Subsection 3.3.
- We validate our comparison result with simulations on quadratic functions (Subsection 4.1) and experiments on real datasets (Subsection 4.2). The experimental results exhibit that SFL outperforms PFL on extremely heterogeneous data in cross-device settings.

## 3 Convergence theory

We consider three typical cases for convergence theory, i.e., the strongly convex case, the general convex case and the non-convex case, where all local objectives $F_1, F_2, \ldots, F_M$ are $\mu$-strongly convex, general convex ($\mu = 0$) and non-convex.

## 3.1 Assumptions

We assume that (i) $F$ is lower bounded by $F^*$ for all cases and there exists a minimizer $\mathbf{x}^*$ such that $F(\mathbf{x}^*) = F^*$ for strongly and general convex cases; (ii) each local objective function is $L$-smooth (Assumption 1). Furthermore, we need to make assumptions on the diversities: (iii) the assumptions on the stochasticity bounding the diversity of $\{f_m(\cdot; \xi_m^i) : i \in [|\mathcal{D}_m|]\}$ with respect to $i$ inside each client (Assumption 2); (iv) the assumptions on the heterogeneity bounding the diversity of local objectives $\{F_m : m \in [M]\}$ with respect to $m$ across clients (Assumptions 3a, 3b).

**Assumption 1** ($L$-Smoothness). *Each local objective function $F_m$ is $L$-smooth, $m \in \{1, 2, \ldots, M\}$, i.e., there exists a constant $L > 0$ such that $\|\nabla F_m(\mathbf{x}) - \nabla F_m(\mathbf{y})\| \leq L \|\mathbf{x} - \mathbf{y}\|$ for all $\mathbf{x}, \mathbf{y} \in \mathbb{R}^d$.*

*Assumptions on the stochasticity.* Since both Algorithms 1 and 2 use SGD (data samples are chosen with replacement) as the local solver, the stochastic gradient at each client is an (conditionally) unbiased estimate of the gradient of the local objective function: $\mathbb{E}_{\xi \sim \mathcal{D}_m}[f_m(\mathbf{x}; \xi) | \mathbf{x}] = \nabla F_m(\mathbf{x})$. Then we use Assumption 2 to bound the stochasticity, where $\sigma$ measures the level of stochasticity.

**Assumption 2.** *The variance of the stochastic gradient at each client is bounded:*

$$\mathbb{E}_{\xi \sim \mathcal{D}_m} \left[ \|\nabla f_m(\mathbf{x}; \xi) - \nabla F_m(\mathbf{x})\|^2 \Big| \mathbf{x} \right] \leq \sigma^2, \quad \forall m \in \{1, 2, \ldots, M\} \tag{1}$$

*Assumptions on the heterogeneity.* Now we make assumptions on the diversity of the local objective functions in Assumption 3a and Assumption 3b, also known as the heterogeneity in FL. Assumption 3a is made for non-convex cases, where the constants $\beta$ and $\zeta$ measure the heterogeneity of the local objective functions, and they equal zero when all the local objective functions are identical to each other. Further, if the local objective functions are strongly and general convex, we use one weaker assumption 3b as Koloskova et al. (2020), which bounds the diversity only at the optima.

**Assumption 3a.** *There exist constants $\beta^2$ and $\zeta^2$ such that*

$$\frac{1}{M} \sum_{m=1}^{M} \|\nabla F_m(\mathbf{x}) - \nabla F(\mathbf{x})\|^2 \leq \beta^2 \|\nabla F(\mathbf{x})\|^2 + \zeta^2 \tag{2}$$

**Assumption 3b.** *There exists one constant $\zeta_*^2$ such that*

$$\frac{1}{M} \sum_{m=1}^{M} \|\nabla F_m(\mathbf{x}^*)\|^2 = \zeta_*^2 \tag{3}$$

*where $\mathbf{x}^* \in \arg\min_{\mathbf{x} \in \mathbb{R}^d} F(\mathbf{x})$ is one global minimizer.*

## 3.2 Convergence analysis of SFL

**Theorem 1.** *For SFL (Algorithm 1), there exist a constant effective learning rate $\tilde{\eta} := MK\eta$ and weights $w_r$, such that the weighted average of the global parameters $\bar{\mathbf{x}}^{(R)} := \frac{1}{W_R} \sum_{r=0}^{R} w_r \mathbf{x}^{(r)}$ ($W_R = \sum_{r=0}^{R} w_r$) satisfies the following upper bounds:*

***Strongly convex:*** *Under Assumptions 1, 2, 3b, there exist a constant effective learning rate $\frac{1}{\mu R} \leq \tilde{\eta} \leq \frac{1}{6L}$ and weights $w_r = (1 - \frac{\mu \tilde{\eta}}{2})^{-(r+1)}$, such that it holds that*

$$\mathbb{E}\left[F(\bar{\mathbf{x}}^{(R)}) - F(\mathbf{x}^*)\right] \leq \frac{9}{2} \mu D^2 \exp\left(-\frac{\mu \tilde{\eta} R}{2}\right) + \frac{12 \tilde{\eta} \sigma^2}{MK} + \frac{18 L \tilde{\eta}^2 \sigma^2}{MK} + \frac{18 L \tilde{\eta}^2 \zeta_*^2}{M} \tag{4}$$

***General convex:*** *Under Assumptions 1, 2, 3b, there exist a constant effective learning rate $\tilde{\eta} \leq \frac{1}{6L}$ and weights $w_r = 1$, such that it holds that*

$$\mathbb{E}\left[F(\bar{\mathbf{x}}^{(R)}) - F(\mathbf{x}^*)\right] \leq \frac{3D^2}{\tilde{\eta} R} + \frac{12 \tilde{\eta} \sigma^2}{MK} + \frac{18 L \tilde{\eta}^2 \sigma^2}{MK} + \frac{18 L \tilde{\eta}^2 \zeta_*^2}{M} \tag{5}$$

***Non-convex:*** *Under Assumptions 1, 2, 3a, there exist a constant effective learning rate $\tilde{\eta} \leq \frac{1}{6L(\beta+1)}$ and weights $w_r = 1$, such that it holds that*

$$\min_{0 \leq r \leq R} \mathbb{E}\left[\|\nabla F(\mathbf{x}^{(r)})\|^2\right] \leq \frac{3A}{\tilde{\eta} R} + \frac{3L \tilde{\eta} \sigma^2}{MK} + \frac{27 L^2 \tilde{\eta}^2 \sigma^2}{8MK} + \frac{27 L^2 \tilde{\eta}^2 \zeta^2}{8M} \tag{6}$$

*where $D := \|x^{(0)} - x^*\|$ for the convex cases and $A := F(\mathbf{x}^{(0)}) - F^*$ for the non-convex case.*

*The effective learning rate $\tilde{\eta} := MK\eta$ is used in the upper bounds as* Karimireddy et al. (2020); Wang et al. (2020) *did. All these upper bounds consist of two parts: the optimization part (the first term) and the error part (the last three terms). Setting $\tilde{\eta}$ larger can make the optimization part vanishes at a higher rate, yet cause the error part to be larger. This implies that we need to choose an appropriate $\tilde{\eta}$ to achieve a balance between these two parts, which is actually done in Corollary 1. Here we choose the best learning rate with a prior knowledge of the total training rounds $R$, as done in the previous works (Karimireddy et al., 2020; Reddi et al., 2021).*

**Corollary 1.** *Applying the results of Theorem 1. By choosing a appropriate learning rate (see the proof of Theorem 1 in Appendix D), we can obtain the convergence bounds for SFL as follows:*

**Strongly convex**: *Under Assumptions 1, 2, 3b, there exist a constant effective learning rate $\frac{1}{\mu R} \leq \tilde{\eta} \leq \frac{1}{6L}$ and weights $w_r = (1 - \frac{\mu\tilde{\eta}}{2})^{-(r+1)}$, such that it holds that*

$$\mathbb{E}\left[F(\bar{\mathbf{x}}^{(R)}) - F(\mathbf{x}^*)\right] = \tilde{\mathcal{O}}\left(\frac{\sigma^2}{\mu MKR} + \frac{L\sigma^2}{\mu^2 MKR^2} + \frac{L\zeta_*^2}{\mu^2 MR^2} + \mu D^2 \exp\left(-\frac{\mu R}{12L}\right)\right) \quad (7)$$

**General convex**: *Under Assumptions 1, 2, 3b, there exist a constant effective learning rate $\tilde{\eta} \leq \frac{1}{6L}$ and weights $w_r = 1$, such that it holds that*

$$\mathbb{E}\left[F(\bar{\mathbf{x}}^{(R)}) - F(\mathbf{x}^*)\right] = \mathcal{O}\left(\frac{\sigma D}{\sqrt{MKR}} + \frac{(L\sigma^2 D^4)^{1/3}}{(MK)^{1/3}R^{2/3}} + \frac{(L\zeta_*^2 D^4)^{1/3}}{M^{1/3}R^{2/3}} + \frac{LD^2}{R}\right) \quad (8)$$

**Non-convex**: *Under Assumptions 1, 2, 3a, there exist a constant effective learning rate $\tilde{\eta} \leq \frac{1}{6L(\beta+1)}$ and weights $w_r = 1$, such that it holds that*

$$\min_{0 \leq r \leq R} \mathbb{E}\left[\|\nabla F(\mathbf{x}^{(r)})\|^2\right] = \mathcal{O}\left(\frac{(L\sigma^2 A)^{1/2}}{\sqrt{MKR}} + \frac{(L^2\sigma^2 A^2)^{1/3}}{(MK)^{1/3}R^{2/3}} + \frac{(L^2\zeta_*^2 A^2)^{1/3}}{M^{1/3}R^{2/3}} + \frac{L\beta A}{R}\right) \quad (9)$$

*where $\mathcal{O}$ omits absolute constants, $\tilde{\mathcal{O}}$ omits absolute constants and polylogarithmic factors, $D := \|x^{(0)} - x^*\|$ for the convex cases and $A := F(\mathbf{x}^{(0)}) - F^*$ for the non-convex case.*

*Convergence rate.* By Corollary 1, for sufficiently large $R$, the convergence rate is determined by the first term for all cases, resulting in convergence rates of $\tilde{\mathcal{O}}(1/MKR)$ for strongly convex cases, $\mathcal{O}(1/\sqrt{MKR})$ for general convex cases and $\mathcal{O}(1/\sqrt{MKR})$ for non-convex cases.

*SGD-RR vs. SFL.* Recall that SGD-RR can be seen as one special case of SFL, where one step of GD is performed on each local objective $F_m$ (i.e, $K = 1$ and $\sigma = 0$). The bound of SFL turns to $\tilde{\mathcal{O}}\left(\frac{L\zeta_*^2}{\mu^2 MR^2} + \mu D^2 \exp\left(-\frac{\mu R}{L}\right)\right)$ when $K = 1$ and $\sigma = 0$ for the strongly convex case. Then let us borrow the upper bound from Mishchenko et al. (2020)'s Corollary 1,

$$\text{(Strongly convex)} \quad \mathbb{E}\left[\|\mathbf{x}^{(R)} - \mathbf{x}^*\|^2\right] = \tilde{\mathcal{O}}\left(\frac{L\zeta_*^2}{\mu^3 MR^2} + D^2 \exp\left(-\frac{\mu MR}{L}\right)\right).$$

As we can see, the bound of SGD-RR only has an advantage on the second term (marked in red), which can be omitted for sufficiently large $R$. The difference on the constant $\mu$ is because their bound is for $\mathbb{E}\left[\|\mathbf{x}^{(R)} - \mathbf{x}^*\|^2\right]$ (see Stich (2019b)). Furthermore, our bound also matches the lower bound $\Omega\left(\frac{L\zeta_*^2}{\mu^2 MR^2}\right)$ of SGD-RR suggested by Cha et al. (2023)'s Theorem 3.1 for sufficiently large $R$. For the general convex and non-convex cases, the bounds of SFL (when $K = 1$ and $\sigma = 0$) also match that of SGD-RR (see Mishchenko et al. (2020)'s Theorems 3, 4). These all suggest our bounds are tight. Yet a specialized lower bound for SFL is still required.

*Effect of local steps.* Two comments are included: i) It can be seen that local updates can help the convergence with proper learning rate choices (small enough) by Corollary 1. Yet this increases the total steps (iterations), leading to a higher computation cost. ii) Excessive local updates do not benefit the dominant term of the convergence rate. Take the strongly convex case as an example. When $\frac{\sigma^2}{\mu MKR} \leq \frac{L\zeta_*^2}{\mu^2 MR^2}$, the latter turns dominant, which is unaffected by $K$. In other words, when the value of $K$ exceed $\tilde{\Omega}\left(\sigma^2/\zeta_*^2 \cdot \mu/L \cdot R\right)$, increasing local updates will no longer benefit the dominant term of the convergence rate. Note that the maximum value of $K$ is affected by $\sigma^2/\zeta_*^2$, $\mu/L$ and $R$. This analysis follows Reddi et al. (2021); Khaled et al. (2020).

## 3.3 PFL vs. SFL on heterogeneous data

Table 1: Upper bounds in the strongly convex case with absolute constants and polylogarithmic factors omitted. All results are for heterogeneous settings.

| Method | Bound ($D = \|x^{(0)} - x^*\|$) |
|---|---|
| SGD (Stich, 2019b) | $\frac{\sigma^2}{\mu MKR} + LD^2 \exp\left(-\frac{\mu R}{L}\right)$ [1] |
| PFL | |
|    (Karimireddy et al., 2020) | $\frac{\sigma^2}{\mu MKR} + \frac{L\sigma^2}{\mu^2 KR^2} + \frac{L\zeta^2}{\mu^2 R^2} + \mu D^2 \exp\left(-\frac{\mu R}{L}\right)$ [2] |
|    (Koloskova et al., 2020) | $\frac{\sigma_*^2}{\mu MKR} + \frac{L\sigma_*^2}{\mu^2 KR^2} + \frac{L\zeta_*^2}{\mu^2 R^2} + LKD^2 \exp\left(-\frac{\mu R}{L}\right)$ [3] |
|    Theorem 2 | $\frac{\sigma^2}{\mu MKR} + \frac{L\sigma^2}{\mu^2 KR^2} + \frac{L\zeta_*^2}{\mu^2 R^2} + \mu D^2 \exp\left(-\frac{\mu R}{L}\right)$ |
| SFL | |
|    Theorem 1 | $\frac{\sigma^2}{\mu MKR} + \frac{L\sigma^2}{\mu^2 MKR^2} + \frac{L\zeta_*^2}{\mu^2 MR^2} + \mu D^2 \exp\left(-\frac{\mu R}{L}\right)$ |

[1] SGD with a large mini-batch size. We get the bound in the table by replacing $\sigma^2$ in the Stich (2019b)'s result with $\frac{\sigma^2}{MK}$. See Woodworth et al. (2020b) for more details about Minibatch SGD.

[2] Karimireddy et al. (2020) use $\frac{1}{M}\sum_{m=1}^{M}\|\nabla F_m(\mathbf{x})\|^2 \leq B^2 \|\nabla F(\mathbf{x})\| + G^2$ to bound the heterogeneity, which is equivalent to Assumption 3a. The global learning rate is not considered in this work.

[3] Koloskova et al. (2020) use $\sigma_*^2 \coloneqq \frac{1}{M}\sum_{m=1}^{M}\mathbb{E}\left[\|\nabla f_m(\mathbf{x}^*;\xi) - \nabla F_m(\mathbf{x}^*)\|^2\right]$ to bound the stochasticity, which is weaker than Assumption 3b.

Unless otherwise stated, our comparison is in terms of training rounds, which is also adopted in Gao et al. (2020, 2021). This comparison (running for the same total training rounds $R$) is fair considering the same total computation cost for both methods.

*Convergence results of PFL.* We summarize the existing convergence results of PFL for the strongly convex case in Table 1. Here we slightly improve the convergence result for strongly convex cases by combining the works of Karimireddy et al. (2020); Koloskova et al. (2020). Besides, we note that to derive a unified theory of Decentralized SGD, the proofs of Koloskova et al. (2020) are different from other works focusing on PFL. So we reproduce the bounds for general convex and non-convex cases based on Karimireddy et al. (2020). All our results of PFL are in Theorem 2 (see Appendix E).

*The convergence guarantee of SFL is better than PFL on heterogeneous data.* Take the strongly convex case as an example. According to Table 1, the upper bound of SFL is better than that of PFL, with an advantage of $1/M$ on the second and third terms (marked in red). This benefits from its sequential and shuffling-based training manner. Besides, we can also note that the upper bounds of both PFL and SFL are worse than that of Minibatch SGD.

*Partial client participation.* In the more challenging cross-device settings, only a small fraction of clients participate in each round. Following the works (Li et al., 2019; Yang et al., 2021), we provide the upper bounds of PFL and SFL with partial client participation as follows:

$$\text{PFL:} \quad \tilde{\mathcal{O}}\left(\frac{\sigma^2}{\mu SKR} + \frac{\zeta_*^2}{\mu R}\frac{M-S}{S(M-1)} + \frac{L\sigma^2}{\mu^2 KR^2} + \frac{L\zeta_*^2}{\mu^2 R^2} + \mu D^2 \exp\left(-\frac{\mu R}{L}\right)\right) \quad (10)$$

$$\text{SFL:} \quad \tilde{\mathcal{O}}\left(\frac{\sigma^2}{\mu SKR} + \frac{\zeta_*^2}{\mu R}\frac{(M-S)}{S(M-1)} + \frac{L\sigma^2}{\mu^2 SKR^2} + \frac{L\zeta_*^2}{\mu^2 SR^2} + \mu D^2 \exp\left(-\frac{\mu R}{L}\right)\right) \quad (11)$$

where $S$ clients are selected randomly without replacement. There are additional terms (marked in blue) for both PFL and SFL, which is due to partial client participation and random sampling (Yang et al., 2021). It can be seen that the advantage of $1/S$ (marked in red) of SFL also exists, similar to the full client participation setup.

# 4 Experiments

We run experiments on quadratic functions (Subsection 4.1) and real datasets (Subsection 4.2) to validate our theory. The main findings are i) in extremely heterogeneous settings, SFL performs better than PFL, ii) while in moderately heterogeneous settings, this may not be the case.

## 4.1 Experiments on quadratic functions

According to Table 1, SFL outperforms PFL on heterogeneous settings (in the worst case). Here we show that the counterintuitive result (in contrast to Gao et al. (2020, 2021)) can appear even for simple one-dimensional quadratic functions (Karimireddy et al., 2020).

*Results of simulated experiments.* As shown in Table 2, we use four groups of experiments with various degrees of heterogeneity. To further catch the heterogeneity, in addition to Assumption 3b, we also use bounded Hessian heterogeneity in Karimireddy et al. (2020):

$$\max_m \left\| \nabla^2 F_m(\mathbf{x}) - \nabla^2 F(\mathbf{x}) \right\| \le \delta .$$

Choosing larger values of $\zeta_*$ and $\delta$ means higher heterogeneity. The experimental results of Table 2 are shown in Figure 1. When $\zeta_* = 0$ and $\delta = 0$, SFL outperforms PFL (Group 1). When $\zeta_* = 1$ and $\delta = 0$, the heterogeneity has no bad effect on the performance of PFL while hurts that of SFL significantly (Group 2). When the heterogeneity continues to increase to $\delta > 0$, SFL outperforms PFL with a faster rate and better result (Groups 3 and 4). This in fact tells us that the comparison between PFL and SFL can be associated with the data heterogeneity, and SFL outperforms PFL when meeting high data heterogeneity, which coincides with our theoretical conclusion.

*Limitation and intuitive explanation.* The bounds (see Table 1) above suggest that SFL outperforms PFL regardless of heterogeneity (the value of $\zeta_*$), while the simulated results show that it only holds in extremely heterogeneous settings. This inconsistency is because existing theoretical works (Karimireddy et al., 2020; Koloskova et al., 2020) with Assumptions 3a, 3b may underestimate the capacity of PFL, where the function of global aggregation is omitted. In particular, Wang et al. (2022) have provided rigorous analyses showing that PFL performs much better than the bounds suggest in moderately heterogeneous settings. Hence, the comparison turns vacuous under this condition. Intuitively, PFL updates the global model less frequently with more accurate gradients (with the global aggregation). In contrast, SFL updates the global model more frequently with less accurate gradients. In homogeneous (gradients of both are accurate) and extremely heterogeneous settings (gradients of both are inaccurate), the benefits of frequent updates become dominant, and thus SFL outperforms PFL. In moderately heterogeneous settings, it's the opposite.

Table 2: Settings of simulated experiments. Each group has two local objectives (i.e., $M = 2$) and shares the same global objective. The heterogeneity increases from Group 1 to Group 4.

|  | Group 1 | Group 2 | Group 3 | Group 4 |
|---|---|---|---|---|
| Settings | $\begin{cases} F_1(x) = \frac{1}{2}x^2 \\ F_2(x) = \frac{1}{2}x^2 \end{cases}$ | $\begin{cases} F_1(x) = \frac{1}{2}x^2 + x \\ F_2(x) = \frac{1}{2}x^2 - x \end{cases}$ | $\begin{cases} F_1(x) = \frac{2}{3}x^2 + x \\ F_2(x) = \frac{1}{3}x^2 - x \end{cases}$ | $\begin{cases} F_1(x) = x^2 + x \\ F_2(x) = -x \end{cases}$ |
| $\zeta_*, \delta$ | $\zeta_* = 0, \delta = 0$ | $\zeta_* = 1, \delta = 0$ | $\zeta_* = 1, \delta = \frac{1}{3}$ | $\zeta_* = 1, \delta = 1$ |

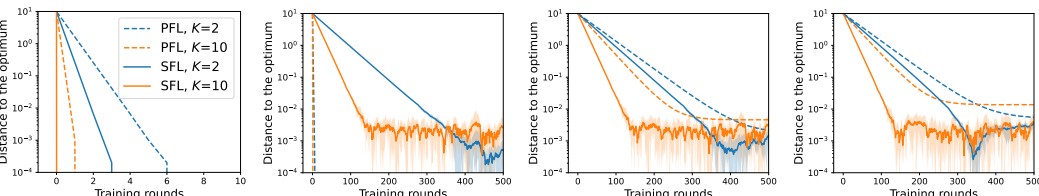

Figure 1: Simulations on quadratic functions. It displays the experimental results from Group 1 to Group 4 in Table 2 from left to right. Shaded areas show the min-max values.

## 4.2 Experiments on real datasets

**Extended Dirichlet strategy.** This is to generate arbitrarily heterogeneous data across clients by extending the popular Dirichlet-based data partition strategy (Yurochkin et al., 2019; Hsu et al., 2019). The difference is to add a step of allocating classes (labels) to determine the number of classes per client (denoted by $C$) before allocating samples via Dirichlet distribution (with concentrate parameter $\alpha$). Thus, the extended strategy can be denoted by $\text{ExDir}(C, \alpha)$. The implementation is as follows (with more details deferred to Appendix G.1):

- Allocating classes: We randomly allocate $C$ different classes to each client. After assigning the classes, we can obtain the prior distribution $q_c$ for each class $c$.

- Allocating samples: For each class $c$, we draw $p_c \sim \text{Dir}(\alpha q_c)$ and then allocate a $p_{c,m}$ proportion of the samples of class $c$ to client $m$. For example, $q_c = [1, 1, 0, 0, \dots,]$ means that the samples of class $c$ are only allocated to the first 2 clients.

**Experiments in cross-device settings.** We next validate the theory in cross-device settings (Kairouz et al., 2021) with partial client participation on real datasets.

*Setup.* We consider the common CV tasks training VGGs (Simonyan and Zisserman, 2014) and ResNets (He et al., 2016) on CIFAR-10 (Krizhevsky et al., 2009) and CINIC-10 (Darlow et al., 2018). Specifically, we use the models VGG-9 (Lin et al., 2020) and ResNet-18 (Acar et al., 2021). We partition the training sets of CIFAR-10 into 500 clients / CINIC-10 into 1000 clients by $\text{ExDir}(1, 10.0)$ and $\text{ExDir}(2, 10.0)$; and spare the test sets for computing test accuracy. As both partitions share the same parameter $\alpha = 10.0$, we use $C = 1$ (where each client owns samples from one class) and $C = 2$ (where each client owns samples from two classes) to represent them, respectively. Note that these two partitions are not rare in FL (Li et al., 2022). They are called extremely heterogeneous data and moderately heterogeneous data respectively in this paper. We fix the number of participating clients to 10 and the mini-batch size to 20. The local solver is SGD with learning rate being constant, momentem being 0 and weight decay being 1e-4. We apply gradient clipping to both algorithms (Appendix G.2) and tune the learning rate by grid search (Appendix G.3).

*The best learning rate of SFL is smaller than that of PFL.* We have the following observations from Figure 2: i) the best learning rates of SFL is smaller than that of PFL (by comparing PFL and SFL), and ii) the best learning rate of SFL becomes smaller as data heterogeneity increases (by comparing the top row and bottom row). These observations are critical for hyperparameter selection.

*Effect of local steps.* Figure 3 is aimed to study the effects of local steps. In both plots, it can be seen that the performance of SFL improves as $K$ increases from 1 to 5. This validates the theoretical conclusion that local steps can help the convergence of SFL even on heterogeneous data. Then, the performance of SFL deteriorates as $K$ increases from 5 to 10, whereas the upper bound of SFL always diminishes as long as $K$ increases. This is because when $K$ exceeds one threshold, the dominant term of the upper bound will be immune to its change as stated in Subsection 3.2. Then, considering "catastrophic forgetting" (Kirkpatrick et al., 2017; Sheller et al., 2019) problems in SFL, it can be expected to see such phenomenon.

*SFL outperforms PFL on extremely heterogeneous data.* The test accuracy results for various tasks are collected in Table 3. When $C = 1$ (extremely heterogeneous), the performance of SFL is better than that of PFL across all

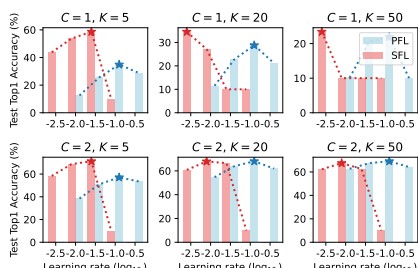

Figure 2: Test accuracies after training VGG-9 on CIFAR-10 for 1000 training rounds with different learning rates.

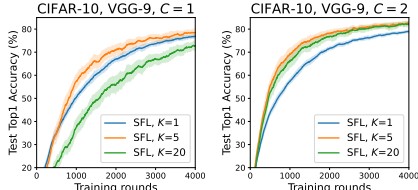

Figure 3: Effect of local steps.

tried settings. When $C = 2$ (moderately heterogeneous), PFL can achieve the close or even slightly better performance than SFL in some cases (e.g., CIFAR-10/$C = 2/K = 50$). This is consistent with our observation and analysis in Subsection 4.1. Notably, on the more complicated dataset CINIC-10, SFL shows better for all settings, which may be due to higher heterogeneity.

Table 3: Test accuracy results in cross-device settings. We run PFL and SFL for 4000 training rounds on CIFAR-10 and CINIC-10. Results are computed across three random seeds and the last 100 training rounds. The better results (with larger than 2% test accuracy gain) between PFL and SFL in each setting are marked in bold.

| Setup | | | $C = 1$ | | | $C = 2$ | | |
|---|---|---|---|---|---|---|---|---|
| Dataset | Model | Method | $K = 5$ | $K = 20$ | $K = 50$ | $K = 5$ | $K = 20$ | $K = 50$ |
| CIFAR-10 | VGG-9 | PFL | $67.61_{\pm4.02}$ | $62.00_{\pm4.90}$ | $45.77_{\pm5.91}$ | $78.42_{\pm1.47}$ | $78.88_{\pm1.35}$ | $78.01_{\pm1.50}$ |
| | | SFL | $\mathbf{78.43}_{\pm2.46}$ | $\mathbf{72.61}_{\pm3.27}$ | $\mathbf{68.86}_{\pm4.19}$ | $\mathbf{82.56}_{\pm1.68}$ | $\mathbf{82.18}_{\pm1.97}$ | $79.67_{\pm2.30}$ |
| | ResNet-18 | PFL | $52.12_{\pm6.09}$ | $44.58_{\pm4.79}$ | $34.29_{\pm4.99}$ | $80.27_{\pm1.52}$ | $82.27_{\pm1.55}$ | $79.88_{\pm2.18}$ |
| | | SFL | $\mathbf{83.44}_{\pm1.83}$ | $\mathbf{76.97}_{\pm4.82}$ | $\mathbf{68.91}_{\pm4.29}$ | $\mathbf{87.16}_{\pm1.34}$ | $\mathbf{84.90}_{\pm3.53}$ | $79.38_{\pm4.49}$ |
| CINIC-10 | VGG-9 | PFL | $52.61_{\pm3.19}$ | $45.98_{\pm4.29}$ | $34.08_{\pm4.77}$ | $55.84_{\pm0.55}$ | $53.41_{\pm0.62}$ | $52.04_{\pm0.79}$ |
| | | SFL | $\mathbf{59.11}_{\pm0.74}$ | $\mathbf{58.71}_{\pm0.98}$ | $\mathbf{56.67}_{\pm1.18}$ | $\mathbf{60.82}_{\pm0.61}$ | $\mathbf{59.78}_{\pm0.79}$ | $\mathbf{56.87}_{\pm1.42}$ |
| | ResNet-18 | PFL | $41.12_{\pm4.28}$ | $33.19_{\pm4.73}$ | $24.71_{\pm4.89}$ | $57.70_{\pm1.04}$ | $55.59_{\pm1.32}$ | $46.99_{\pm1.73}$ |
| | | SFL | $\mathbf{60.36}_{\pm1.37}$ | $\mathbf{51.84}_{\pm2.15}$ | $\mathbf{44.95}_{\pm2.97}$ | $\mathbf{64.17}_{\pm1.06}$ | $\mathbf{58.05}_{\pm2.54}$ | $\mathbf{56.28}_{\pm2.32}$ |

# 5 Conclusion

In this paper, we have derived the convergence guarantees of SFL for strongly convex, general convex and non-convex objectives on heterogeneous data. Furthermore, we have compared SFL against PFL, showing that the guarantee of SFL is better than PFL on heterogeneous data. Experimental results validate that SFL outperforms PFL on extremely heterogeneous data in cross-device settings.

Future directions include i) lower bounds for SFL (this work focuses on the upper bounds of SFL), ii) other potential factors that may affect the performance of PFL and SFL (this work focuses on data heterogeneity) and iii) new algorithms to facilitate our findings (no new algorithm in this work).

## Acknowledgments

This work was supported in part by the National Key Research and Development Program of China under Grant 2021YFB2900302, in part by the National Science Foundation of China under Grant 62001048, and in part by the Fundamental Research Funds for the Central Universities under Grant 2242022k60006.

We thank the reviewers in NeurIPS 2023 for the insightful suggestions. We thank Sai Praneeth Karimireddy for helping us clear some doubts when proving the bounds.

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

# Appendix

# A Applicable to Split Learning

Split Learning is proposed to address the computation bottleneck of resource-constrained devices, where the full model is split into two parts: the *client-side model* (front-part) and the *server-side model* (back-part). There are two typical algorithms in SL, i.e., Sequential Split Learning (SSL)[2] (Gupta and Raskar, 2018) and Split Federated Learning (SplitFed)[3] (Thapa et al., 2022). The overviews of these four paradigms are illustrated in Figure 4

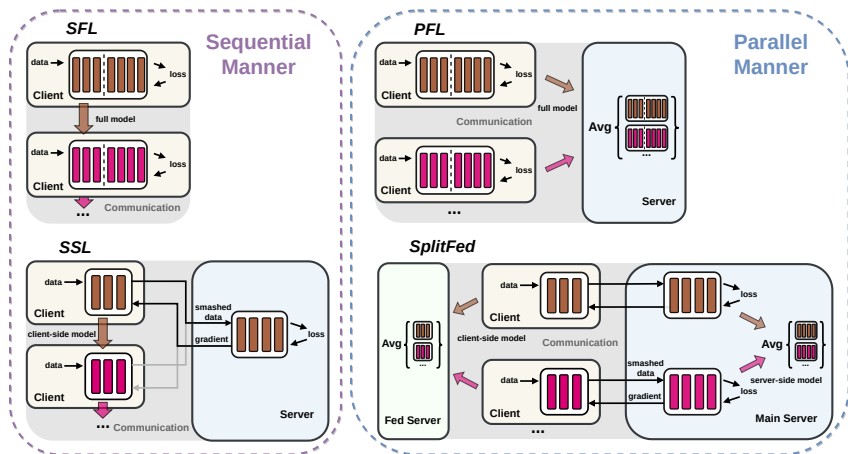

Figure 4: Overviews of paradigms in FL and SL. The top row shows the FL algorithms, SFL and PFL. The bottom row shows the SL algorithms, SSL and SplitFed.

*Training process of SSL.* Each client keeps one client-side model and the server keeps one server-side model. Thus, each client and the server can collaborate to complete the local training of a full model (over the local data kept in clients). Each local step of local training includes the following operations: 1) the client executes the forward pass on the local data, and sends the activations of the cut-layer (called *smashed data*) and labels to the server; 2) the server executes the forward pass with received the smashed data and computes the loss with received labels; 3) the server executes the backward pass and send the gradients of the smashed data to the client; 4) the client executes the backward pass with the received gradients. After finishing the local training, the client sends the updated parameters of its client-side model to the next client. This process continues until all clients complete their local training. See the bottom-left subfigure and Algorithms 3, 4. For clarity, we adjust the superscripts and subscripts in this section, and provide the required notations in Table 4.

*Training process of SplitFed.* Each client keeps one client-side model and the server keeps (named as *main server*) keeps multiple server-side models, whose quantity is the same as the number of clients. Thus, each client and its corresponding server-side model in the main server can complete the local training of a full model in parallel. The local training operations of each client in SplitFed are identical to that in SSL. After the local training with the server, clients send the updated parameters to the fed server, one server introduced in SplitFed to achieve the aggregation of client-side models. The fed server aggregates the received parameters and sends the aggregated (averaged) parameters to the clients. The main server also aggregates the parameters of server-side models it kept and updates them accordingly. See the bottom-right subfigure and Thapa et al. (2021)'s Algorithm 2.

*Applicable to SL.* According to the complete training process of SSL and SplitFed, we can conclude the relationships between SFL and SSL, and, PFL and SplitFed as follows:

- SSL and SplitFed can be viewed as the practical implementations of SFL and PFL respectively in the context of SL.
- SSL and SplitFed share the same update rules with SFL (Algorithm 1) and PFL (Algorithm 2) respectively, and hence, the same convergence results.

---

[2]In SSL, client-side model parameters can be synchronized in two modes, the *peer-to-peer mode* and *centralized mode*. In the peer-to-peer mode, parameters are sent to the next client directly, while in the centralized mode, parameters are relayed to the next client through the server. This paper considers the peer-to-peer mode.

[3]There are two versions of SplitFed and the first version is considered in this paper by default.

Table 4: Additional notations for Section A.

| Symbol | Description |
|---|---|
| $\tau_m, k$ | number, index of local update steps (when training) with client $\pi_m$ |
| $\mathbf{x}_m^{(r,k)}/\mathbf{x}_{c,m}^{(r,k)}/\mathbf{x}_{s,m}^{(r,k)}$ | full/client-side/server-side local model parameters ($\mathbf{x}_m^{(r,k)} = [\mathbf{x}_{c,m}^{(r,k)}; \mathbf{x}_{s,m}^{(r,k)}]$) after $k$ local updates with client $\pi_m$ in the $r$-th round |
| $\boldsymbol{X}_m^{(r,k)}/\boldsymbol{Y}_m^{(r,k)}/\hat{\boldsymbol{Y}}_m^{(r,k)}$ | features/labels/predictors after $k$ local updates with client $\pi_m$ in the $r$-th round |
| $\mathbf{x}^{(r)}/\mathbf{x}_c^{(r)}/\mathbf{x}_s^{(r)}$ | full/client-side/server-side global model parameters in the $r$-th round |
| $\boldsymbol{A}_m^{(r,k)}$ | smashed data (activation of the cut layer) after $k$ local updates with client $\pi_m$ in the $r$-th round |
| $\ell_{\pi_m}$ | loss function with client $\pi_m$ |
| $\nabla\ell_{\pi_m}(\mathbf{x}_{s,m}^{(r,k)}; \boldsymbol{A}_m^{(r,k)})$ | gradients of the loss regarding $\mathbf{x}_{s,m}^{(r,k)}$ on input $\boldsymbol{A}_m^{(r,k)}$ |
| $\nabla\ell_{\pi_m}(\boldsymbol{A}_m^{(r,k)}; \mathbf{x}_{s,m}^{(r,k)})$ | gradients of the loss regarding $\boldsymbol{A}_m^{(r,k)}$ on parameters $\mathbf{x}_{s,m}^{(r,k)}$ |
| $\nabla\ell_{\pi_m}(\mathbf{x}_{c,m}^{(r,k)}; \boldsymbol{X}_m^{(r,k)})$ | gradients of the loss regarding $\mathbf{x}_{c,m}^{(r,k)}$ on input $\boldsymbol{X}_m^{(r,k)}$ |

---

**Algorithm 3:** Sequential Split Learning (Server-side operations)

---

**Main Server** executes:

1 Initialize server-side global parameters $\mathbf{x}_s^{(0)}$
2 **for** *round* $r = 0, \ldots, R-1$ **do**
3      Sample a permutation $\pi_1, \pi_2, \ldots, \pi_M$ of $\{1, 2, \ldots, M\}$ as clients' update order
4      **for** $m = 1, 2, \ldots, M$ **in sequence do**
5          Initialize server-side local parameters: $\mathbf{x}_{s,m}^{(r,0)} \leftarrow \begin{cases} \mathbf{x}_s^{(r)}, & m = 1 \\ \mathbf{x}_{s,m-1}^{(r,\tau_{m-1})}, & m > 1 \end{cases}$
6          **for** *local update step* $k = 0, \ldots, \tau_m - 1$ **do**
7              Receive $(\boldsymbol{A}_m^{(r,k)}, \boldsymbol{Y}_m^{(r,k)})$ from client $m$          // Com.
8              Execute forward passes with smashed data $\boldsymbol{A}_m^{(r,k)}$
9              Calculate the loss with $(\hat{\boldsymbol{Y}}_m^{(r,k)}, \boldsymbol{Y}_m^{(r,k)})$
10             Execute backward passes and compute $\nabla\ell_{\pi_m}(\mathbf{x}_{s,m}^{(r,k)}; \boldsymbol{A}_m^{(r,k)})$
11             Send $\nabla\ell_{\pi_m}(\boldsymbol{A}_m^{(r,k)}; \mathbf{x}_{s,m}^{(r,k)})$ to client $m$          // Com.
12             Update server-side parameters: $\mathbf{x}_{s,m}^{(r,k+1)} \leftarrow \mathbf{x}_{s,m}^{(r,k)} - \eta\nabla\ell_{\pi_m}(\mathbf{x}_{s,m}^{(r,k)}; \boldsymbol{A}_m^{(r,k)})$

---

**Algorithm 4:** Sequential Split Learning (Client-side operations)

---

**Client** $\pi_m$ executes:

1 Request the latest client-side parameters from the previous client          // Com.
2 Initialize client-side parameters: $\mathbf{x}_{c,m}^{(r,0)} \leftarrow \begin{cases} \mathbf{x}_c^{(r)}, & m = 1 \\ \mathbf{x}_{c,m-1}^{(r,\tau_{m-1})}, & m > 1 \end{cases}$
3 **for** *local update step* $k = 0, \ldots, \tau_m - 1$ **do**
4      Execute forward passes with data features $\boldsymbol{X}_m^{(r,k)}$
5      Send $(\boldsymbol{A}_m^{(r,k)}, \boldsymbol{Y}_m^{(r,k)})$ to the server          // Com.
6      Receive $\nabla\ell_{\pi_m}(\boldsymbol{A}_m^{(r,k)}; \mathbf{x}_{s,m}^{(r,k)})$          // Com.
7      Execute backward passes and compute $\nabla\ell_{\pi_m}(\mathbf{x}_{c,m}^{(r,k)}; \boldsymbol{X}_m^{(r,k)})$
8      Update client-side parameters: $\mathbf{x}_{c,m}^{(r,k+1)} \leftarrow \mathbf{x}_{c,m}^{(r,k)} - \eta\nabla\ell_{\pi_m}(\mathbf{x}_{c,m}^{(r,k)}; \boldsymbol{X}_m^{(r,k)})$

---

# B   Related work

**Convergence of PFL.**   The convergence of PFL (also known as Local SGD, `FedAvg`) has developed rapidly recently, with weaker assumptions, tighter bounds and more complex scenarios. Zhou and Cong (2017); Stich (2019a); Khaled et al. (2020); Wang and Joshi (2021) analyzed the convergence of PFL on homogeneous data. Li et al. (2019) derived the convergence guarantees for PFL with the bounded gradients assumption on heterogeneous data. Yet this assumption has been shown too stronger (Khaled et al., 2020). To further catch the heterogeneity, Karimireddy et al. (2020); Koloskova et al. (2020) assumed the variance of the gradients of local objectives is bounded either uniformly (Assumption 3a) or on the optima (Assumption 3b). Moreover, Li et al. (2019); Karimireddy et al. (2020); Yang et al. (2021) also consider the convergence with partial client participation. The lower bounds of PFL are also studied in Woodworth et al. (2020a,b); Yun et al. (2022). There are other variants in PFL, which show a faster convergence rate than the vanilla one (Algorithm 2), e.g., `SCAFFOLD` (Karimireddy et al., 2020).

**Convergence of SGD-RR.**   Random Reshuffling (SGD-RR) has attracted more attention recently, as it (where data samples are sampled without replacement) is more common in practice than its counterpart algorithms SGD (where data samples are sample with replacement). Early works (Gürbüzbalaban et al., 2021; Haochen and Sra, 2019) prove the upper bounds for strongly convex and twice-smooth objectives. Subsequent works (Nagaraj et al., 2019; Ahn et al., 2020; Mishchenko et al., 2020) further prove upper bounds for strongly convex, convex and non-convex cases. The lower bounds of SGD-RR are also investigated in the quadratic case Safran and Shamir (2020, 2021) and the strongly convex case (Rajput et al., 2020; Cha et al., 2023). In particular, the lower bounds in Cha et al. (2023) are shown to match the upper bounds in Mishchenko et al. (2020) for the both strongly convex and general convex cases. These works have reached a consensus that SGD-RR is better than SGD a least when the number of epochs (passes over the data) is large enough. In this paper, we use the bounds of SGD-RR to exam the tightness of our bounds of SFL.

There are also works studying the randomized incremental gradient methods are also relevant (Ram et al., 2009; Johansson et al., 2010; Ayache and El Rouayheb, 2021; Mao et al., 2020; Cyffers and Bellet, 2022), which consider a single update at each client and focus on random walks.

**Shuffling-based methods in FL.**   Recently, shuffling-based methods have appeared in FL (Mishchenko et al., 2022; Yun et al., 2022; Cho et al., 2023). Mishchenko et al. (2022) gave the convergence result of Federated Random Reshuffling (FedRR) as a application to Federated Learning of their theory for Proximal Random Reshuffling (ProxRR). Yun et al. (2022) analyzed the convergence of Minibatch RR and Local RR, the variants of Minibatch SGD and Local SGD (Local SGD is equivalent to PFL in this work), where clients perform SGD-RR locally (in parallel) instead of SGD. Both FedRR and Local RR are different from SFL from the algorithm perspective. See Yun et al. (2022)'s Appendix A for comparison.

The most relevant works are FL with cyclic client participation (Cho et al., 2023) and FL with shuffling client participation (Malinovsky et al., 2023) (we note them when preparing this version).

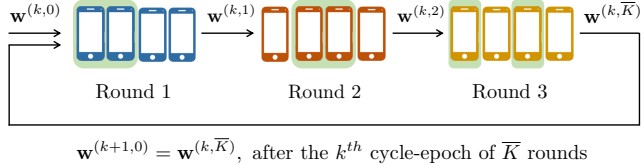

$\mathbf{w}^{(k+1,0)} = \mathbf{w}^{(k,\overline{K})}$, after the $k^{th}$ cycle-epoch of $\overline{K}$ rounds

Figure 5: Illustration of FL with cyclic client participation with $M = 12$ clients divided into $\overline{K} = 3$ groups. In each training round, $N = 2$ clients are selected for training from the client group. All groups are traversed once in a cycle-epoch consisting of $\overline{K}$ training rounds. (Cho et al., 2023).

*Discussion about FL with cyclic client participation.* Cho et al. (2023) consider the scenario where the total $M$ clients are divided into $\overline{K}$ non-overlapping client groups such that each group contains $M/\overline{K}$ clients. In each training round, the sever selects a subset of $N$ clients from a group without replacement for training in this round. One example is shown in Figure 5. As said in the paragraph "Cyclic Client Participation (CyCP)" in their Section 3 (Problem Formulation), the groups' training

order of FL with cyclic client participation is pre-determined and fixed. In contrast, the clients' training order of SFL (precisely, Algorithm 1) will be shuffled at the beginning of each round.

It can be verified by Cho et al. (2023)'s Theorem 2. Letting $\overline{K} = 1$ and $N = M/\overline{K} = M$ and $\overline{K} = M$ and $N = M/\overline{K} = 1$, we get the bounds for PFL and SFL, respectively:

$$\text{PFL:} \quad \tilde{\mathcal{O}}\left(\frac{L\sigma^2}{\mu^2 MKR} + \frac{L^2\zeta^2}{\mu^3 M^2 R^2}\right)$$

where we have omitted the optimization term and changed their notations to ours (change $\alpha$ to 0, $\gamma$ to $\zeta$, $\nu$ to $\zeta$, $T$ to $R$).

$$\text{SFL:} \quad \tilde{\mathcal{O}}\left(\frac{L\sigma^2}{\mu^2 KR} + \frac{L^2\zeta^2}{\mu^3 R^2} + \frac{L^2\zeta^2}{\mu^3 M^2 R^2}\right)$$

where we have omitted the optimization term and changed their notations to ours (change $\alpha$ to $\zeta$, $\gamma$ to 0, $\nu$ to $\zeta$, $T$ to $MR$). As we can see, we do not see a clear advantage of SFL like ours.

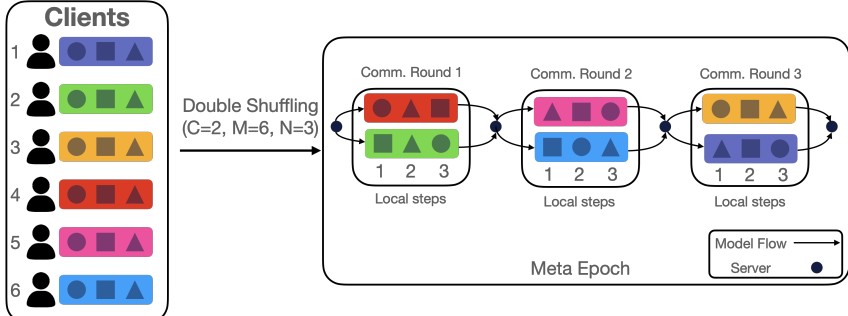

Figure 6: Visualization of FL with shuffling client participation for 6 clients, each with 3 datapoints. Two clients are sampled in each communication round (Malinovsky et al., 2023).

*Discussion about FL with shuffling client participation.* At the beginning of each meta epoch, FL with shuffling client participation partition all $M$ clients into $M/C$ cohorts, each with size $C$. These cohorts are obtained using the without replacement sampling of clients. Each meta epoch contains $R = M/C$ communication rounds. At each communication round, clients in a cohort participate in the training process. One example is shown in Figure 6. It is noteworthy that SGD-RR is used as the local solver in Malinovsky et al. (2023) while SGD is used in this paper.

Letting $C = 1$, $R = M$, $T = R$ in their Theorem 6.1, we can get the bounds for SFL:

$$\text{SFL (Malinovsky et al., 2023):} \quad \mathcal{O}\left(\frac{1}{\mu} \cdot LMK^2\eta^2\zeta_*^2 + D^2\exp\left(-\mu MKR\eta\right)\right) \text{ with } \eta \leq \frac{1}{L}$$

$$\text{SFL (Theorem 1):} \quad \mathcal{O}\left(LMK^2\eta^2\zeta_*^2 + \mu D^2\exp\left(-\mu MKR\eta\right)\right) \text{ with } \eta \leq \frac{1}{LMK}$$

where we only considered the heterogeneity terms (let $\sigma_*^2 = 0$ for Malinovsky et al. (2023) and $\sigma = 0$ for ours) and changed their notations to ours (change $\gamma$ to $\eta$, $N$ to $K$, $\tilde{\sigma}_*$ to $\zeta_*$). Their bound almost matches ours, with some differences on the constants $\mu$, $L$ and restrictions on $\eta$, which is caused by using different local solvers. However, their results are limited to the case where the number of local steps equals the size of the local dataset. It is still uncertain whether their results for SGD-RR can be generalized to situations with varying numbers of local steps.

# C Notations and technical lemmas

## C.1 Notations

Table 5 summarizes the notations appearing in this paper.

Table 5: Summary of key notations.

| Symbol | Description |
| --- | --- |
| $R, r$ | number, index of training rounds |
| $M, m$ | number, index of clients |
| $K, k$ | number, index of local update steps |
| $S$ | number of participating clients |
| $\pi$ | $\{\pi_1, \pi_2, \ldots, \pi_M\}$ is a permutation of $\{1, 2, \ldots, M\}$ |
| $\eta$ | learning rate (or stepsize) |
| $\tilde{\eta}$ | effective learning rate ($\tilde{\eta} := MK\eta$ in SFL and $\tilde{\eta} := K\eta$ in PFL) |
| $\mu$ | $\mu$-strong convexity constant |
| $L$ | $L$-smoothness constant (Asm. 1) |
| $\sigma$ | upper bound on variance of stochastic gradients at each client (Asm. 2) |
| $\beta, \zeta$ | constants in Asm. 3a to bound heterogeneity everywhere |
| $\zeta_*$ | constants in Asm. 3b to bound heterogeneity at the optima |
| $F/F_m$ | global objective/local objective of client $m$ |
| $\mathbf{x}^{(r)}$ | global model parameters in the $r$-th round |
| $\mathbf{x}_{m,k}^{(r)}$ | local model parameters of the $m$-th client after $k$ local steps in the $r$-th round |
| $\mathbf{g}_{\pi_m,k}^{(r)}$ | $\mathbf{g}_{\pi_m,k}^{(r)} := \nabla f_{\pi_m}(\mathbf{x}_{m,k}^{(r)}; \xi)$ denotes the stochastic gradients of $F_{\pi_m}$ regarding $\mathbf{x}_{m,k}^{(r)}$ |
| $\text{ExDir}(C, \alpha)$ | Extended Dirichlet strategy with parameters $C$ and $\alpha$ (see Sec. G.1) |

## C.2 Basic identities and inequalities

These identities and inequalities are mostly from Zhou (2018); Khaled et al. (2020); Mishchenko et al. (2020); Karimireddy et al. (2020); Garrigos and Gower (2023).

For any random variable $\mathbf{x}$, letting the variance can be decomposed as

$$\mathbb{E}\left[\|\mathbf{x} - \mathbb{E}[\mathbf{x}]\|^2\right] = \mathbb{E}\left[\|\mathbf{x}\|^2\right] - \|\mathbb{E}[\mathbf{x}]\|^2 \tag{12}$$

In particular, its version for vectors with finite number of values gives

$$\frac{1}{n}\sum_{i=1}^n \|\boldsymbol{x}_i - \bar{\boldsymbol{x}}\|^2 = \frac{1}{n}\sum_{i=1}^n \|\boldsymbol{x}_i\|^2 - \left\|\frac{1}{n}\sum_{i=1}^n \boldsymbol{x}_i\right\|^2 \tag{13}$$

where vectors $\boldsymbol{x}_1, \ldots, \boldsymbol{x}_n \in \mathbb{R}^d$ are the values of $\mathbf{x}$ and their average is $\bar{\boldsymbol{x}} = \frac{1}{n}\sum_{i=1}^n \boldsymbol{x}_i$.

**Jensen's inequality.** For any convex function $h$ and any vectors $\boldsymbol{x}_1, \ldots, \boldsymbol{x}_n$ we have

$$h\left(\frac{1}{n}\sum_{i=1}^n \boldsymbol{x}_i\right) \leq \frac{1}{n}\sum_{i=1}^n h(\boldsymbol{x}_i). \tag{14}$$

As a special case with $h(\boldsymbol{x}) = \|\boldsymbol{x}\|^2$, we obtain

$$\left\|\frac{1}{n}\sum_{i=1}^n \boldsymbol{x}_i\right\|^2 \leq \frac{1}{n}\sum_{i=1}^n \|\boldsymbol{x}_i\|^2. \tag{15}$$

**Smoothness and general convexity, strong convexity.** There are some useful inequalities with respect to $L$-smoothness (Assumption 1), convexity and $\mu$-strong convexity. Their proofs can be found in Zhou (2018); Garrigos and Gower (2023).

*Bregman Divergence* associated with function $h$ and arbitrary $\boldsymbol{x}, \boldsymbol{y}$ is denoted as

$$D_h(\boldsymbol{x}, \boldsymbol{y}) := h(\boldsymbol{x}) - h(\boldsymbol{y}) - \langle \nabla h(\boldsymbol{y}), \boldsymbol{x} - \boldsymbol{y} \rangle$$

When the function $h$ is convex, the divergence is strictly non-negative. A more formal definition can be found in Orabona (2019). One corollary (Chen and Teboulle, 1993) called *three-point-identity* is,

$$D_h(\boldsymbol{z}, \boldsymbol{x}) + D_h(\boldsymbol{x}, \boldsymbol{y}) - D_h(\boldsymbol{z}, \boldsymbol{y}) = \langle \nabla h(\boldsymbol{y}) - \nabla h(\boldsymbol{x}), \boldsymbol{z} - \boldsymbol{x} \rangle$$

where $\boldsymbol{x}, \boldsymbol{y}, \boldsymbol{z}$ is three points in the domain.

Let $h$ be $L$-smooth. With the definition of Bregman divergence, a consequence of $L$-smoothness is

$$D_h(\boldsymbol{x}, \boldsymbol{y}) = h(\boldsymbol{x}) - h(\boldsymbol{y}) - \langle \nabla h(\boldsymbol{y}), \boldsymbol{x} - \boldsymbol{y} \rangle \leq \frac{L}{2} \|\boldsymbol{x} - \boldsymbol{y}\|^2 \tag{16}$$

Further, If $h$ is $L$-smooth and lower bounded by $h_*$, then

$$\|\nabla h(\boldsymbol{x})\|^2 \leq 2L \left( h(\boldsymbol{x}) - h_* \right). \tag{17}$$

If $h$ is $L$-smooth and convex (The definition of convexity can be found in Boyd et al. (2004)), then

$$D_h(\boldsymbol{x}, \boldsymbol{y}) \geq \frac{1}{2L} \|\nabla h(\boldsymbol{x}) - \nabla h(\boldsymbol{y})\|^2. \tag{18}$$

The function $h : \mathbb{R}^d \to \mathbb{R}$ is $\mu$-strongly convex if and only if there exists a convex function $g : \mathbb{R}^d \to \mathbb{R}$ such that $h(\boldsymbol{x}) = g(\boldsymbol{x}) + \frac{\mu}{2} \|\boldsymbol{x}\|^2$.

If $h$ is $\mu$-strongly convex, it holds that

$$\frac{\mu}{2} \|\boldsymbol{x} - \boldsymbol{y}\|^2 \leq D_h(\boldsymbol{x}, \boldsymbol{y}) \tag{19}$$

### C.3 Technical lemmas

**Lemma 1** (Karimireddy et al. (2020)). *Let $\{\xi_i\}_{i=1}^n$ be a sequence of random variables. And the random sequence $\{\mathbf{x}_i\}_{i=1}^n$ satisfy that $\mathbf{x}_i \in \mathbb{R}^d$ is a function of $\xi_i, \xi_{i-1}, \ldots, \xi_1$ for all $i$. Suppose that the conditional expectation is $\mathbb{E}_{\xi_i} [\mathbf{x}_i | \xi_{i-1}, \ldots \xi_1] = \mathbf{e}_i$ (i.e., the vectors $\{\mathbf{x}_i - \mathbf{e}_i\}$ form a martingale difference sequence with respect to $\{\xi_i\}$), and the variance is bounded by $\mathbb{E}_{\xi_i} \left[ \|\mathbf{x}_i - \mathbf{e}_i\|^2 \Big| \xi_{i-1}, \ldots \xi_1 \right] \leq \sigma^2$. Then it holds that*

$$\mathbb{E} \left[ \left\| \sum_{i=1}^n (\mathbf{x}_i - \mathbf{e}_i) \right\|^2 \right] = \sum_{i=1}^n \mathbb{E} \left[ \|\mathbf{x}_i - \mathbf{e}_i\|^2 \right] \leq n\sigma^2 \tag{20}$$

*Proof.* This conclusion has appeared in Stich and Karimireddy (2019)'s Lemma 15, Karimireddy et al. (2020)'s Lemma 4 (separating mean and variance) and Wang et al. (2020)'s Lemma 2, which is useful for bounding the stochasticity.

$$\mathbb{E} \left[ \left\| \sum_{i=1}^n (\mathbf{x}_i - \mathbf{e}_i) \right\|^2 \right] = \sum_{i=1}^n \mathbb{E} \left[ \|\mathbf{x}_i - \mathbf{e}_i\|^2 \right] + \sum_{i=1}^n \sum_{j \neq i} \mathbb{E} \left[ (\mathbf{x}_i - \mathbf{e}_i)^\top (\mathbf{x}_j - \mathbf{e}_j) \right]$$

Without loss of generality, we can assume that $i < j$. Then the cross terms in the preceding equation can be computed by the law of total expectation:

$$\begin{aligned} \mathbb{E} \left[ (\mathbf{x}_i - \mathbf{e}_i)^\top (\mathbf{x}_j - \mathbf{e}_j) \right] &= \mathbb{E} \left[ \mathbb{E} \left[ (\mathbf{x}_i - \mathbf{e}_i)^\top (\mathbf{x}_j - \mathbf{e}_j) | \xi_i, \ldots, \xi_1 \right] \right] \\ &= \mathbb{E} \big[ (\mathbf{x}_i - \mathbf{e}_i)^\top \underbrace{\mathbb{E}[(\mathbf{x}_j - \mathbf{e}_j) | \xi_i, \ldots, \xi_1]}_{=\mathbf{0}} \big] \\ &= 0 \end{aligned}$$

Here $\mathbb{E}[(\mathbf{x}_j - \mathbf{e}_j)|\xi_i, \ldots, \xi_1] = \mathbf{0}$ can be proved by mathematical induction and the law of total expectation. Then,

$$\mathbb{E}\left[\left\|\sum_{i=1}^n (\mathbf{x}_i - \mathbf{e}_i)\right\|^2\right] = \sum_{i=1}^n \mathbb{E}\left[\|\mathbf{x}_i - \mathbf{e}_i\|^2\right] = \sum_{i=1}^n \mathbb{E}\left[\mathbb{E}\left[\|\mathbf{x}_i - \mathbf{e}_i\|^2\Big|\xi_{i-1}, \ldots, \xi_1\right]\right] \le n\sigma^2,$$

which is the claim of this lemma. Note that since $\mathbb{E}_{\xi_i}[\mathbf{x}_i|\xi_{i-1}, \ldots \xi_1] = \mathbf{e}_i$, the conditional expectation $\mathbf{e}_i$ is not deterministic but a function of $\xi_{i-1}, \ldots, \xi_1$. $\qquad\square$

**Lemma 2** (Karimireddy et al. (2020)). *The following holds for any $L$-smooth and $\mu$-strongly convex function $h$, and any $\boldsymbol{x}, \boldsymbol{y}, \boldsymbol{z}$ in the domain of $h$:*

$$\langle \nabla h(\boldsymbol{x}), \boldsymbol{z} - \boldsymbol{y} \rangle \ge h(\boldsymbol{z}) - h(\mathbf{y}) + \frac{\mu}{4}\|\boldsymbol{y} - \boldsymbol{z}\|^2 - L\|\boldsymbol{z} - \boldsymbol{x}\|^2. \tag{21}$$

*Proof.* Using the *three-point-identity*, we get

$$\langle \nabla h(\boldsymbol{x}), \boldsymbol{z} - \boldsymbol{y} \rangle = D_h(\boldsymbol{y}, \boldsymbol{x}) - D_h(\boldsymbol{z}, \boldsymbol{x}) + h(\boldsymbol{z}) - h(\boldsymbol{y})$$

Then, we get the following two inequalities using smoothness and strong convexity of $h$:

$$\langle \nabla h(\boldsymbol{x}), \boldsymbol{z} - \boldsymbol{y} \rangle \ge \frac{\mu}{2}\|\boldsymbol{y} - \boldsymbol{x}\|^2 - \frac{L}{2}\|\boldsymbol{z} - \boldsymbol{x}\|^2 + h(\boldsymbol{z}) - h(\boldsymbol{y})$$

Further, using Jensen's inequality (i.e., $\|\boldsymbol{y} - \boldsymbol{z}\|^2 \le 2(\|\boldsymbol{x} - \boldsymbol{z}\|^2 + \|\boldsymbol{y} - \boldsymbol{x}\|^2)$), we have

$$\frac{\mu}{2}\|\boldsymbol{y} - \boldsymbol{x}\|^2 \ge \frac{\mu}{4}\|\boldsymbol{y} - \boldsymbol{z}\|^2 - \frac{\mu}{2}\|\boldsymbol{x} - \boldsymbol{z}\|^2.$$

Combining all the inequalities together we have

$$\langle \nabla h(\boldsymbol{x}), \boldsymbol{z} - \boldsymbol{y} \rangle \ge h(\boldsymbol{z}) - h(\boldsymbol{y}) + \frac{\mu}{4}\|\boldsymbol{y} - \boldsymbol{z}\|^2 - \frac{L + \mu}{2}\|\boldsymbol{z} - \boldsymbol{x}\|^2$$

$$\ge h(\boldsymbol{z}) - h(\boldsymbol{y}) + \frac{\mu}{4}\|\boldsymbol{y} - \boldsymbol{z}\|^2 - L\|\boldsymbol{z} - \boldsymbol{x}\|^2 \qquad \triangleright \mu \le L$$

which is the claim of this lemma. $\qquad\square$

**Lemma 3** (Simple Random Sampling). *Let $\boldsymbol{x}_1, \boldsymbol{x}_2, \ldots, \boldsymbol{x}_n$ be fixed units (e.g., vectors). The population mean and population variance are give as*

$$\overline{\boldsymbol{x}} := \tfrac{1}{n}\sum_{i=1}^n \boldsymbol{x}_i \qquad\qquad \zeta^2 := \tfrac{1}{n}\sum_{i=1}^n \|\boldsymbol{x}_i - \overline{\boldsymbol{x}}\|^2$$

*Draw $s \in [n] = \{1, 2, \ldots, n\}$ random units $\mathbf{x}_{\pi_1}, \mathbf{x}_{\pi_2}, \ldots \mathbf{x}_{\pi_s}$ randomly from the population. There are two possible ways of simple random sampling, well known as "sampling with replacement (SWR)" and "sampling without replacement (SWOR)". For these two ways, the expectation and variance of the sample mean $\overline{\mathbf{x}}_\pi := \frac{1}{s}\sum_{p=1}^s \mathbf{x}_{\pi_p}$ satisfies*

$$SWR \ : \ \mathbb{E}[\overline{\mathbf{x}}_\pi] = \overline{\boldsymbol{x}} \qquad\qquad \mathbb{E}\left[\|\overline{\mathbf{x}}_\pi - \overline{\boldsymbol{x}}\|^2\right] = \frac{\zeta^2}{s} \tag{22}$$

$$SWOR: \ \mathbb{E}[\overline{\mathbf{x}}_\pi] = \overline{\boldsymbol{x}} \qquad\qquad \mathbb{E}\left[\|\overline{\mathbf{x}}_\pi - \overline{\boldsymbol{x}}\|^2\right] = \frac{n - s}{s(n - 1)}\zeta^2 \tag{23}$$

*Proof.* The proof of this lemma is mainly based on Mishchenko et al. (2020)'s Lemma 1 (A lemma for sampling without replacement) and Wang et al. (2020)'s Appendix G (Extension: Incorporating Client Sampling). Since the probability of each unit being selected equals $\frac{1}{n}$ in each draw, we can get the expectation and variance of any random unit $\mathbf{x}_{\pi_p}$ at the $p$-th draw:

$$\mathbb{E}\left[\mathbf{x}_{\pi_p}\right] = \sum_{i=1}^n \boldsymbol{x}_i \cdot \Pr(\mathbf{x}_{\pi_p} = \boldsymbol{x}_i) = \sum_{i=1}^n \boldsymbol{x}_i \cdot \frac{1}{n} = \overline{\boldsymbol{x}},$$

$$\mathrm{Var}(\mathbf{x}_{\pi_p}) = \mathbb{E}\left[\|\mathbf{x}_{\pi_p} - \overline{\boldsymbol{x}}\|^2\right] = \sum_{i=1}^n \|\boldsymbol{x}_i - \overline{\boldsymbol{x}}\|^2 \cdot \Pr(\mathbf{x}_{\pi_p} = \boldsymbol{x}_i) = \sum_{i=1}^n \|\boldsymbol{x}_i - \overline{\boldsymbol{x}}\|^2 \cdot \frac{1}{n} = \zeta^2,$$

where the preceding equations hold for both sampling ways. Thus, we can compute the expectations of the sample mean for both sampling ways as

$$\mathbb{E}\left[\overline{\mathbf{x}}_\pi\right] = \mathbb{E}\left[\frac{1}{s}\sum_{p=1}^{s}\mathbf{x}_{\pi_p}\right] = \frac{1}{s}\sum_{p=1}^{s}\mathbb{E}\left[\mathbf{x}_{\pi_p}\right] = \overline{\boldsymbol{x}},$$

which indicates that the sample means for both ways are unbiased estimators of the population mean. The variance of the sample mean can be decomposed as

$$\mathbb{E}\left\|\overline{\mathbf{x}}_\pi - \overline{\boldsymbol{x}}\right\|^2 = \mathbb{E}\left\|\frac{1}{s}\sum_{p=1}^{s}(\mathbf{x}_{\pi_p} - \overline{\boldsymbol{x}})\right\|^2 = \frac{1}{s^2}\sum_{p=1}^{s}\text{Var}(\mathbf{x}_{\pi_p}) + \frac{1}{s^2}\sum_{p=1}^{s}\sum_{q\neq p}^{s}\text{Cov}(\mathbf{x}_{\pi_p},\mathbf{x}_{\pi_q})$$

Next, we deal with these two ways separately:

- SWR: It holds that $\text{Cov}(\mathbf{x}_{\pi_p},\mathbf{x}_{\pi_q}) = 0, \forall p \neq q$ since $\mathbf{x}_{\pi_p}, \mathbf{x}_{\pi_q}$ are independent for SWR. Thus, we can get $\mathbb{E}\left\|\overline{\mathbf{x}}_\pi - \overline{\boldsymbol{x}}\right\|^2 = \frac{1}{s^2}\sum_{p=1}^{s}\text{Var}(\mathbf{x}_{\pi_p}) = \frac{\zeta^2}{s}$.

- SWOR: For $p \neq q$, we have

$$\text{Cov}(\mathbf{x}_{\pi_p},\mathbf{x}_{\pi_q}) = \mathbb{E}\left[\langle\mathbf{x}_{\pi_p} - \overline{\boldsymbol{x}}, \mathbf{x}_{\pi_q} - \overline{\boldsymbol{x}}\rangle\right] = \sum_{i=1}^{n}\sum_{j\neq i}^{n}\langle\boldsymbol{x}_i - \overline{\boldsymbol{x}}, \boldsymbol{x}_j - \overline{\boldsymbol{x}}\rangle \cdot \text{Pr}(\mathbf{x}_{\pi_p} = \boldsymbol{x}_i, \mathbf{x}_{\pi_q} = \boldsymbol{x}_j),$$

Since there are $n(n-1)$ possible combinations of $(\mathbf{x}_{\pi_p},\mathbf{x}_{\pi_q})$ and each has the same probability, we get $\text{Pr}(\mathbf{x}_{\pi_p} = \boldsymbol{x}_i, \mathbf{x}_{\pi_q} = \boldsymbol{x}_j) = \frac{1}{n(n-1)}$. As a consequence, we have

$$\begin{aligned}\text{Cov}(\mathbf{x}_{\pi_p},\mathbf{x}_{\pi_q}) &= \frac{1}{n(n-1)}\sum_{i=1}^{n}\sum_{j\neq i}^{n}[\langle\boldsymbol{x}_i - \overline{\boldsymbol{x}}, \boldsymbol{x}_j - \overline{\boldsymbol{x}}\rangle] \\ &= \frac{1}{n(n-1)}\left\|\sum_{i=1}^{n}(\boldsymbol{x}_i - \overline{\boldsymbol{x}})\right\|^2 - \frac{1}{n(n-1)}\sum_{i=1}^{n}\|\boldsymbol{x}_i - \overline{\boldsymbol{x}}\|^2 \\ &= -\frac{\zeta^2}{n-1}\end{aligned} \tag{24}$$

Thus we have $\mathbb{E}\left\|\overline{\mathbf{x}}_\pi - \overline{\boldsymbol{x}}\right\|^2 = \frac{\zeta^2}{s} - \frac{s(s-1)}{s^2}\cdot\frac{\zeta^2}{n-1} = \frac{(n-s)}{s(n-1)}\zeta^2$.

When $n$ is infinite (or large enough), we get $\frac{(\mathbb{E}\|\overline{\mathbf{x}}_\pi - \overline{\boldsymbol{x}}\|^2)_{\text{SWOR}}}{(\mathbb{E}\|\overline{\mathbf{x}}_\pi - \overline{\boldsymbol{x}}\|^2)_{\text{SWR}}} \approx 1 - \frac{s}{n}$. This constant has appeared in Karimireddy et al. (2020)'s Lemma 7 (one round progress) and Woodworth et al. (2020b)'s Section 7 (Using a Subset of Machines in Each Round). □

**Lemma 4.** *Under the same conditions of Lemma 3, use the way "sampling without replacement" and let* $b_{m,k}(i) = \begin{cases} K-1, & i \leq m-1 \\ k-1, & i = m \end{cases}$. *Then for* $S \leq M$ ($M \geq 2$), *it holds that*

$$\sum_{m=1}^{S}\sum_{k=0}^{K-1}\mathbb{E}\left[\left\|\sum_{i=1}^{m}\sum_{j=0}^{b_{m,k}(i)}(\mathbf{x}_{\pi_i} - \overline{\boldsymbol{x}})\right\|^2\right] \leq \frac{1}{2}S^2K^3\zeta^2 \tag{25}$$

*Proof.* Let us focus on the term in the following:

$$\begin{aligned}\mathbb{E}\left[\left\|\sum_{i=1}^{m}\sum_{j=0}^{b_{m,k}(i)}(\mathbf{x}_{\pi_i} - \overline{\boldsymbol{x}})\right\|^2\right] &= \mathbb{E}\left[\left\|K\sum_{i=1}^{m-1}(\mathbf{x}_{\pi_i} - \overline{\boldsymbol{x}}) + k(\mathbf{x}_{\pi_m} - \overline{\boldsymbol{x}})\right\|^2\right] \\ &= K^2\mathbb{E}\left[\left\|\sum_{i=1}^{m-1}(\mathbf{x}_{\pi_i} - \overline{\boldsymbol{x}})\right\|^2\right] + k^2\mathbb{E}\left[\|\mathbf{x}_{\pi_m} - \overline{\boldsymbol{x}}\|^2\right] + 2Kk\mathbb{E}\left[\left\langle\sum_{i=1}^{m-1}(\mathbf{x}_{\pi_i} - \overline{\boldsymbol{x}}), (\mathbf{x}_{\pi_m} - \overline{\boldsymbol{x}})\right\rangle\right]\end{aligned} \tag{26}$$

For the first term on the right hand side in (26), using (23), we have

$$K^2 \mathbb{E}\left[\left\|\sum_{i=1}^{m-1}\left(\mathbf{x}_{\pi_i} - \overline{\boldsymbol{x}}\right)\right\|^2\right] \overset{(23)}{=} \frac{(m-1)(M-(m-1))}{M-1}K^2\zeta^2.$$

For the second term on the right hand side in (26), we have

$$k^2 \mathbb{E}\left[\left\|\mathbf{x}_{\pi_m} - \overline{\boldsymbol{x}}\right\|^2\right] = k^2 \mathbb{E}\left[\left\|\mathbf{x}_{\pi_m} - \overline{\boldsymbol{x}}\right\|^2\right] = k^2\zeta^2.$$

For the third term on the right hand side in (26), we have

$$2Kk\mathbb{E}\left[\left\langle\sum_{i=1}^{m-1}\left(\mathbf{x}_{\pi_i} - \overline{\boldsymbol{x}}\right), \left(\mathbf{x}_{\pi_m} - \overline{\boldsymbol{x}}\right)\right\rangle\right] = 2Kk\sum_{i=1}^{m-1}\mathbb{E}\left[\left\langle\mathbf{x}_{\pi_i} - \overline{\boldsymbol{x}}, \mathbf{x}_{\pi_m} - \overline{\boldsymbol{x}}\right\rangle\right] \overset{(24)}{=} -\frac{2(m-1)}{M-1}Kk\zeta^2,$$

where we use (24) in the last equality, since $i \in \{1, 2, \ldots, m-1\} \neq m$. With these three preceding equations, we get

$$\mathbb{E}\left[\left\|\sum_{i=1}^{m}\sum_{j=0}^{b_{m,k}(i)}\left(\mathbf{x}_{\pi_i} - \overline{\boldsymbol{x}}\right)\right\|^2\right] = \frac{(m-1)(M-(m-1))}{M-1}K^2\zeta^2 + k^2\zeta^2 - \frac{2(m-1)}{M-1}Kk\zeta^2$$

Then summing the preceding terms over $m$ and $k$, we can get

$$\sum_{m=1}^{S}\sum_{k=0}^{K-1}\mathbb{E}\left[\left\|\sum_{i=1}^{m}\sum_{j=0}^{b_{m,k}(i)}\left(\mathbf{x}_{\pi_i} - \overline{\boldsymbol{x}}\right)\right\|^2\right]$$

$$= \frac{MK^3\zeta^2}{M-1}\sum_{m=1}^{S}(m-1) - \frac{K^3\zeta^2}{M-1}\sum_{m=1}^{S}(m-1)^2 + S\zeta^2\sum_{k=0}^{K-1}k^2 - \frac{2K\zeta^2}{M-1}\sum_{m=1}^{S}(m-1)\sum_{k=0}^{K-1}k$$

Then applying the fact that $\sum_{k=1}^{K-1}k = \frac{(K-1)K}{2}$ and $\sum_{k=1}^{K-1}k^2 = \frac{(K-1)K(2K-1)}{6}$, we can simplify the preceding equation as

$$\sum_{m=1}^{M}\sum_{k=0}^{K-1}\mathbb{E}\left\|\sum_{i=1}^{m}\sum_{j=0}^{b_{m,k}(i)}\left(\mathbf{x}_{\pi_i} - \overline{\boldsymbol{x}}\right)\right\|^2$$

$$= \frac{1}{2}SK^2(SK-1) - \frac{1}{6}SK(K^2-1) - \frac{1}{M-1}(S-1)S\left(\frac{1}{6}(2S-1)K - \frac{1}{2}\right) \leq \frac{1}{2}S^2K^3\zeta^2,$$

which is the claim of this lemma. $\qquad\square$

# D  Proofs of Theorem 1

In this section, we provide the proof of Theorem 1 for the strongly convex, general convex and non-convex cases in D.1, D.2 and D.3, respectively.

In the following proof, we consider the partial client participation setting. So we assume that $\pi = \{\pi_1, \pi_2, \ldots, \pi_M\}$ is a permutation of $\{1, 2, \ldots, M\}$ in a certain training round and only the first $S$ selected clients $\{\pi_1, \pi_2, \ldots, \pi_S\}$ will participate in this round. Without otherwise stated, we use $\mathbb{E}[\cdot]$ to represent the expectation with respect to both types of randomness (i.e., sampling data samples $\xi$ and sampling clients $\pi$).

### D.1 Strongly convex case

#### D.1.1 Finding the recursion

**Lemma 5.** *Let Assumptions 1, 2, 3b hold and assume that all the local objectives are $\mu$-strongly convex. If the learning rate satisfies $\eta \leq \frac{1}{6LSK}$, then it holds that*

$$
\mathbb{E}\left[\left\|\mathbf{x}^{(r+1)} - \mathbf{x}^*\right\|^2\right] \leq \left(1 - \frac{\mu SK\eta}{2}\right)\mathbb{E}\left[\left\|\mathbf{x}^{(r)} - \mathbf{x}^*\right\|^2\right] + 4SK\eta^2\sigma^2 + 4S^2K^2\eta^2\frac{M-S}{S(M-1)}\zeta_*^2
$$
$$
- \frac{2}{3}SK\eta\mathbb{E}\left[D_F(\mathbf{x}^{(r)}, \mathbf{x}^*)\right] + \frac{8}{3}L\eta\sum_{m=1}^{S}\sum_{k=0}^{K-1}\mathbb{E}\left[\left\|\mathbf{x}_{m,k}^{(r)} - \mathbf{x}^{(r)}\right\|^2\right] \quad (27)
$$

*Proof.* According to Algorithm 1, the overall model updates of SFL after one complete training round (with $S$ clients selected for training) is

$$
\Delta\mathbf{x} = \mathbf{x}^{(r+1)} - \mathbf{x}^{(r)} = -\eta\sum_{m=1}^{S}\sum_{k=0}^{K-1}\mathbf{g}_{\pi_m,k}^{(r)},
$$

where $\mathbf{g}_{\pi_m,k}^{(r)} = \nabla f_{\pi_m}(\mathbf{x}_{m,k}^{(r)};\xi)$ is the stochastic gradient of $F_{\pi_m}$ regarding the vector $\mathbf{x}_{m,k}^{(r)}$. Thus,

$$
\mathbb{E}\left[\Delta\mathbf{x}\right] = -\eta\sum_{m=1}^{S}\sum_{k=0}^{K-1}\mathbb{E}\left[\nabla F_{\pi_m}(\mathbf{x}_{m,k})\right]
$$

In the following, we focus on a single training round, and hence we drop the superscripts $r$ for a while, e.g., writing $\mathbf{x}_{m,k}$ to replace $\mathbf{x}_{m,k}^{(r)}$. Specially, we would like to use $\mathbf{x}$ to replace $\mathbf{x}_{1,0}^{(r)}$. Without otherwise stated, the expectation is conditioned on $\mathbf{x}^{(r)}$.

We start by substituting the overall updates:

$$
\mathbb{E}\left[\|\mathbf{x} + \Delta\mathbf{x} - \mathbf{x}^*\|^2\right]
$$
$$
= \|\mathbf{x} - \mathbf{x}^*\|^2 + 2\mathbb{E}\left[\langle\mathbf{x} - \mathbf{x}^*, \Delta\mathbf{x}\rangle\right] + \mathbb{E}\left[\|\Delta\mathbf{x}\|^2\right]
$$
$$
= \|\mathbf{x} - \mathbf{x}^*\|^2 - 2\eta\sum_{m=1}^{S}\sum_{k=0}^{K-1}\mathbb{E}\left[\langle\nabla F_{\pi_m}(\mathbf{x}_{m,k}), \mathbf{x} - \mathbf{x}^*\rangle\right] + \eta^2\mathbb{E}\left[\left\|\sum_{m=1}^{S}\sum_{k=0}^{K-1}\mathbf{g}_{\pi_m,k}\right\|^2\right] \quad (28)
$$

We can apply Lemma 2 with $\boldsymbol{x} = \mathbf{x}_{m,k}$, $\boldsymbol{y} = \mathbf{x}^*$, $\boldsymbol{z} = \mathbf{x}$ and $h = F_{\pi_m}$ for the second term on the right hand side in (28):

$$
-2\eta\sum_{m=1}^{S}\sum_{k=0}^{K-1}\mathbb{E}\left[\langle\nabla F_{\pi_m}(\mathbf{x}_{m,k}), \mathbf{x} - \mathbf{x}^*\rangle\right]
$$
$$
\leq -2\eta\sum_{m=1}^{S}\sum_{k=0}^{K-1}\mathbb{E}\left[F_{\pi_m}(\mathbf{x}) - F_{\pi_m}(\mathbf{x}^*) + \frac{\mu}{4}\|\mathbf{x} - \mathbf{x}^*\|^2 - L\|\mathbf{x}_{m,k} - \mathbf{x}\|^2\right]
$$
$$
\leq -2SK\eta D_F(\mathbf{x}, \mathbf{x}^*) - \frac{1}{2}\mu SK\eta\|\mathbf{x} - \mathbf{x}^*\|^2 + 2L\eta\sum_{m=1}^{S}\sum_{k=0}^{K-1}\mathbb{E}\left[\|\mathbf{x}_{m,k} - \mathbf{x}\|^2\right] \quad (29)
$$

For the third term on the right hand side in (28), using Jensen's inequality, we have

$$
\mathbb{E}\left[\left\|\sum_{m=1}^{S}\sum_{k=0}^{K-1}\mathbf{g}_{\pi_m,k}\right\|^2\right]
$$

$$
\leq 4\mathbb{E}\left[\left\|\sum_{m=1}^{S}\sum_{k=0}^{K-1}(\mathbf{g}_{\pi_m,k}-\nabla F_{\pi_m}(\mathbf{x}_{m,k}))\right\|^2\right] + 4\mathbb{E}\left[\left\|\sum_{m=1}^{S}\sum_{k=0}^{K-1}(\nabla F_{\pi_m}(\mathbf{x}_{m,k})-\nabla F_{\pi_m}(\mathbf{x}))\right\|^2\right]
$$

$$
+ 4\mathbb{E}\left[\left\|\sum_{m=1}^{S}\sum_{k=0}^{K-1}(\nabla F_{\pi_m}(\mathbf{x})-\nabla F_{\pi_m}(\mathbf{x}^*))\right\|^2\right] + 4\mathbb{E}\left[\left\|\sum_{m=1}^{S}\sum_{k=0}^{K-1}\nabla F_{\pi_m}(\mathbf{x}^*)\right\|^2\right] \tag{30}
$$

Seeing the data sample $\xi_{m,k}$, the stochastic gradient $\mathbf{g}_{\pi_m,k}$, the gradient $\nabla F_{\pi_m}(\xi_{m,k})$ as $\xi_i$, $\mathbf{x}_i$, $\mathbf{e}_i$ in Lemma 1 respectively and applying the result of Lemma 1, the first term on the right hand side in (30) can be bounded by $4SK\sigma^2$. For the second term on the right hand side in (30), we have

$$
4\mathbb{E}\left[\left\|\sum_{m=1}^{S}\sum_{k=0}^{K-1}(\nabla F_{\pi_m}(\mathbf{x}_{m,k})-\nabla F_{\pi_m}(\mathbf{x}))\right\|^2\right] \overset{(15)}{\leq} 4SK\sum_{m=1}^{S}\sum_{k=0}^{K-1}\mathbb{E}\left[\|\nabla F_{\pi_m}(\mathbf{x}_{m,k})-\nabla F_{\pi_m}(\mathbf{x})\|^2\right]
$$

$$
\overset{\text{Asm. }1}{\leq} 4L^2SK\sum_{m=1}^{S}\sum_{k=0}^{K-1}\mathbb{E}\left[\|\mathbf{x}_{m,k}-\mathbf{x}\|^2\right]
$$

For the third term on the right hand side in (30), we have

$$
4\mathbb{E}\left[\left\|\sum_{m=1}^{S}\sum_{k=0}^{K-1}(\nabla F_{\pi_m}(\mathbf{x})-\nabla F_{\pi_m}(\mathbf{x}^*))\right\|^2\right] \overset{(15)}{\leq} 4SK\sum_{m=1}^{S}\sum_{k=0}^{K-1}\mathbb{E}\left[\|\nabla F_{\pi_m}(\mathbf{x})-\nabla F_{\pi_m}(\mathbf{x}^*)\|^2\right]
$$

$$
\overset{(18)}{\leq} 8LSK\sum_{m=1}^{S}\sum_{k=0}^{K-1}\mathbb{E}\left[D_{F_{\pi_m}}(\mathbf{x},\mathbf{x}^*)\right]
$$

$$
= 8LS^2K^2 D_F(\mathbf{x},\mathbf{x}^*), \tag{31}
$$

where the last inequality is because $\mathbb{E}\left[D_{F_{\pi_m}}(\mathbf{x},\mathbf{x}^*)\right] = D_F(\mathbf{x},\mathbf{x}^*)$. The forth term on the right hand side in (30) can be bounded by Lemma 3 as follows:

$$
4\mathbb{E}\left[\left\|\sum_{m=1}^{S}\sum_{k=0}^{K-1}\nabla F_{\pi_m}(\mathbf{x}^*)\right\|^2\right] \overset{(23)}{\leq} 4S^2K^2\frac{M-S}{S(M-1)}\zeta_*^2.
$$

With the preceding four inequalities, we can bound the third term on the right hand side in (28):

$$
\mathbb{E}\left[\left\|\sum_{m=1}^{S}\sum_{k=0}^{K-1}\mathbf{g}_{\pi_m,k}\right\|^2\right]
$$

$$
\leq 4SK\sigma^2 + 4L^2SK\sum_{m=1}^{S}\sum_{k=0}^{K-1}\mathbb{E}\left[\|\mathbf{x}_{m,k}-\mathbf{x}\|^2\right] + 8LS^2K^2 D_F(\mathbf{x},\mathbf{x}^*) + 4S^2K^2\frac{M-S}{S(M-1)}\zeta_*^2 \tag{32}
$$

Then substituting (29) and (32) into (28), we have

$$
\mathbb{E}\left[\|\mathbf{x}+\Delta\mathbf{x}-\mathbf{x}^*\|^2\right] \leq \left(1-\frac{\mu SK\eta}{2}\right)\|\mathbf{x}-\mathbf{x}^*\|^2 + 4SK\eta^2\sigma^2 + 4S^2K^2\eta^2\frac{M-S}{S(M-1)}\zeta_*^2
$$

$$
- 2SK\eta(1-4LSK\eta)D_F(\mathbf{x},\mathbf{x}^*) + 2L\eta(1+2LSK\eta)\sum_{m=1}^{S}\sum_{k=0}^{K-1}\mathbb{E}\left[\|\mathbf{x}_{m,k}-\mathbf{x}\|^2\right]
$$

$$
\leq \left(1-\frac{\mu SK\eta}{2}\right)\|\mathbf{x}-\mathbf{x}^*\|^2 + 4SK\eta^2\sigma^2 + 4S^2K^2\eta^2\frac{M-S}{S(M-1)}\zeta_*^2
$$

$$
- \frac{2}{3}SK\eta D_F(\mathbf{x},\mathbf{x}^*) + \frac{8}{3}L\eta\sum_{m=1}^{S}\sum_{k=0}^{K-1}\mathbb{E}\left[\|\mathbf{x}_{m,k}-\mathbf{x}\|^2\right],
$$

where we use the condition that $\eta \leq \frac{1}{6LSK}$ in the last inequality. The claim of this lemma follows after recovering the superscripts and taking unconditional expectation. $\qquad\square$

### D.1.2 Bounding the client drift with Assumption 3b

Similar to the "client drift" in PFL (Karimireddy et al., 2020), we define the client drift in SFL:

$$E_r := \sum_{m=1}^{S} \sum_{k=0}^{K-1} \mathbb{E}\left[\left\|\mathbf{x}_{m,k}^{(r)} - \mathbf{x}^{(r)}\right\|^2\right] \tag{33}$$

**Lemma 6.** *Let Assumptions 1, 2, 3b hold and assume that all the local objectives are $\mu$-strongly convex. If the learning rate satisfies $\eta \leq \frac{1}{6LSK}$, then the client drift is bounded:*

$$E_r \leq \frac{9}{4}S^2K^2\eta^2\sigma^2 + \frac{9}{4}S^2K^3\eta^2\zeta_*^2 + 3LS^3K^3\eta^2\mathbb{E}\left[D_F(\mathbf{x}^{(r)}, \mathbf{x}^*)\right] \tag{34}$$

*Proof.* According to Algorithm 1, the model updates of SFL from $\mathbf{x}^{(r)}$ to $\mathbf{x}_{m,k}^{(r)}$ is

$$\mathbf{x}_{m,k}^{(r)} - \mathbf{x}^{(r)} = -\eta \sum_{i=1}^{m} \sum_{j=0}^{b_{m,k}(i)} \mathbf{g}_{\pi_i,j}^{(r)}$$

with $b_{m,k}(i) := \begin{cases} K-1, & i \leq m-1 \\ k-1, & i = m \end{cases}$. In the following, we focus on a single training round, and

hence we drop the superscripts $r$ for a while, e.g., writing $\mathbf{x}_{m,k}$ to replace $\mathbf{x}_{m,k}^{(r)}$. Specially, we would like to use $\mathbf{x}$ to replace $\mathbf{x}_{1,0}^{(r)}$. Without otherwise stated, the expectation is conditioned on $\mathbf{x}^{(r)}$.

We use Jensen's inequality to bound the term $\mathbb{E}\left[\|\mathbf{x}_{m,k} - \mathbf{x}\|^2\right] = \eta^2\mathbb{E}\left[\left\|\sum_{i=1}^{m}\sum_{j=0}^{b_{m,k}(i)} \mathbf{g}_{\pi_i,j}\right\|^2\right]$:

$$\mathbb{E}\left[\|\mathbf{x}_{m,k} - \mathbf{x}\|^2\right]$$

$$\leq 4\eta^2\mathbb{E}\left[\left\|\sum_{i=1}^{m}\sum_{j=0}^{b_{m,k}(i)}(\mathbf{g}_{\pi_i,j} - \nabla F_{\pi_i}(\mathbf{x}_{i,j}))\right\|^2\right] + 4\eta^2\mathbb{E}\left[\left\|\sum_{i=1}^{m}\sum_{j=0}^{b_{m,k}(i)}(\nabla F_{\pi_i}(\mathbf{x}_{i,j}) - \nabla F_{\pi_i}(\mathbf{x}))\right\|^2\right]$$

$$+ 4\eta^2\mathbb{E}\left[\left\|\sum_{i=1}^{m}\sum_{j=0}^{b_{m,k}(i)}(\nabla F_{\pi_i}(\mathbf{x}) - \nabla F_{\pi_i}(\mathbf{x}^*))\right\|^2\right] + 4\eta^2\mathbb{E}\left[\left\|\sum_{i=1}^{m}\sum_{j=0}^{b_{m,k}(i)}\nabla F_{\pi_i}(\mathbf{x}^*)\right\|^2\right]$$

Applying Lemma 1 to the first term and Jensen's inequality to the second, third terms on the right hand side in the preceding inequality respectively, we can get

$$\mathbb{E}\left[\|\mathbf{x}_{m,k} - \mathbf{x}\|^2\right]$$

$$\leq 4\eta^2\sum_{i=1}^{m}\sum_{j=0}^{b_{m,k}(i)}\mathbb{E}\left[\|\mathbf{g}_{\pi_i,j} - \nabla F_{\pi_i}(\mathbf{x}_{i,j})\|^2\right] + 4\eta^2\mathcal{B}_{m,k}\sum_{i=1}^{m}\sum_{j=0}^{b_{m,k}(i)}\mathbb{E}\left[\|\nabla F_{\pi_i}(\mathbf{x}_{i,j}) - \nabla F_{\pi_i}(\mathbf{x})\|^2\right]$$

$$+ 4\eta^2\mathcal{B}_{m,k}\sum_{i=1}^{m}\sum_{j=0}^{b_{m,k}(i)}\mathbb{E}\left[\|\nabla F_{\pi_i}(\mathbf{x}) - \nabla F_{\pi_i}(\mathbf{x}^*)\|^2\right] + 4\eta^2\mathbb{E}\left[\left\|\sum_{i=1}^{m}\sum_{j=0}^{b_{m,k}(i)}\nabla F_{\pi_i}(\mathbf{x}^*)\right\|^2\right] \tag{35}$$

where $\mathcal{B}_{m,k} := \sum_{i=1}^{m}\sum_{j=0}^{b_{m,k}(i)} 1 = (m-1)K + k$. The first term on the right hand side in (35) is bounded by $4\mathcal{B}_{m,k}\eta^2\sigma^2$. For the second term on the right hand side in (35), we have

$$\mathbb{E}\left[\|\nabla F_{\pi_i}(\mathbf{x}_{i,j}) - \nabla F_{\pi_i}(\mathbf{x})\|^2\right] \overset{\text{Asm. 1}}{\leq} L^2\mathbb{E}\left[\|\mathbf{x}_{i,j} - \mathbf{x}\|^2\right]$$

For the third term on the right hand side in (35), we have

$$\mathbb{E}\left[\|\nabla F_{\pi_i}(\mathbf{x}) - \nabla F_{\pi_i}(\mathbf{x}^*)\|^2\right] \overset{(18)}{\leq} 2L\mathbb{E}\left[D_{F_{\pi_i}}(\mathbf{x}, \mathbf{x}^*)\right] = 2LD_F(\mathbf{x}, \mathbf{x}^*) \tag{36}$$

As a result, we can get

$$\mathbb{E}\left[\|\mathbf{x}_{m,k} - \mathbf{x}\|^2\right] \leq 4\mathcal{B}_{m,k}\eta^2\sigma^2 + 4L^2\eta^2\mathcal{B}_{m,k}\sum_{i=1}^{m}\sum_{j=0}^{b(i)}\mathbb{E}\left[\|\mathbf{x}_{i,j} - \mathbf{x}\|^2\right] + 8L\eta^2\mathcal{B}_{m,k}^2 D_F(\mathbf{x}, \mathbf{x}^*)$$

$$+ 4\eta^2\mathbb{E}\left[\left\|\sum_{i=1}^{m}\sum_{j=0}^{b_{m,k}(i)}\nabla F_{\pi_i}(\mathbf{x}^*)\right\|^2\right]$$

Then, returning to $E_r := \sum_{m=1}^{S}\sum_{k=0}^{K-1}\mathbb{E}\left[\|\mathbf{x}_{m,k} - \mathbf{x}\|^2\right]$, we have

$$E_r \leq 4\eta^2\sigma^2\sum_{m=1}^{S}\sum_{k=0}^{K-1}\mathcal{B}_{m,k} + 4L^2\eta^2\sum_{m=1}^{S}\sum_{k=0}^{K-1}\mathcal{B}_{m,k}\sum_{i=1}^{m}\sum_{j=0}^{b_{m,k}(i)}\mathbb{E}\left[\|\mathbf{x}_{i,j} - \mathbf{x}\|^2\right]$$

$$+ 8L\eta^2\sum_{m=1}^{S}\sum_{k=0}^{K-1}\mathcal{B}_{m,k}^2 D_F(\mathbf{x}, \mathbf{x}^*) + 4\eta^2\sum_{m=1}^{S}\sum_{k=0}^{K-1}\mathbb{E}\left[\left\|\sum_{i=1}^{m}\sum_{j=0}^{b_{m,k}(i)}\nabla F_{\pi_i}(\mathbf{x}^*)\right\|^2\right]$$

Applying Lemma 4 with $\mathbf{x}_{\pi_i} = \nabla F_{\pi_i}(\mathbf{x}^*)$ and $\overline{\mathbf{x}} = \nabla F(\mathbf{x}^*) = 0$ and the fact that

$$\sum_{m=1}^{S}\sum_{k=0}^{K-1}\mathcal{B}_{m,k} = \frac{1}{2}SK(SK-1) \leq \frac{1}{2}S^2K^2,$$

$$\sum_{m=1}^{S}\sum_{k=0}^{K-1}\mathcal{B}_{m,k}^2 = \frac{1}{3}(SK-1)SK(SK-\tfrac{1}{2}) \leq \frac{1}{3}S^3K^3,$$

we can simplify the preceding inequality:

$$E_r \leq 2S^2K^2\eta^2\sigma^2 + 2L^2S^2K^2\eta^2 E_r + \frac{8}{3}LS^3K^3\eta^2 D_F(\mathbf{x}, \mathbf{x}^*) + 2S^2K^3\eta^2\zeta_*^2$$

After rearranging the preceding inequality, we get

$$(1 - 2L^2S^2K^2\eta^2)E_r \leq 2S^2K^2\eta^2\sigma^2 + 2S^2K^3\eta^2\zeta_*^2 + \frac{8}{3}LS^3K^3\eta^2 D_F(\mathbf{x}, \mathbf{x}^*)$$

Finally, using the condition that $\eta \leq \frac{1}{6LSK}$, which implies $1 - 2L^2S^2K^2\eta^2 \geq \frac{8}{9}$, we have

$$E_r \leq \frac{9}{4}S^2K^2\eta^2\sigma^2 + \frac{9}{4}S^2K^3\eta^2\zeta_*^2 + 3LS^3K^3\eta^2 D_F(\mathbf{x}, \mathbf{x}^*).$$

The claim follows after recovering the superscripts and taking unconditional expectations. $\qquad\square$

### D.1.3 Tuning the learning rate

Here we make a clear version of Karimireddy et al. (2020)'s Lemma 1 (linear convergence rate) based on Stich (2019b); Stich and Karimireddy (2019)'s works.

**Lemma 7** (Karimireddy et al. (2020)). *Two non-negative sequences $\{r_t\}_{t\geq 0}$, $\{s_t\}_{t\geq 0}$, which satisfies the relation*

$$r_{t+1} \leq (1 - a\gamma_t)r_t - b\gamma_t s_t + c\gamma_t^2, \tag{37}$$

*for all $t \geq 0$ and for parameters $b > 0$, $a, c \geq 0$ and non-negative learning rates $\{\gamma_t\}_{t\geq 0}$ with $\gamma_t \leq \frac{1}{d}$, $\forall t \geq 0$, for a parameter $d \geq a$, $d > 0$.*

***Selection of weights for average.*** *Then there exists a constant learning rate $\gamma_t = \gamma \le \frac{1}{d}$ and the weights $w_t := (1 - a\gamma)^{-(t+1)}$ and $W_T := \sum_{t=0}^{T} w_t$, making it hold that:*

$$\Psi_T = \frac{b}{W_T} \sum_{t=0}^{T} s_t w_t \le 3ar_0(1 - a\gamma)^{(T+1)} + c\gamma \le 3ar_0 \exp\left[-a\gamma(T+1)\right] + c\gamma. \qquad (38)$$

***Tuning the learning rate carefully.*** *By tuning the learning rate in* (38), *for $(T+1) \ge \frac{1}{2a\gamma}$, we have*

$$\Psi_T = \tilde{\mathcal{O}}\left(ar_0 \exp\left(-\frac{aT}{d}\right) + \frac{c}{aT}\right). \qquad (39)$$

*Proof.* We start by rearranging (37) and multiplying both sides with $w_t$:

$$bs_t w_t \le \frac{w_t(1 - a\gamma)r_t}{\gamma} - \frac{w_t r_{t+1}}{\gamma} + c\gamma w_t = \frac{w_{t-1} r_t}{\gamma} - \frac{w_t r_{t+1}}{\gamma} + c\gamma w_t\,.$$

By summing from $t = 0$ to $t = T$, we obtain a telescoping sum:

$$\frac{b}{W_T} \sum_{t=0}^{T} s_t w_t \le \frac{1}{\gamma W_T}\left(w_0(1 - a\gamma)r_0 - w_T r_{T+1}\right) + c\gamma\,,$$

and hence

$$\Psi_T = \frac{b}{W_T} \sum_{t=0}^{T} s_t w_t \le \frac{b}{W_T} \sum_{t=0}^{T} s_t w_t + \frac{w_T r_{T+1}}{\gamma W_T} \le \frac{r_0}{\gamma W_T} + c\gamma \qquad (40)$$

Note that the proof of Stich (2019b)'s Lemma 2 used $W_T \ge w_T = (1 - a\gamma)^{-(T+1)}$ to estimate $W_T$. It is reasonable given that $w_T$ is extremely larger than all the terms $w_t$ $(t < T)$ when $T$ is large. Yet Karimireddy et al. (2020) goes further, showing that $W_T$ can be estimated more precisely:

$$W_T = \sum_{0}^{T} w_t = (1 - a\gamma)^{-(T+1)} \sum_{t=0}^{T} (1 - a\gamma)^t = (1 - a\gamma)^{-(T+1)}\left(\frac{1 - (1 - a\gamma)^{T+1}}{a\gamma}\right)$$

When $(T+1) \ge \frac{1}{2a\gamma}$, $(1 - a\gamma)^{T+1} \le \exp(-a\gamma(T+1)) \le e^{-\frac{1}{2}} \le \frac{2}{3}$, so it follows that

$$W_T = (1 - a\gamma)^{-(T+1)}\left(\frac{1 - (1 - a\gamma)^{T+1}}{a\gamma}\right) \ge \frac{(1 - a\gamma)^{-(T+1)}}{3a\gamma}$$

With the estimates

- $W_T = (1 - a\gamma)^{-(T+1)} \sum_{t=0}^{T}(1 - a\gamma)^t \le \frac{w_T}{a\gamma}$ (here we leverage $a\gamma \le \frac{a}{d} \le 1$),

- and $W_T \ge \frac{(1-a\gamma)^{-(T+1)}}{3a\gamma}$,

we can further simplify (40):

$$\Psi_T \le 3ar_0(1 - a\gamma)^{(T+1)} + c\gamma \le 3ar_0 \exp\left[-a\gamma(T+1)\right] + c\gamma$$

which is the first result of this lemma.

Now the lemma follows by carefully tuning $\gamma$ in (38). Consider the two cases:

- If $\frac{1}{d} > \frac{\ln(\max\{2, a^2 r_0 T/c\})}{aT}$ then we choose $\gamma = \frac{\ln(\max\{2, a^2 r_0 T/c\})}{aT}$ and get that

$$\tilde{\mathcal{O}}\left(3ar_0 \exp[-\ln(\max\{2, a^2 r_0 T/c\})]\right) + \tilde{\mathcal{O}}\left(\frac{c}{aT}\right) = \tilde{\mathcal{O}}\left(\frac{c}{aT}\right),$$

as in case $2 \ge a^2 r_0 T/c$ it holds $ar_0 \le \frac{2c}{aT}$.

- If otherwise $\frac{1}{2a(T+1)} \leq \frac{1}{d} \leq \frac{\ln(\max\{2,a^2r_0T/c\})}{aT}$ (Note $\frac{1}{2a(T+1)} \leq \frac{\ln(2)}{aT} \leq \frac{\ln(\max\{2,a^2r_0T/c\})}{aT}$) then we pick $\gamma = \frac{1}{d}$ and get that

$$3ar_0 \exp\left(-\frac{aT}{d}\right) + \frac{c}{d} \leq 3ar_0 \exp\left(-\frac{aT}{d}\right) + \frac{c\ln(\max\{2, a^2r_0T/c\})}{aT} = \tilde{\mathcal{O}}\left(ar_0 \exp\left(-\frac{aT}{d}\right) + \frac{c}{aT}\right)$$

Combining these two cases, we get

$$\Psi_T = \tilde{\mathcal{O}}\left(ar_0 \exp\left(-\frac{aT}{d}\right) + \frac{c}{aT}\right)$$

Note that this lemma holds when $(T+1) \geq \frac{1}{2a\gamma}$, so it restricts the value of $T$, while there is no such restriction in Stich (2019b)'s Lemma 2. $\qquad\square$

### D.1.4 Proof of strongly convex case of Theorem 1 and Corollary 1

*Proof of the strongly convex case of Theorem 1.* Substituting (34) into (27) and using $\eta \leq \frac{1}{6LSK}$, we can simplify the recursion as,

$$\mathbb{E}\left[\left\|\mathbf{x}^{(r+1)} - \mathbf{x}^*\right\|^2\right] \leq \left(1 - \frac{\mu SK\eta}{2}\right)\mathbb{E}\left[\left\|\mathbf{x}^{(r)} - \mathbf{x}^*\right\|^2\right] - \frac{1}{3}SK\eta\mathbb{E}\left[D_F(\mathbf{x}^{(r)}, \mathbf{x}^*)\right]$$
$$+ 4SK\eta^2\sigma^2 + 4S^2K^2\eta^2\frac{M-S}{S(M-1)}\zeta_*^2 + 6LS^2K^2\eta^3\sigma^2 + 6LS^2K^3\eta^3\zeta_*^2$$

Let $\tilde{\eta} = MK\eta$, we have

$$\mathbb{E}\left[\left\|\mathbf{x}^{(r+1)} - \mathbf{x}^*\right\|^2\right] \leq \left(1 - \frac{\mu\tilde{\eta}}{2}\right)\mathbb{E}\left[\left\|\mathbf{x}^{(r)} - \mathbf{x}^*\right\|^2\right] - \frac{\tilde{\eta}}{3}\mathbb{E}\left[D_F(\mathbf{x}^{(r)}, \mathbf{x}^*)\right]$$
$$+ \frac{4\tilde{\eta}^2\sigma^2}{SK} + \frac{4\tilde{\eta}^2(M-S)\zeta_*^2}{S(M-1)} + \frac{6L\tilde{\eta}^3\sigma^2}{SK} + \frac{6L\tilde{\eta}^3\zeta_*^2}{S} \tag{41}$$

Applying Lemma 7 with $t = r$ ($T = R$), $\gamma = \tilde{\eta}$, $r_t = \mathbb{E}\left[\left\|\mathbf{x}^{(r)} - \mathbf{x}^*\right\|^2\right]$, $a = \frac{\mu}{2}$, $b = \frac{1}{3}$, $s_t = \mathbb{E}\left[D_F(\mathbf{x}^{(r)}, \mathbf{x}^*)\right]$, $w_t = (1 - \frac{\mu\tilde{\eta}}{2})^{-(r+1)}$, $c_1 = \frac{4\sigma^2}{SK} + \frac{4(M-S)\zeta_*^2}{S(M-1)}$, $c_2 = \frac{6L\sigma^2}{SK} + \frac{6L\zeta_*^2}{S}$ and $\frac{1}{d} = \frac{1}{6L}$ ($\tilde{\eta} = MK\eta \leq \frac{1}{6L}$), it follows that

$$\mathbb{E}\left[F(\bar{\mathbf{x}}^{(R)}) - F(\mathbf{x}^*)\right] \leq \frac{1}{W_R}\sum_{r=0}^{R} w_r\mathbb{E}\left[F(\mathbf{x}^{(r)}) - F(\mathbf{x}^*)\right]$$
$$\leq \frac{9}{2}\mu\left\|\mathbf{x}^{(0)} - \mathbf{x}^*\right\|^2 \exp\left(-\frac{1}{2}\mu\tilde{\eta}R\right) + \frac{12\tilde{\eta}\sigma^2}{SK} + \frac{12\tilde{\eta}(M-S)\zeta_*^2}{S(M-1)} + \frac{18L\tilde{\eta}^2\sigma^2}{SK} + \frac{18L\tilde{\eta}^2\zeta_*^2}{S} \tag{42}$$

where $\bar{\mathbf{x}}^{(R)} = \frac{1}{W_R}\sum_{r=0}^{R} w_r\mathbf{x}^{(r)}$ and we use Jensen's inequality ($F$ is convex) in the first inequality. Note that there are no terms containing $\gamma^3$ in Lemma 7. As the terms containing $\gamma^3$ is not the determinant factor for the convergence rate, Lemma 7 can also be applied to this case (Karimireddy et al., 2020; Koloskova et al., 2020). Thus, by tuning the learning rate carefully, we get

$$\mathbb{E}\left[F(\bar{\mathbf{x}}^{(R)}) - F(\mathbf{x}^*)\right] = \tilde{\mathcal{O}}\left(\mu D^2 \exp\left(-\frac{\mu R}{12L}\right) + \frac{\sigma^2}{\mu SKR} + \frac{(M-S)\zeta_*^2}{\mu SR(M-1)} + \frac{L\sigma^2}{\mu^2 SKR^2} + \frac{L\zeta_*^2}{\mu^2 SR^2}\right) \tag{43}$$

where $D := \left\|\mathbf{x}^{(0)} - \mathbf{x}^*\right\|$. Eq. (42) and Eq. (43) are the upper bounds with partial client participation. In particular, when $S = M$, we can get the claim of the strongly convex case of Theorem 1 and Corollary 1. $\qquad\square$

## D.2 General convex case

### D.2.1 Tuning the learning rate

**Lemma 8** ([Koloskova et al. (2020)](#)). *Two non-negative sequences $\{r_t\}_{t\geq 0}$, $\{s_t\}_{t\geq 0}$, which satisfies the relation*

$$r_{t+1} \leq r_t - b\gamma_t s_t + c_1\gamma_t^2 + c_2\gamma_t^3$$

*for all $t \geq 0$ and for parameters $b > 0$, $c_1, c_2 \geq 0$ and non-negative learning rates $\{\gamma_t\}_{t\geq 0}$ with $\gamma_t \leq \frac{1}{d}$, $\forall t \geq 0$, for a parameter $d > 0$.*

***Selection of weights for average.*** *Then there exists a constant learning rate $\gamma = \gamma_t \leq \frac{1}{d}$ and the weights $w_t = 1$ and $W_T = \sum_{t=0}^{T} w_t$, making it hod that:*

$$\Psi_T := \frac{b}{T+1} \sum_{t=0}^{T} s_t \leq \frac{r_0}{\gamma(T+1)} + c_1\gamma + c_2\gamma^2 \tag{44}$$

***Tuning the learning rate carefully.*** *By tuning the learning rate carefully in* [(44)](#)*, we have*

$$\Psi_T \leq 2c_1^{\frac{1}{2}} \left(\frac{r_0}{T+1}\right)^{\frac{1}{2}} + 2c_2^{\frac{1}{3}} \left(\frac{r_0}{T+1}\right)^{\frac{2}{3}} + \frac{dr_0}{T+1}. \tag{45}$$

*Proof.* For constant learning rates $\gamma_t = \gamma$ we can derive the estimate

$$\Psi_T = \frac{1}{\gamma(T+1)} \sum_{t=0}^{T} (r_t - r_{t+1}) + c_1\gamma + c_2\gamma^2 \leq \frac{r_0}{\gamma(T+1)} + c_1\gamma + c_2\gamma^2,$$

which is the first result [(44)](#) of this lemma. Let $\frac{r_0}{\gamma(T+1)} = c_1\gamma$ and $\frac{r_0}{\gamma(T+1)} = c_2\gamma^2$, yielding two choices of $\gamma$, $\gamma = \left(\frac{r_0}{c_1(T+1)}\right)^{\frac{1}{2}}$ and $\gamma = \left(\frac{r_0}{c_2(T+1)}\right)^{\frac{1}{3}}$. Then choosing $\gamma = \min\left\{\left(\frac{r_0}{c_1(T+1)}\right)^{\frac{1}{2}}, \left(\frac{r_0}{c_2(T+1)}\right)^{\frac{1}{3}}, \frac{1}{d}\right\} \leq \frac{1}{d}$, there are three cases:

- If $\gamma = \frac{1}{d}$, which implies that $\gamma = \frac{1}{d} \leq \left(\frac{r_0}{c_1(T+1)}\right)^{\frac{1}{2}}$ and $\gamma = \frac{1}{d} \leq \left(\frac{r_0}{c_2(T+1)}\right)^{\frac{1}{3}}$, then

$$\Psi_T \leq \frac{dr_0}{T+1} + \frac{c_1}{d} + \frac{c_2}{d^2} \leq \frac{dr_0}{T+1} + c_1^{\frac{1}{2}}\left(\frac{r_0}{T+1}\right)^{\frac{1}{2}} + c_2^{\frac{1}{3}}\left(\frac{r_0}{T+1}\right)^{\frac{2}{3}}$$

- If $\gamma = \left(\frac{r_0}{c_1(T+1)}\right)^{\frac{1}{2}}$, which implies that $\gamma = \left(\frac{r_0}{c_1(T+1)}\right)^{\frac{1}{2}} \leq \left(\frac{r_0}{c_2(T+1)}\right)^{\frac{1}{3}}$, then

$$\Psi_T \leq 2c_1\left(\frac{r_0}{c_1(T+1)}\right)^{\frac{1}{2}} + c_2\left(\frac{r_0}{c_1(T+1)}\right) \leq 2c_1^{\frac{1}{2}}\left(\frac{r_0}{T+1}\right)^{\frac{1}{2}} + c_2^{\frac{1}{3}}\left(\frac{r_0}{T+1}\right)^{\frac{2}{3}}$$

- If $\gamma = \left(\frac{r_0}{c_2(T+1)}\right)^{\frac{1}{3}}$, which implies that $\gamma = \left(\frac{r_0}{c_2(T+1)}\right)^{\frac{1}{3}} \leq \left(\frac{r_0}{c_1(T+1)}\right)^{\frac{1}{2}}$, then

$$\Psi_T \leq c_1\left(\frac{r_0}{c_2(T+1)}\right)^{\frac{1}{3}} + 2c_2^{\frac{1}{3}}\left(\frac{r_0}{T+1}\right)^{\frac{2}{3}} \leq c_1^{\frac{1}{2}}\left(\frac{r_0}{T+1}\right)^{\frac{1}{2}} + 2c_2^{\frac{1}{3}}\left(\frac{r_0}{T+1}\right)^{\frac{2}{3}}$$

Combining these three cases, we get the second result of this lemma. $\square$

### D.2.2 Proof of general convex case of Theorem 1 and Corollary 1

*Proof of the general convex case of Theorem 1.* Letting $\mu = 0$ in (41), we get the recursion of the general convex case,

$$\mathbb{E}\left[\left\|\mathbf{x}^{(r+1)} - \mathbf{x}^*\right\|^2\right] \leq \mathbb{E}\left[\left\|\mathbf{x}^{(r)} - \mathbf{x}^*\right\|^2\right] - \frac{\tilde{\eta}}{3}\mathbb{E}\left[D_F(\mathbf{x}^{(r)}, \mathbf{x}^*)\right]$$
$$+ \frac{4\tilde{\eta}^2\sigma^2}{SK} + \frac{4\tilde{\eta}^2(M-S)\zeta_*^2}{S(M-1)} + \frac{6L\tilde{\eta}^3\sigma^2}{SK} + \frac{6L\tilde{\eta}^3\zeta_*^2}{S}$$

Applying Lemma 8 with $t = r$ ($T = R$), $\gamma = \tilde{\eta}$, $r_t = \mathbb{E}\left[\left\|\mathbf{x}^{(r)} - \mathbf{x}^*\right\|^2\right]$, $b = \frac{1}{3}$, $s_t = \mathbb{E}\left[D_F(\mathbf{x}^{(r)}, \mathbf{x}^*)\right]$, $w_t = 1$, $c_1 = \frac{4\sigma^2}{SK} + \frac{4(M-S)\zeta_*^2}{S(M-1)}$, $c_2 = \frac{6L\sigma^2}{SK} + \frac{6L\zeta_*^2}{S}$ and $\frac{1}{d} = \frac{1}{6L}$ ($\tilde{\eta} = MK\eta \leq \frac{1}{6L}$), it follows that

$$\mathbb{E}\left[F(\bar{\mathbf{x}}^{(R)}) - F(\mathbf{x}^*)\right] \leq \frac{1}{W_R}\sum_{r=0}^{R} w_r\left(F(\mathbf{x}^{(r)}) - F(\mathbf{x}^*)\right)$$
$$\leq \frac{3\left\|\mathbf{x}^{(0)} - \mathbf{x}^*\right\|^2}{\tilde{\eta}R} + \frac{12\tilde{\eta}\sigma^2}{SK} + \frac{12\tilde{\eta}(M-S)\zeta_*^2}{S(M-1)} + \frac{18L\tilde{\eta}^2\sigma^2}{SK} + \frac{18L\tilde{\eta}^2\zeta_*^2}{S} \quad (46)$$

where $\bar{\mathbf{x}}^{(R)} = \frac{1}{W_R}\sum_{r=0}^{R} w_r\mathbf{x}^{(r)}$ and we use Jensen's inequality ($F$ is convex) in the first inequality. By tuning the learning rate carefully, we get

$$F(\bar{\mathbf{x}}^{(R)}) - F(\mathbf{x}^*) = \mathcal{O}\left(\frac{\sigma D}{\sqrt{SKR}} + \sqrt{1 - \frac{S}{M}} \cdot \frac{\zeta_* D}{\sqrt{SR}} + \frac{\left(L\sigma^2 D^4\right)^{1/3}}{(SK)^{1/3}R^{2/3}} + \frac{\left(L\zeta_*^2 D^4\right)^{1/3}}{S^{1/3}R^{2/3}} + \frac{LD^2}{R}\right) \quad (47)$$

where $D := \left\|\mathbf{x}^{(0)} - \mathbf{x}^*\right\|$. Eq. (46) and Eq. (47) are the upper bounds with partial client participation. In particular, when $S = M$, we can get the claim of the strongly convex case of Theorem 1 and Corollary 1. $\qquad\square$

### D.3 Non-convex case

**Lemma 9.** *Let Assumptions 1, 2, 3b hold. If the learning rate satisfies $\eta \leq \frac{1}{6LSK}$, then it holds that*

$$\mathbb{E}\left[F(\mathbf{x}^{(r+1)}) - F(\mathbf{x}^{(r)})\right] \leq -\frac{SK\eta}{2}\mathbb{E}\left[\left\|\nabla F(\mathbf{x}^{(r)})\right\|^2\right] + LSK\eta^2\sigma^2$$
$$+ \frac{L^2\eta}{2}\sum_{m=1}^{S}\sum_{k=0}^{K-1}\mathbb{E}\left[\left\|\mathbf{x}_{m,k}^{(r)} - \mathbf{x}^{(r)}\right\|^2\right] \quad (48)$$

*Proof.* According to Algorithm 1, the overall model updates of SFL after one complete training round (with $S$ clients selected for training) is

$$\Delta\mathbf{x} = \mathbf{x}^{(r+1)} - \mathbf{x}^{(r)} = -\eta\sum_{m=1}^{S}\sum_{k=0}^{K-1}\mathbf{g}_{\pi_m,k}^{(r)},$$

where $\mathbf{g}_{\pi_m,k}^{(r)} = \nabla f_{\pi_m}(\mathbf{x}_{m,k}^{(r)}; \xi)$ is the stochastic gradient of $F_{\pi_m}$ regarding the vector $\mathbf{x}_{m,k}^{(r)}$. Thus,

$$\mathbb{E}\left[\Delta\mathbf{x}\right] = -\eta\sum_{m=1}^{S}\sum_{k=0}^{K-1}\mathbb{E}\left[\nabla F_{\pi_m}(\mathbf{x}_{m,k})\right]$$

In the following, we focus on a single training round, and hence we drop the superscripts $r$ for a while, e.g., writing $\mathbf{x}_{m,k}$ to replace $\mathbf{x}_{m,k}^{(r)}$. Specially, we would like to use $\mathbf{x}$ to replace $\mathbf{x}_{1,0}^{(r)}$. Without otherwise stated, the expectation is conditioned on $\mathbf{x}^{(r)}$.

Starting from the smoothness of $F$ (applying Eq. (16), $D_F(\boldsymbol{x}, \boldsymbol{y}) \leq \frac{L}{2} \|\boldsymbol{x} - \boldsymbol{y}\|^2$ with $\boldsymbol{x} = \mathbf{x} + \Delta\mathbf{x}$, $\boldsymbol{y} = \mathbf{x}$), and substituting the overall updates, we have

$$
\mathbb{E}\left[F(\mathbf{x} + \Delta\mathbf{x}) - F(\mathbf{x})\right]
$$

$$
\leq \mathbb{E}\left[\langle \nabla F(\mathbf{x}), \Delta\mathbf{x}\rangle\right] + \frac{L}{2}\mathbb{E}\left[\|\Delta\mathbf{x}\|^2\right]
$$

$$
\leq -\eta \sum_{m=1}^{S} \sum_{k=0}^{K-1} \mathbb{E}\left[\langle \nabla F(\mathbf{x}), \nabla F_{\pi_m}(\mathbf{x}_{m,k})\rangle\right] + \frac{L\eta^2}{2}\mathbb{E}\left[\left\|\sum_{m=1}^{S} \sum_{k=0}^{K-1} \mathbf{g}_{\pi_m,k}\right\|^2\right] \quad (49)
$$

For the first term on the right hand side in (49), using the fact that $2\langle a, b\rangle = \|a\|^2 + \|b\|^2 - \|a - b\|^2$ with $a = \nabla F(\mathbf{x})$ and $b = \nabla F_{\pi_m}(\mathbf{x}_{m,k})$, we have

$$
-\eta \sum_{m=1}^{S} \sum_{k=0}^{K-1} \mathbb{E}\left[\langle \nabla F(\mathbf{x}), \nabla F_{\pi_m}(\mathbf{x}_{m,k})\rangle\right]
$$

$$
= -\frac{\eta}{2} \sum_{m=1}^{S} \sum_{k=0}^{K-1} \mathbb{E}\left[\|\nabla F(\mathbf{x})\|^2 + \|\nabla F_{\pi_m}(\mathbf{x}_{m,k})\|^2 - \|\nabla F_{\pi_m}(\mathbf{x}_{m,k}) - \nabla F(\mathbf{x})\|^2\right]
$$

$$
\overset{\text{Asm. 1}}{\leq} -\frac{SK\eta}{2}\|\nabla F(\mathbf{x})\|^2 - \frac{\eta}{2}\sum_{m=1}^{S}\sum_{k=0}^{K-1}\mathbb{E}\left[\|\nabla F_{\pi_m}(\mathbf{x}_{m,k})\|^2\right] + \frac{L^2\eta}{2}\sum_{m=1}^{S}\sum_{k=0}^{K-1}\mathbb{E}\left[\|\mathbf{x}_{m,k} - \mathbf{x}\|^2\right]
$$

$$
\tag{50}
$$

For the third term on the right hand side in (49), using Jensen's inequality, we have

$$
\frac{L\eta^2}{2}\mathbb{E}\left[\left\|\sum_{m=1}^{S}\sum_{k=0}^{K-1}\mathbf{g}_{\pi_m,k}\right\|^2\right]
$$

$$
\leq L\eta^2\mathbb{E}\left[\left\|\sum_{m=1}^{S}\sum_{k=0}^{K-1}\mathbf{g}_{\pi_m,k} - \sum_{m=1}^{S}\sum_{k=0}^{K-1}\nabla F_{\pi_m}(\mathbf{x}_{m,k})\right\|^2\right] + L\eta^2\mathbb{E}\left[\left\|\sum_{m=1}^{S}\sum_{k=0}^{K-1}\nabla F_{\pi_m}(\mathbf{x}_{m,k})\right\|^2\right]
$$

$$
\leq LSK\eta^2\sigma^2 + LSK\eta^2\sum_{m=1}^{S}\sum_{k=0}^{K-1}\mathbb{E}\left[\|\nabla F_{\pi_m}(\mathbf{x}_{m,k})\|^2\right], \quad (51)
$$

where we apply Lemma 1 by seeing the data sample $\xi_{m,k}$, the stochastic gradient $\mathbf{g}_{\pi_m,k}$, the gradient $\nabla F_{\pi_m}(\xi_{m,k})$ as $\xi_i$, $\mathbf{x}_i$, $\mathbf{e}_i$ respectively in Lemma 1 for the first term and Jensen's inequality for the second term in the preceding inequality.

Substituting (50) and (51) into (49), we have

$$
\mathbb{E}\left[F(\mathbf{x} + \Delta\mathbf{x}) - F(\mathbf{x})\right] \leq -\frac{SK\eta}{2}\|\nabla F(\mathbf{x})\|^2 + LSK\eta^2\sigma^2 + \frac{L^2\eta}{2}\sum_{m=1}^{S}\sum_{k=0}^{K-1}\mathbb{E}\left[\|\mathbf{x}_{m,k} - \mathbf{x}\|^2\right]
$$

$$
- \frac{\eta}{2}(1 - 2LSK\eta)\sum_{m=1}^{S}\sum_{k=0}^{K-1}\mathbb{E}\left[\|\nabla F_{\pi_m}(\mathbf{x}_{m,k})\|^2\right]
$$

Since $\eta \leq \frac{1}{6LSK}$, the last term on the right hand side in the preceding inequality is negative. Then

$$
\mathbb{E}\left[F(\mathbf{x} + \Delta\mathbf{x}) - F(\mathbf{x})\right] \leq -\frac{SK\eta}{2}\|\nabla F(\mathbf{x})\|^2 + LSK\eta^2\sigma^2 + \frac{L^2\eta}{2}\sum_{m=1}^{S}\sum_{k=0}^{K-1}\mathbb{E}\left[\|\mathbf{x}_{m,k} - \mathbf{x}\|^2\right]
$$

The claim follows after recovering the superscripts and taking unconditional expectation. $\qquad\square$

### D.3.1 Bounding the client drift with Assumption 3a

Since Eq. (18), which holds only for convex functions, is used in the proof of Lemma 6 (i.e., Eq. (36)), we cannot use the result of Lemma 6. Next, we use Assumption 3a to bound the client drift (defined in (33)).

**Lemma 10.** *Let Assumptions 1, 2, 3a hold. If the learning rate satisfies $\eta \leq \frac{1}{6LSK}$, then the client drift is bounded:*

$$E_r \leq \frac{9}{4}S^2K^2\eta^2\sigma^2 + \frac{9}{4}S^2K^3\eta^2\zeta^2 + \left(\frac{9}{4}\beta^2S^2K^3\eta^2 + \frac{3}{2}S^3K^3\eta^2\right)\mathbb{E}\left[\left\|\nabla F(\mathbf{x}^{(r)})\right\|^2\right] \quad (52)$$

*Proof.* According to Algorithm 1, the model updates of SFL from $\mathbf{x}^{(r)}$ to $\mathbf{x}_{m,k}^{(r)}$ is

$$\mathbf{x}_{m,k}^{(r)} - \mathbf{x}^{(r)} = -\eta \sum_{i=1}^{m} \sum_{j=0}^{b_{m,k}(i)} \mathbf{g}_{\pi_i,j}^{(r)}$$

with $b_{m,k}(i) := \begin{cases} K-1, & i \leq m-1 \\ k-1, & i = m \end{cases}$. In the following, we focus on a single training round, and hence we drop the superscripts $r$ for a while, e.g., writing $\mathbf{x}_{m,k}$ to replace $\mathbf{x}_{m,k}^{(r)}$. Specially, we would like to use $\mathbf{x}$ to replace $\mathbf{x}_{1,0}^{(r)}$. Without otherwise stated, the expectation is conditioned on $\mathbf{x}^{(r)}$.

We use Jensen's inequality to bound the term $\mathbb{E}\left[\|\mathbf{x}_{m,k} - \mathbf{x}\|^2\right] = \eta^2\mathbb{E}\left[\left\|\sum_{i=1}^{m}\sum_{j=0}^{b_{m,k}(i)}\mathbf{g}_{\pi_i,j}\right\|^2\right]$:

$$\mathbb{E}\left[\|\mathbf{x}_{m,k} - \mathbf{x}\|^2\right]$$

$$\leq 4\eta^2\mathbb{E}\left[\left\|\sum_{i=1}^{m}\sum_{j=0}^{b_{m,k}(i)}(\mathbf{g}_{\pi_i,j} - \nabla F_{\pi_i}(\mathbf{x}_{i,j}))\right\|^2\right] + 4\eta^2\mathbb{E}\left[\left\|\sum_{i=1}^{m}\sum_{j=0}^{b_{m,k}(i)}(\nabla F_{\pi_i}(\mathbf{x}_{i,j}) - \nabla F_{\pi_i}(\mathbf{x}))\right\|^2\right]$$

$$+ 4\eta^2\underbrace{\mathbb{E}\left[\left\|\sum_{i=1}^{m}\sum_{j=0}^{b_{m,k}(i)}(\nabla F_{\pi_i}(\mathbf{x}) - \nabla F(\mathbf{x}))\right\|^2\right]}_{T_1} + 4\eta^2\mathbb{E}\left[\left\|\sum_{i=1}^{m}\sum_{j=0}^{b_{m,k}(i)}\nabla F(\mathbf{x})\right\|^2\right]$$

Applying Lemma 1, Jensen's inequality and Jensen's inequality to the first, third and forth terms on the right hand side in the preceding inequality respectively, we can get

$$\mathbb{E}\left[\|\mathbf{x}_{m,k} - \mathbf{x}\|^2\right]$$

$$\leq 4\eta^2\sum_{i=1}^{m}\sum_{j=0}^{b_{m,k}(i)}\mathbb{E}\left[\|\mathbf{g}_{\pi_i,j} - \nabla F_{\pi_i}(\mathbf{x}_{i,j})\|^2\right] + 4\eta^2\mathcal{B}_{m,k}\sum_{i=1}^{m}\sum_{j=0}^{b_{m,k}(i)}\mathbb{E}\left[\|\nabla F_{\pi_i}(\mathbf{x}_{i,j}) - \nabla F_{\pi_i}(\mathbf{x})\|^2\right]$$

$$+ 4\eta^2 T_1 + 4\mathcal{B}_{m,k}^2\eta^2\|\nabla F(\mathbf{x})\|^2 \quad (53)$$

where $\mathcal{B}_{m,k} := \sum_{i=1}^{m}\sum_{j=0}^{b_{m,k}(i)} 1 = (m-1)K + k$. The first term on the right hand side in (53) is bounded by $4\mathcal{B}_{m,k}\eta^2\sigma^2$ with Assumption 2. The second term on the right hand side in (53) can be bounded by $4L^2\eta^2\mathcal{B}_{m,k}\sum_{i=1}^{m}\sum_{j=0}^{b_{m,k}(i)}\mathbb{E}\left[\|\mathbf{x}_{i,j} - \mathbf{x}\|^2\right]$ with Assumption 1. Then, we have

$$\mathbb{E}\left[\|\mathbf{x}_{m,k} - \mathbf{x}\|^2\right] \leq 4\mathcal{B}_{m,k}\eta^2\sigma^2 + 4L^2\eta^2\mathcal{B}_{m,k}\sum_{i=1}^{m}\sum_{j=0}^{b_{m,k}(i)}\mathbb{E}\left[\|\mathbf{x}_{i,j} - \mathbf{x}\|^2\right] + 4\eta^2 T_1 + 4\mathcal{B}_{m,k}^2\eta^2\|\nabla F(\mathbf{x})\|^2$$

Then, returning to $E_r := \sum_{m=1}^{S}\sum_{k=0}^{K-1}\mathbb{E}\left[\|\mathbf{x}_{m,k} - \mathbf{x}\|^2\right]$, we have

$$E_r \leq 4\eta^2\sigma^2\sum_{m=1}^{S}\sum_{k=0}^{K-1}\mathcal{B}_{m,k} + 4L^2\eta^2\sum_{m=1}^{S}\sum_{k=0}^{K-1}\mathcal{B}_{m,k}\sum_{i=1}^{m}\sum_{j=0}^{b_{m,k}(i)}\mathbb{E}\left[\|\mathbf{x}_{i,j} - \mathbf{x}\|^2\right]$$

$$+ 4\eta^2\sum_{m=1}^{S}\sum_{k=0}^{K-1}\mathbb{E}\left[\left\|\sum_{i=1}^{m}\sum_{j=0}^{b_{m,k}(i)}(\nabla F_{\pi_i}(\mathbf{x}) - \nabla F(\mathbf{x}))\right\|^2\right] + 4\eta^2\sum_{m=1}^{S}\sum_{k=0}^{K-1}\mathcal{B}_{m,k}^2\|\nabla F(\mathbf{x})\|^2$$

Applying Lemma 4 with $\mathbf{x}_{\pi_i} = \nabla F_{\pi_i}(\mathbf{x})$ and $\overline{\mathbf{x}} = \nabla F(\mathbf{x})$ to the third term and $\sum_{m=1}^{S} \sum_{k=0}^{K-1} \mathcal{B}_{m,k} \leq \frac{1}{2} S^2 K^2$ and $\sum_{m=1}^{S} \sum_{k=0}^{K-1} \mathcal{B}_{m,k}^2 \leq \frac{1}{3} S^3 K^3$ to the other terms on the right hand side in the preceding inequality, we can simplify it:

$$E_r \leq 2S^2 K^2 \eta^2 \sigma^2 + 2L^2 S^2 K^2 \eta^2 E_r + 2S^2 K^3 \eta^2 \left( \frac{1}{M} \sum_{i=1}^{M} \|\nabla F_i(\mathbf{x}) - \nabla F(\mathbf{x})\|^2 \right) + \frac{4}{3} S^3 K^3 \eta^2 \|\nabla F(\mathbf{x})\|^2$$

$$\overset{\text{Asm. } 3a}{\leq} 2S^2 K^2 \eta^2 \sigma^2 + 2L^2 S^2 K^2 \eta^2 E_r + 2S^2 K^3 \eta^2 \zeta^2 + 2\beta^2 S^2 K^3 \eta^2 \|\nabla F(\mathbf{x})\|^2 + \frac{4}{3} S^3 K^3 \eta^2 \|\nabla F(\mathbf{x})\|^2$$

After rearranging the preceding inequality, we get

$$(1 - 2L^2 S^2 K^2 \eta^2) E_r \leq 2S^2 K^2 \eta^2 \sigma^2 + 2S^2 K^3 \eta^2 \zeta^2 + 2\beta^2 S^2 K^3 \eta^2 \|\nabla F(\mathbf{x})\|^2 + \frac{4}{3} S^3 K^3 \eta^2 \|\nabla F(\mathbf{x})\|^2$$

Finally, using the condition that $\eta \leq \frac{1}{6LSK}$, which implies $1 - 2L^2 S^2 K^2 \eta^2 \geq \frac{8}{9}$, we have

$$E_r \leq \frac{9}{4} S^2 K^2 \eta^2 \sigma^2 + \frac{9}{4} S^2 K^3 \eta^2 \zeta^2 + \frac{9}{4} \beta^2 S^2 K^3 \eta^2 \|\nabla F(\mathbf{x})\|^2 + \frac{3}{2} S^3 K^3 \eta^2 \|\nabla F(\mathbf{x})\|^2 .$$

The claim follows after recovering the superscripts and taking unconditional expectations. $\qquad\square$

### D.3.2 Proof of non-convex case of Theorem 1 and Corollary 1

*Proof of non-convex case of Theorem 1.* Substituting (52) into (48) and using $\eta \leq \frac{1}{6LSK} \min\left\{1, \frac{\sqrt{S}}{\beta}\right\}$, we can simplify the recursion as follows:

$$\mathbb{E}\left[ F(\mathbf{x}^{(r+1)}) - F(\mathbf{x}^{(r)}) \right] \leq -\frac{1}{3} SK\eta \mathbb{E}\left[ \left\| \nabla F(\mathbf{x}^{(r)}) \right\|^2 \right] + LSK\eta^2 \sigma^2 + \frac{9}{8} L^2 S^2 K^2 \eta^3 \sigma^2 + \frac{9}{8} L^2 S^2 K^3 \eta^3 \zeta^2$$

Letting $\tilde{\eta} := SK\eta$, subtracting $F^*$ from both sides and then rearranging the terms, we have

$$\mathbb{E}\left[ F(\mathbf{x}^{(r+1)}) - F^* \right] \leq \mathbb{E}\left[ F(\mathbf{x}^{(r)}) - F^* \right] - \frac{\tilde{\eta}}{3} \mathbb{E}\left[ \left\| \nabla F(\mathbf{x}^{(r)}) \right\|^2 \right] + \frac{L\tilde{\eta}^2 \sigma^2}{SK} + \frac{9L^2 \tilde{\eta}^3 \sigma^2}{8SK} + \frac{9L^2 \tilde{\eta}^3 \zeta^2}{8S}$$

Then applying Lemma 8 with $t = r$ ($T = R$), $\gamma = \tilde{\eta}$, $r_t = \mathbb{E}\left[ F(\mathbf{x}^{(r)}) - F^* \right]$, $b = \frac{1}{3}$, $s_t = \mathbb{E}\left[ \left\| \nabla F(\mathbf{x}^{(r)}) \right\|^2 \right]$, $w_t = 1$, $c_1 = \frac{L\sigma^2}{SK}$, $c_2 = \frac{9L^2 \sigma^2}{8SK} + \frac{9L^2 \zeta^2}{8S}$ and $\frac{1}{d} = \frac{1}{6L} \min\left\{1, \frac{\sqrt{S}}{\beta}\right\}$ ($\tilde{\eta} = SK\eta \leq \frac{1}{6L} \min\left\{1, \frac{\sqrt{S}}{\beta}\right\}$), we have

$$\min_{0 \leq r \leq R} \mathbb{E}\left[ \left\| \nabla F(\mathbf{x}^{(r)}) \right\|^2 \right] \leq \frac{3 \left( F(\mathbf{x}^0) - F^* \right)}{\tilde{\eta} R} + \frac{3L\tilde{\eta}\sigma^2}{SK} + \frac{27L^2 \tilde{\eta}^2 \sigma^2}{8SK} + \frac{27L^2 \tilde{\eta}^2 \zeta^2}{8S} \qquad (54)$$

where we use $\min_{0 \leq r \leq R} \mathbb{E}\left[ \left\| \nabla F(\mathbf{x}^{(r)}) \right\|^2 \right] \leq \frac{1}{R+1} \sum_{r=0}^{R} \mathbb{E}\left[ \left\| \nabla F(\mathbf{x}^{(r)}) \right\|^2 \right]$. Then, using $\tilde{\eta} \leq \frac{1}{6L(\beta+1)} \leq \min\left\{1, \frac{\sqrt{S}}{\beta}\right\}$ and tuning the learning rate carefully, we get

$$\min_{0 \leq r \leq R} \mathbb{E}\left[ \left\| \nabla F(\mathbf{x}^{(r)}) \right\|^2 \right] = \mathcal{O}\left( \frac{(L\sigma^2 A)^{1/2}}{\sqrt{SKR}} + \frac{(L^2 \sigma^2 A^2)^{1/3}}{(SK)^{1/3} R^{2/3}} + \frac{(L^2 \zeta^2 A^2)^{1/3}}{S^{1/3} R^{2/3}} + \frac{L\beta A}{R} \right) \qquad (55)$$

where $A := F(\mathbf{x}^0) - F^*$. Eq. (54) and Eq. (55) are the upper bounds with partial client participation. In particular, when $S = M$, we get the claim of the non-convex case of Theorem 1 and Corollary 1. $\qquad\square$

# E  Proofs of Theorem 2

Here we slightly improve the convergence guarantee for the strongly convex case by combining the works of Karimireddy et al. (2020); Koloskova et al. (2020). Moreover, we reproduce the guarantees for the general convex and non-convex cases based on Karimireddy et al. (2020) for completeness. The results are given in Theorem 2. We provide the proof of Theorem 2 for the strongly convex, general convex and non-convex cases in Sections E.1, E.2 and E.3, respectively.

In the following proof, we consider the partial client participation setting, specifically, selecting partial clients without replacement. So we assume that $\pi = \{\pi_1, \pi_2, \ldots, \pi_M\}$ is a permutation of $\{1, 2, \ldots, M\}$ in a certain training round and only the first $S$ selected clients $\{\pi_1, \pi_2, \ldots, \pi_S\}$ will participate in this round. Without otherwise stated, we use $\mathbb{E}[\cdot]$ to represent the expectation with respect to both types of randomness (i.e., sampling data samples $\xi$ and sampling clients $\pi$).

**Theorem 2.** *For PFL (Algorithm 2), there exist a constant effective learning rate $\tilde{\eta} := K\eta$ and weights $w_r$, such that the weighted average of the global parameters $\bar{\mathbf{x}}^{(R)} := \frac{1}{W_R} \sum_{r=0}^{R} w_r \mathbf{x}^{(r)}$ ($W_R = \sum_{r=0}^{R} w_r$) satisfies the following upper bounds:*

*Strongly convex: Under Assumptions 1, 2, 3b, there exist a constant effective learning rate $\frac{1}{\mu R} \leq \tilde{\eta} \leq \frac{1}{6L}$ and weights $w_r = (1 - \frac{\mu\tilde{\eta}}{2})^{-(r+1)}$, such that it holds that*

$$\mathbb{E}\left[F(\bar{\mathbf{x}}^{(R)}) - F(\mathbf{x}^*)\right] \leq \frac{9}{2}\mu D^2 \exp\left(-\frac{\mu\tilde{\eta}R}{2}\right) + \frac{12\tilde{\eta}\sigma^2}{MK} + \frac{18L\tilde{\eta}^2\sigma^2}{K} + 12L\tilde{\eta}^2\zeta_*^2 \tag{56}$$

*General convex: Under Assumptions 1, 2, 3b, there exist a constant effective learning rate $\tilde{\eta} \leq \frac{1}{6L}$ and weights $w_r = 1$, such that it holds that*

$$\mathbb{E}\left[F(\bar{\mathbf{x}}^{(R)}) - F(\mathbf{x}^*)\right] \leq \frac{3D^2}{\tilde{\eta}R} + \frac{12\tilde{\eta}\sigma^2}{MK} + \frac{18L\tilde{\eta}^2\sigma^2}{K} + 12L\tilde{\eta}^2\zeta_*^2 \tag{57}$$

*Non-convex: Under Assumptions 1, 2, 3a, there exist a constant effective learning rate $\tilde{\eta} \leq \frac{1}{6L(\beta+1)}$ and weights $w_r = 1$, such that it holds that*

$$\min_{0 \leq r \leq R} \mathbb{E}\left[\|\nabla F(\mathbf{x}^{(r)})\|^2\right] \leq \frac{3A}{\tilde{\eta}R} + \frac{3L\tilde{\eta}\sigma^2}{SK} + \frac{27L^2\tilde{\eta}^2\sigma^2}{8K} + \frac{9}{4}L^2\tilde{\eta}^2\zeta^2 \tag{58}$$

*where $D := \|x^{(0)} - x^*\|$ for the convex cases and $A := F(\mathbf{x}^{(0)}) - F^*$ for the non-convex case.*

**Corollary 2.** *Applying the results of Theorem 2. By choosing a appropriate learning rate (see the proof of Theorem 2 in Appendix E), we can obtain the convergence rates for PFL as follows:*

*Strongly convex: Under Assumptions 1, 2, 3b, there exist a constant effective learning rate $\frac{1}{\mu R} \leq \tilde{\eta} \leq \frac{1}{6L}$ and weights $w_r = (1 - \frac{\mu\tilde{\eta}}{2})^{-(r+1)}$, such that it holds that*

$$\mathbb{E}\left[F(\bar{\mathbf{x}}^{(R)}) - F(\mathbf{x}^*)\right] = \tilde{\mathcal{O}}\left(\frac{\sigma^2}{\mu MKR} + \frac{L\sigma^2}{\mu^2 KR^2} + \frac{L\zeta_*^2}{\mu^2 R^2} + \mu D^2 \exp\left(-\frac{\mu R}{12L}\right)\right) \tag{59}$$

*General convex: Under Assumptions 1, 2, 3b, there exist a constant effective learning rate $\tilde{\eta} \leq \frac{1}{6L}$ and weights $w_r = 1$, such that it holds that*

$$\mathbb{E}\left[F(\bar{\mathbf{x}}^{(R)}) - F(\mathbf{x}^*)\right] = \mathcal{O}\left(\frac{\sigma D}{\sqrt{MKR}} + \frac{\left(L\sigma^2 D^4\right)^{1/3}}{K^{1/3}R^{2/3}} + \frac{\left(L\zeta_*^2 D^4\right)^{1/3}}{R^{2/3}} + \frac{LD^2}{R}\right) \tag{60}$$

*Non-convex: Under Assumptions 1, 2, 3a, there exist a constant effective learning rate $\tilde{\eta} \leq \frac{1}{6L(\beta+1)}$ and weights $w_r = 1$, such that it holds that*

$$\min_{0 \leq r \leq R} \mathbb{E}\left[\|\nabla F(\mathbf{x}^{(r)})\|^2\right] = \mathcal{O}\left(\frac{\left(L\sigma^2 A\right)^{1/2}}{\sqrt{MKR}} + \frac{\left(L^2\sigma^2 A^2\right)^{1/3}}{K^{1/3}R^{2/3}} + \frac{\left(L^2\zeta^2 A^2\right)^{1/3}}{R^{2/3}} + \frac{L\beta A}{R}\right) \tag{61}$$

*where $\mathcal{O}$ omits absolute constants, $\tilde{\mathcal{O}}$ omits absolute constants and polylogarithmic factors, $D := \|x^{(0)} - x^*\|$ for the convex cases and $A := F(\mathbf{x}^{(0)}) - F^*$ for the non-convex case.*

### E.1 Strongly convex case

#### E.1.1 Find the per-round recursion

**Lemma 11.** *Let Assumptions 1, 2, 3b hold and assume that all the local objectives are $\mu$-strongly convex. If the learning rate satisfies $\eta \leq \frac{1}{6LK}$, then it holds that*

$$
\mathbb{E}\left[\left\|\mathbf{x}^{(r+1)} - \mathbf{x}^*\right\|^2\right] \leq \left(1 - \frac{\mu K \eta}{2}\right) \mathbb{E}\left[\left\|\mathbf{x}^{(r)} - \mathbf{x}^*\right\|^2\right] + \frac{4K\eta^2\sigma^2}{S} + 4K^2\eta^2 \frac{M-S}{S(M-1)}\zeta_*^2
$$
$$
- \frac{2}{3}K\eta\mathbb{E}\left[D_F(\mathbf{x}^{(r)}, \mathbf{x}^*)\right] + \frac{8}{3}L\eta\frac{1}{S}\sum_{m=1}^{S}\sum_{k=0}^{K-1}\mathbb{E}\left[\left\|\mathbf{x}_{m,k}^{(r)} - \mathbf{x}^{(r)}\right\|^2\right] \quad (62)
$$

*Proof.* According to Algorithm 2, the overall model updates of PFL after one complete training round (with $S$ clients selected for training) is

$$
\Delta\mathbf{x} = \mathbf{x}^{(r+1)} - \mathbf{x}^{(r)} = -\frac{\eta}{S}\sum_{m=1}^{S}\sum_{k=0}^{K-1}\mathbf{g}_{\pi_m,k}^{(r)} \,,
$$

where $\mathbf{g}_{\pi_m,k}^{(r)} = \nabla f_{\pi_m}(\mathbf{x}_{m,k}^{(r)}; \xi)$ is the stochastic gradient of $F_{\pi_m}$ regarding the vector $\mathbf{x}_{m,k}^{(r)}$. Thus,

$$
\mathbb{E}\left[\Delta\mathbf{x}\right] = -\frac{\eta}{S}\sum_{m=1}^{S}\sum_{k=0}^{K-1}\mathbb{E}\left[\nabla F_{\pi_m}(\mathbf{x}_{m,k})\right]
$$

In the following, we focus on a single training round, and hence we drop the superscripts $r$ for a while, e.g., writing $\mathbf{x}_{m,k}$ to replace $\mathbf{x}_{m,k}^{(r)}$. Specially, we would like to use $\mathbf{x}$ to replace $\mathbf{x}_{1,0}^{(r)}$. Without otherwise stated, the expectation is conditioned on $\mathbf{x}^{(r)}$.

We start by substituting the overall updates:

$$
\mathbb{E}\left[\|\mathbf{x} + \Delta\mathbf{x} - \mathbf{x}^*\|^2\right]
$$
$$
= \|\mathbf{x} - \mathbf{x}^*\|^2 + 2\mathbb{E}\left[\langle \mathbf{x} - \mathbf{x}^*, \Delta\mathbf{x}\rangle\right] + \mathbb{E}\left[\|\Delta\mathbf{x}\|^2\right]
$$
$$
= \|\mathbf{x} - \mathbf{x}^*\|^2 - \frac{2\eta}{S}\sum_{m=1}^{S}\sum_{k=0}^{K-1}\mathbb{E}\left[\langle \nabla F_{\pi_m}(\mathbf{x}_{m,k}), \mathbf{x} - \mathbf{x}^*\rangle\right] + \frac{\eta^2}{S^2}\mathbb{E}\left[\left\|\sum_{m=1}^{S}\sum_{k=0}^{K-1}\mathbf{g}_{\pi_m,k}\right\|^2\right] \quad (63)
$$

We can apply Lemma 2 with $\boldsymbol{x} = \mathbf{x}_{m,k}$, $\boldsymbol{y} = \mathbf{x}^*$, $\boldsymbol{z} = \mathbf{x}$ and $h = F_{\pi_m}$ for the second term on the right hand side in (63):

$$
- \frac{2\eta}{S}\sum_{m=1}^{S}\sum_{k=0}^{K-1}\mathbb{E}\left[\langle \nabla F_{\pi_m}(\mathbf{x}_{m,k}), \mathbf{x} - \mathbf{x}^*\rangle\right]
$$
$$
\leq -\frac{2\eta}{S}\sum_{m=1}^{S}\sum_{k=0}^{K-1}\mathbb{E}\left[F_{\pi_m}(\mathbf{x}) - F_{\pi_m}(\mathbf{x}^*) + \frac{\mu}{4}\|\mathbf{x} - \mathbf{x}^*\|^2 - L\|\mathbf{x}_{m,k} - \mathbf{x}\|^2\right]
$$
$$
\leq -2K\eta D_F(\mathbf{x}, \mathbf{x}^*) - \frac{1}{2}\mu K\eta\|\mathbf{x} - \mathbf{x}^*\|^2 + \frac{2L\eta}{S}\sum_{m=1}^{S}\sum_{k=0}^{K-1}\mathbb{E}\left[\|\mathbf{x}_{m,k} - \mathbf{x}\|^2\right] \quad (64)
$$

For the third term on the right hand side in (63), using Jensen's inequality, we have

$$\mathbb{E}\left[\left\|\sum_{m=1}^{S}\sum_{k=0}^{K-1}\mathbf{g}_{\pi_m,k}\right\|^2\right]$$

$$\leq 4\mathbb{E}\left[\left\|\sum_{m=1}^{S}\sum_{k=0}^{K-1}(\mathbf{g}_{\pi_m,k}-\nabla F_{\pi_m}(\mathbf{x}_{m,k}))\right\|^2\right] + 4\mathbb{E}\left[\left\|\sum_{m=1}^{S}\sum_{k=0}^{K-1}(\nabla F_{\pi_m}(\mathbf{x}_{m,k})-\nabla F_{\pi_m}(\mathbf{x}))\right\|^2\right]$$

$$+ 4\mathbb{E}\left[\left\|\sum_{m=1}^{S}\sum_{k=0}^{K-1}(\nabla F_{\pi_m}(\mathbf{x})-\nabla F_{\pi_m}(\mathbf{x}^*))\right\|^2\right] + 4\mathbb{E}\left[\left\|\sum_{m=1}^{S}\sum_{k=0}^{K-1}\nabla F_{\pi_m}(\mathbf{x}^*)\right\|^2\right] \tag{65}$$

For the first term on the right hand side in (65), we have

$$4\mathbb{E}\left[\left\|\sum_{m=1}^{S}\sum_{k=0}^{K-1}(\mathbf{g}_{\pi_m,k}-\nabla F_{\pi_m}(\mathbf{x}_{m,k}))\right\|^2\right] = 4\sum_{m=1}^{S}\mathbb{E}\left[\left\|\sum_{k=0}^{K-1}(\mathbf{g}_{\pi_m,k}-\nabla F_{\pi_m}(\mathbf{x}_{m,k}))\right\|^2\right]$$

$$\overset{\text{Lem. }1}{=} 4\sum_{m=1}^{S}\sum_{k=0}^{K-1}\mathbb{E}\left[\left\|\mathbf{g}_{\pi_m,k}-\nabla F_{\pi_m}(\mathbf{x}_{m,k})\right\|^2\right]$$

$$\overset{\text{Asm. }2}{\leq} 4SK\sigma^2, \tag{66}$$

where we use the fact that clients are independent to each other in the first equality and apply Lemma 1 by seeing the data sample $\xi_{m,k}$, the stochastic gradient $\mathbf{g}_{\pi_m,k}$, the gradient $\nabla F_{\pi_m}(\xi_{m,k})$ as $\xi_i$, $\mathbf{x}_i$, $\mathbf{e}_i$ in the second equality. For the second term on the right hand side in (65), we have

$$4\mathbb{E}\left[\left\|\sum_{m=1}^{S}\sum_{k=0}^{K-1}(\nabla F_{\pi_m}(\mathbf{x}_{m,k})-\nabla F_{\pi_m}(\mathbf{x}))\right\|^2\right] \overset{(15)}{\leq} 4SK\sum_{m=1}^{S}\sum_{k=0}^{K-1}\mathbb{E}\left[\left\|\nabla F_{\pi_m}(\mathbf{x}_{m,k})-\nabla F_{\pi_m}(\mathbf{x})\right\|^2\right]$$

$$\overset{\text{Asm. }1}{\leq} 4L^2SK\sum_{m=1}^{S}\sum_{k=0}^{K-1}\mathbb{E}\left[\left\|\mathbf{x}_{m,k}-\mathbf{x}\right\|^2\right]$$

For the third term on the right hand side in (65), we have

$$4\mathbb{E}\left[\left\|\sum_{m=1}^{S}\sum_{k=0}^{K-1}(\nabla F_{\pi_m}(\mathbf{x})-\nabla F_{\pi_m}(\mathbf{x}^*))\right\|^2\right] \overset{(15)}{\leq} 4SK\sum_{m=1}^{S}\sum_{k=0}^{K-1}\mathbb{E}\left[\left\|\nabla F_{\pi_m}(\mathbf{x})-\nabla F_{\pi_m}(\mathbf{x}^*)\right\|^2\right]$$

$$\overset{(18)}{\leq} 8LSK\sum_{m=1}^{S}\sum_{k=0}^{K-1}\mathbb{E}\left[D_{F_{\pi_m}}(\mathbf{x},\mathbf{x}^*)\right]$$

$$= 8LS^2K^2 D_F(\mathbf{x},\mathbf{x}^*),$$

where the last inequality is because $\mathbb{E}\left[D_{F_{\pi_m}}(\mathbf{x},\mathbf{x}^*)\right] = D_F(\mathbf{x},\mathbf{x}^*)$. The forth term on the right hand side in (65) can be bounded by Lemma 3 as follows:

$$4\mathbb{E}\left[\left\|\sum_{m=1}^{S}\sum_{k=0}^{K-1}\nabla F_{\pi_m}(\mathbf{x}^*)\right\|^2\right] \overset{(23)}{\leq} 4S^2K^2\frac{M-S}{S(M-1)}\zeta_*^2.$$

With the preceding four inequalities, we can bound the third term on the right hand side in (63):

$$\frac{\eta^2}{S^2}\mathbb{E}\left[\left\|\sum_{m=1}^{S}\sum_{k=0}^{K-1}\mathbf{g}_{\pi_m,k}\right\|^2\right] \leq \frac{4K\eta^2\sigma^2}{S} + 4L^2K\eta^2\frac{1}{S}\sum_{m=1}^{S}\sum_{k=0}^{K-1}\mathbb{E}\left[\left\|\mathbf{x}_{m,k}-\mathbf{x}\right\|^2\right]$$

$$+ 8LK^2\eta^2 D_F(\mathbf{x},\mathbf{x}^*) + 4K^2\eta^2\frac{M-S}{S(M-1)}\zeta_*^2 \tag{67}$$

Then substituting (64) and (67) into (63), we have

$$
\mathbb{E}\left[\|\mathbf{x} + \Delta\mathbf{x} - \mathbf{x}^*\|^2\right] \leq \left(1 - \tfrac{\mu K \eta}{2}\right)\|\mathbf{x} - \mathbf{x}^*\|^2 + \frac{4K\eta^2\sigma^2}{S} + 4K^2\eta^2\frac{M-S}{S(M-1)}\zeta_*^2
$$

$$
- 2K\eta(1 - 4LK\eta)D_F(\mathbf{x},\mathbf{x}^*) + 2L\eta(1 + 2LK\eta)\frac{1}{S}\sum_{m=1}^{S}\sum_{k=0}^{K-1}\mathbb{E}\left[\|\mathbf{x}_{m,k} - \mathbf{x}\|^2\right]
$$

$$
\leq \left(1 - \tfrac{\mu K \eta}{2}\right)\|\mathbf{x} - \mathbf{x}^*\|^2 + \frac{4K\eta^2\sigma^2}{S} + 4K^2\eta^2\frac{M-S}{S(M-1)}\zeta_*^2
$$

$$
- \frac{2}{3}K\eta D_F(\mathbf{x},\mathbf{x}^*) + \frac{8}{3}L\eta\frac{1}{S}\sum_{m=1}^{S}\sum_{k=0}^{K-1}\mathbb{E}\left[\|\mathbf{x}_{m,k} - \mathbf{x}\|^2\right],
$$

where we use the condition that $\eta \leq \frac{1}{6LK}$ in the last inequality. The claim of this lemma follows after recovering the superscripts and taking unconditional expectation. $\square$

### E.1.2 Bounding the client drift with Assumption 3b

The "client drift" (Karimireddy et al., 2020) in PFL is defined as follows: :

$$
\mathcal{E}_r := \frac{1}{S}\sum_{m=1}^{S}\sum_{k=0}^{K-1}\mathbb{E}\left[\left\|\mathbf{x}_{m,k}^{(r)} - \mathbf{x}^{(r)}\right\|^2\right] \tag{68}
$$

**Lemma 12.** *Let Assumptions 1, 2, 3b hold and assume that all the local objectives are $\mu$-strongly convex. If the learning rate satisfies $\eta \leq \frac{1}{6LK}$, then the client drift is bounded:*

$$
\mathcal{E}_r \leq \frac{9}{4}K^2\eta^2\sigma^2 + \frac{3}{2}K^3\eta^2\zeta_*^2 + 3LK^3\eta^2\mathbb{E}\left[D_F(\mathbf{x}^{(r)},\mathbf{x}^*)\right] \tag{69}
$$

*Proof.* According to Algorithm 2, the model updates of PFL from $\mathbf{x}^{(r)}$ to $\mathbf{x}_{m,k}^{(r)}$ is

$$
\mathbf{x}_{m,k}^{(r)} - \mathbf{x}^{(r)} = -\eta\sum_{j=0}^{k-1}\mathbf{g}_{\pi_m,j}^{(r)}
$$

In the following, we focus on a single training round, and hence we drop the superscripts $r$ for a while, e.g., writing $\mathbf{x}_{m,k}$ to replace $\mathbf{x}_{m,k}^{(r)}$. Specially, we would like to use $\mathbf{x}$ to replace $\mathbf{x}_{1,0}^{(r)}$. Without otherwise stated, the expectation is conditioned on $\mathbf{x}^{(r)}$.

We use Jensen's inequality to bound the term $\mathbb{E}\left[\|\mathbf{x}_{m,k} - \mathbf{x}\|^2\right] = \eta^2\mathbb{E}\left[\left\|\sum_{j=0}^{k-1}\mathbf{g}_{\pi_m,j}\right\|^2\right]$:

$$
\mathbb{E}\left[\|\mathbf{x}_{m,k} - \mathbf{x}\|^2\right] \leq 4\eta^2\mathbb{E}\left[\left\|\sum_{j=0}^{k-1}(\mathbf{g}_{\pi_m,j} - \nabla F_{\pi_m}(\mathbf{x}_{m,j}))\right\|^2\right] + 4\eta^2\mathbb{E}\left[\left\|\sum_{j=0}^{k-1}(\nabla F_{\pi_m}(\mathbf{x}_{m,j}) - \nabla F_{\pi_m}(\mathbf{x}))\right\|^2\right]
$$

$$
+ 4\eta^2\mathbb{E}\left[\left\|\sum_{j=0}^{k-1}(\nabla F_{\pi_m}(\mathbf{x}) - \nabla F_{\pi_m}(\mathbf{x}^*))\right\|^2\right] + 4\eta^2\mathbb{E}\left[\left\|\sum_{j=0}^{k-1}\nabla F_{\pi_m}(\mathbf{x}^*)\right\|^2\right]
$$

Applying Lemma 1 to the first term and Jensen's inequality to the last three terms on the right hand side in the preceding inequality, respectively, we get

$$
\mathbb{E}\left[\|\mathbf{x}_{m,k} - \mathbf{x}\|^2\right] \leq 4\sum_{j=0}^{k-1}\eta^2\mathbb{E}\left[\|\mathbf{g}_{\pi_m,j} - \nabla F_{\pi_m}(\mathbf{x}_{m,j})\|^2\right] + 4k\sum_{j=0}^{k-1}\eta^2\mathbb{E}\left[\|\nabla F_{\pi_m}(\mathbf{x}_{m,j}) - \nabla F_{\pi_m}(\mathbf{x})\|^2\right]
$$

$$
+ 4k^2\eta^2\mathbb{E}\left[\|\nabla F_{\pi_m}(\mathbf{x}) - \nabla F_{\pi_m}(\mathbf{x}^*)\|^2\right] + 4k^2\eta^2\mathbb{E}\left[\|\nabla F_{\pi_m}(\mathbf{x}^*)\|^2\right]
$$

The first term can be bounded by $4k\eta^2\sigma^2$ with Assumption 2. The second term can be bounded by $4L^2k\eta^2\sum_{j=0}^{k-1}\mathbb{E}\left[\|\mathbf{x}_{m,j} - \mathbf{x}\|^2\right]$ with Assumption 1. The third term can be bounded by $8Lk^2\eta^2\mathbb{E}\left[D_{F_{\pi_m}}(\mathbf{x}, \mathbf{x}^*)\right]$ with Eq. (18). Thus, we have

$$\mathbb{E}\left[\|\mathbf{x}_{m,k} - \mathbf{x}\|^2\right] \leq 4k\eta^2\sigma^2 + 4L^2k\eta^2\sum_{j=0}^{k-1}\mathbb{E}\left[\|\mathbf{x}_{m,j} - \mathbf{x}\|^2\right] + 8Lk^2\eta^2\mathbb{E}\left[D_{F_{\pi_m}}(\mathbf{x}, \mathbf{x}^*)\right] + 4k^2\eta^2\mathbb{E}\left[\|\nabla F_{\pi_m}(\mathbf{x}^*)\|^2\right]$$

$$\leq 4k\eta^2\sigma^2 + 4L^2k\eta^2\sum_{j=0}^{k-1}\mathbb{E}\left[\|\mathbf{x}_{m,j} - \mathbf{x}\|^2\right] + 8Lk^2\eta^2 D_F(\mathbf{x}, \mathbf{x}^*) + 4k^2\eta^2\zeta_*^2$$

Then returning to $\mathcal{E}_r := \frac{1}{S}\sum_{m=1}^{S}\sum_{k=0}^{K-1}\mathbb{E}\left[\|\mathbf{x}_{m,k} - \mathbf{x}\|^2\right]$, we have

$$\mathcal{E}_r = 4\eta^2\sigma^2\sum_{k=0}^{K-1}k + 4L^2\eta^2\frac{1}{S}\sum_{m=1}^{S}\sum_{k=0}^{K-1}k\sum_{j=0}^{k-1}\mathbb{E}\left[\|\mathbf{x}_{m,j} - \mathbf{x}\|^2\right] + 8L\eta^2\sum_{k=0}^{K-1}k^2 D_F(\mathbf{x}, \mathbf{x}^*) + 4\eta^2\sum_{k=0}^{K-1}k^2\zeta_*^2$$

Using the facts that $\sum_{k=1}^{K-1}k = \frac{(K-1)K}{2} \leq \frac{K^2}{2}$ and $\sum_{k=1}^{K-1}k^2 = \frac{(K-1)K(2K-1)}{6} \leq \frac{K^3}{3}$, we can simplify the preceding inequality:

$$\mathcal{E}_r \leq 2K^2\eta^2\sigma^2 + 2L^2K^2\eta^2\frac{1}{S}\sum_{m=1}^{S}\sum_{j=0}^{k-1}\mathbb{E}\left[\|\mathbf{x}_{m,j} - \mathbf{x}\|^2\right] + \frac{8}{3}LK^3\eta^2 D_F(\mathbf{x}, \mathbf{x}^*) + \frac{4}{3}K^3\eta^2\zeta_*^2$$

After rearranging the preceding inequality, we get

$$(1 - 2L^2K^2\eta^2)\mathcal{E}_r \leq 2K^2\eta^2\sigma^2 + \frac{4}{3}K^3\eta^2\zeta_*^2 + \frac{8}{3}LK^3\eta^2 D_F(\mathbf{x}, \mathbf{x}^*)$$

Finally, using the condition that $\eta \leq \frac{1}{6LK}$, which implies $1 - 2L^2K^2\eta^2 \geq \frac{8}{9}$, we have

$$\mathcal{E}_r \leq \frac{9}{4}K^2\eta^2\sigma^2 + \frac{3}{2}K^3\eta^2\zeta_*^2 + 3LK^3\eta^2 D_F(\mathbf{x}, \mathbf{x}^*).$$

The claim follows after recovering the superscripts and taking unconditional expectations. $\qquad\square$

### E.1.3  Proof of strongly convex case of Theorem 2

*Proof of strongly convex case of Theorem 2.* Substituting (69) into (62) and using $\eta \leq \frac{1}{6LK}$, we can simplify the recursion as,

$$\mathbb{E}\left[\left\|\mathbf{x}^{(r+1)} - \mathbf{x}^*\right\|^2\right] \leq \left(1 - \frac{\mu K\eta}{2}\right)\mathbb{E}\left[\left\|\mathbf{x}^{(r)} - \mathbf{x}^*\right\|^2\right] - \frac{1}{3}K\eta\mathbb{E}\left[D_F(\mathbf{x}^{(r)}, \mathbf{x}^*)\right]$$

$$+ \frac{4K\eta^2\sigma^2}{S} + 4K^2\eta^2\frac{M-S}{S(M-1)}\zeta_*^2 + 6LK^2\eta^3\sigma^2 + 4LK^3\eta^3\zeta_*^2$$

Let $\tilde{\eta} = K\eta$, we have

$$\mathbb{E}\left[\left\|\mathbf{x}^{(r+1)} - \mathbf{x}^*\right\|^2\right] \leq \left(1 - \frac{\mu\tilde{\eta}}{2}\right)\mathbb{E}\left[\left\|\mathbf{x}^{(r)} - \mathbf{x}^*\right\|^2\right] - \frac{\tilde{\eta}}{3}\mathbb{E}\left[D_F(\mathbf{x}^{(r)}, \mathbf{x}^*)\right]$$

$$+ \frac{4\tilde{\eta}^2\sigma^2}{SK} + \frac{4\tilde{\eta}^2(M-S)\zeta_*^2}{S(M-1)} + \frac{6L\tilde{\eta}^3\sigma^2}{K} + 4L\tilde{\eta}^3\zeta_*^2 \qquad (70)$$

Applying Lemma 7 with $t = r$ ($T = R$), $\gamma = \tilde{\eta}$, $r_t = \mathbb{E}\left[\|\mathbf{x}^{(r)} - \mathbf{x}^*\|^2\right]$, $a = \frac{\mu}{2}$, $b = \frac{1}{3}$, $s_t = \mathbb{E}\left[D_F(\mathbf{x}^{(r)}, \mathbf{x}^*)\right]$, $w_t = (1 - \frac{\mu\tilde{\eta}}{2})^{-(r+1)}$, $c_1 = \frac{4\sigma^2}{SK} + \frac{4(M-S)\zeta_*^2}{S(M-1)}$, $c_2 = \frac{6L\sigma^2}{K} + 4L\zeta_*^2$ and $\frac{1}{d} = \frac{1}{6L}$ ($\tilde{\eta} = K\eta \leq \frac{1}{6L}$), it follows that

$$\mathbb{E}\left[F(\bar{\mathbf{x}}^{(R)}) - F(\mathbf{x}^*)\right] \leq \frac{1}{W_R}\sum_{r=0}^{R}w_r\mathbb{E}\left[F(\mathbf{x}^{(r)}) - F(\mathbf{x}^*)\right]$$

$$\leq \frac{9}{2}\mu\left\|\mathbf{x}^{(0)} - \mathbf{x}^*\right\|^2\exp\left(-\frac{1}{2}\mu\tilde{\eta}R\right) + \frac{12\tilde{\eta}\sigma^2}{SK} + \frac{12\tilde{\eta}(M-S)\zeta_*^2}{S(M-1)} + \frac{18L\tilde{\eta}^2\sigma^2}{K} + 12L\tilde{\eta}^2\zeta_*^2 \quad (71)$$

where $\bar{\mathbf{x}}^{(R)} = \frac{1}{W_R} \sum_{r=0}^{R} w_r \mathbf{x}^{(r)}$ and we use Jensen's inequality ($F$ is convex) in the first inequality. Thus, by tuning the learning rate carefully, we get

$$\mathbb{E}\left[F(\bar{\mathbf{x}}^{(R)}) - F(\mathbf{x}^*)\right] = \tilde{\mathcal{O}}\left(\mu D^2 \exp\left(-\frac{\mu R}{12L}\right) + \frac{\sigma^2}{\mu SKR} + \frac{(M-S)\zeta_*^2}{\mu SR(M-1)} + \frac{L\sigma^2}{\mu^2 KR^2} + \frac{L\zeta_*^2}{\mu^2 R^2}\right) \tag{72}$$

where $D := \|\mathbf{x}^{(0)} - \mathbf{x}^*\|$. Eq. (71) and Eq. (72) are the upper bounds with partial client participation. When $M$ is large enough, we have $\frac{(M-S)}{S(M-1)} \approx (1 - \frac{S}{M})\frac{1}{S}$. This is the constant appearing in Karimireddy et al. (2020); Woodworth et al. (2020b). In particular, when $S = M$, we can get the claim of the strongly convex case of Theorem 2 and Corollary 2. $\qquad\square$

### E.2 General convex case

#### E.2.1 Proof of general convex case of Theorem 2 and Corollary 2

*Proof of general convex case of Theorem 2.* Let $\mu = 0$ in (70), we get the simplified per-round recursion of general convex case,

$$\mathbb{E}\left[\left\|\mathbf{x}^{(r+1)} - \mathbf{x}^*\right\|^2\right] \leq \mathbb{E}\left[\left\|\mathbf{x}^{(r)} - \mathbf{x}^*\right\|^2\right] - \frac{\tilde{\eta}}{3}\mathbb{E}\left[D_F(\mathbf{x}^{(r)}, \mathbf{x}^*)\right]$$
$$+ \frac{4\tilde{\eta}^2\sigma^2}{SK} + \frac{4\tilde{\eta}^2(M-S)\zeta_*^2}{S(M-1)} + \frac{6L\tilde{\eta}^3\sigma^2}{K} + 4L\tilde{\eta}^3\zeta_*^2$$

Applying Lemma 8 with $t = r$ ($T = R$), $\gamma = \tilde{\eta}$, $r_t = \mathbb{E}\left[\left\|\mathbf{x}^{(r)} - \mathbf{x}^*\right\|^2\right]$, $b = \frac{1}{3}$, $s_t = \mathbb{E}\left[D_F(\mathbf{x}^{(r)}, \mathbf{x}^*)\right]$, $w_t = 1$, $c_1 = \frac{4\sigma^2}{SK} + \frac{4(M-S)\zeta_*^2}{S(M-1)}$, $c_2 = \frac{6L\sigma^2}{K} + 4L\zeta_*^2$ and $\frac{1}{d} = \frac{1}{6L}$ ($\tilde{\eta} = K\eta \leq \frac{1}{6L}$), it follows that

$$\mathbb{E}\left[F(\bar{\mathbf{x}}^{(R)}) - F(\mathbf{x}^*)\right] \leq \frac{1}{W_R}\sum_{r=0}^{R} w_r\left(F(\mathbf{x}^{(r)}) - F(\mathbf{x}^*)\right)$$

$$\leq \frac{3\left\|\mathbf{x}^{(0)} - \mathbf{x}^*\right\|^2}{\tilde{\eta}R} + \frac{12\tilde{\eta}\sigma^2}{SK} + \frac{12\tilde{\eta}(M-S)\zeta_*^2}{S(M-1)} + \frac{18L\tilde{\eta}^2\sigma^2}{K} + 12L\tilde{\eta}^2\zeta_*^2 \tag{73}$$

where $\bar{\mathbf{x}}^{(R)} = \frac{1}{W_R}\sum_{r=0}^{R} w_r\mathbf{x}^{(r)}$ and we use Jensen's inequality ($F$ is convex) in the first inequality. By tuning the learning rate carefully, we get

$$F(\bar{\mathbf{x}}^R) - F(\mathbf{x}^*) = \mathcal{O}\left(\frac{\sigma D}{\sqrt{SKR}} + \sqrt{1 - \frac{S}{M}} \cdot \frac{\zeta_* D}{\sqrt{SR}} + \frac{\left(L\sigma^2 D^4\right)^{1/3}}{K^{1/3}R^{2/3}} + \frac{\left(L\zeta_*^2 D^4\right)^{1/3}}{R^{2/3}} + \frac{LD^2}{R}\right) \tag{74}$$

where $D := \|\mathbf{x}^{(0)} - \mathbf{x}^*\|$. Eq. (73) and Eq. (74) are the upper bounds with partial client participation. In particular, when $S = M$, we can get the claim of the strongly convex case of Theorem 2 and Corollary 2. $\qquad\square$

### E.3 Non-convex case

**Lemma 13.** *Let Assumptions 1, 2, 3b hold. If the learning rate satisfies $\eta \leq \frac{1}{6LK}$, then it holds that*

$$\mathbb{E}\left[F(\mathbf{x}^{(r+1)}) - F(\mathbf{x}^{(r)})\right] \leq -\frac{K\eta}{2}\mathbb{E}\left[\left\|\nabla F(\mathbf{x}^{(r)})\right\|^2\right] + \frac{LK\eta^2\sigma^2}{S}$$

$$+ \frac{L^2\eta}{2S}\sum_{m=1}^{S}\sum_{k=0}^{K-1}\mathbb{E}\left[\left\|\mathbf{x}_{m,k}^{(r)} - \mathbf{x}^{(r)}\right\|^2\right] \tag{75}$$

*Proof.* According to Algorithm 2, the overall model updates of PFL after one complete training round (with $S$ clients selected for training) is

$$\Delta\mathbf{x} = \mathbf{x}^{(r+1)} - \mathbf{x}^{(r)} = -\frac{\eta}{S}\sum_{m=1}^{S}\sum_{k=0}^{K-1}\mathbf{g}_{\pi_m,k}^{(r)},$$

where $\mathbf{g}_{\pi_m,k}^{(r)} = \nabla f_{\pi_m}(\mathbf{x}_{m,k}^{(r)}; \xi)$ is the stochastic gradient of $F_{\pi_m}$ regarding the vector $\mathbf{x}_{m,k}^{(r)}$. Thus,

$$\mathbb{E}[\Delta \mathbf{x}] = -\frac{\eta}{S} \sum_{m=1}^{S} \sum_{k=0}^{K-1} \mathbb{E}[\nabla F_{\pi_m}(\mathbf{x}_{m,k})]$$

In the following, we focus on a single training round, and hence we drop the superscripts $r$ for a while, e.g., writing $\mathbf{x}_{m,k}$ to replace $\mathbf{x}_{m,k}^{(r)}$. Specially, we would like to use $\mathbf{x}$ to replace $\mathbf{x}_{1,0}^{(r)}$. Without otherwise stated, the expectation is conditioned on $\mathbf{x}^{(r)}$.

Starting from the smoothness of $F$ (applying Eq. (16), $D_F(\boldsymbol{x}, \boldsymbol{y}) \leq \frac{L}{2} \|\boldsymbol{x} - \boldsymbol{y}\|^2$ with $\boldsymbol{x} = \mathbf{x} + \Delta \mathbf{x}$, $\boldsymbol{y} = \mathbf{x}$), and substituting the overall updates, we have

$$\mathbb{E}[F(\mathbf{x} + \Delta \mathbf{x}) - F(\mathbf{x})]$$

$$\leq \mathbb{E}[\langle \nabla F(\mathbf{x}), \Delta \mathbf{x} \rangle] + \frac{L}{2} \mathbb{E}\left[\|\Delta \mathbf{x}\|^2\right]$$

$$\leq -\frac{\eta}{S} \sum_{m=1}^{S} \sum_{k=0}^{K-1} \mathbb{E}[\langle \nabla F(\mathbf{x}), \nabla F_{\pi_m}(\mathbf{x}_{m,k}) \rangle] + \frac{L\eta^2}{2S^2} \mathbb{E}\left[\left\|\sum_{m=1}^{S} \sum_{k=0}^{K-1} \mathbf{g}_{\pi_m,k}\right\|^2\right] \quad (76)$$

For the first term on the right hand side in (76), using the fact that $2\langle a, b \rangle = \|a\|^2 + \|b\|^2 - \|a - b\|^2$ with $a = \nabla F(\mathbf{x})$ and $b = \nabla F_{\pi_m}(\mathbf{x}_{m,k})$, we have

$$-\frac{\eta}{S} \sum_{m=1}^{S} \sum_{k=0}^{K-1} \mathbb{E}[\langle \nabla F(\mathbf{x}), \nabla F_{\pi_m}(\mathbf{x}_{m,k}) \rangle]$$

$$= -\frac{\eta}{2S} \sum_{m=1}^{S} \sum_{k=0}^{K-1} \mathbb{E}\left[\|\nabla F(\mathbf{x})\|^2 + \|\nabla F_{\pi_m}(\mathbf{x}_{m,k})\|^2 - \|\nabla F_{\pi_m}(\mathbf{x}_{m,k}) - \nabla F(\mathbf{x})\|^2\right]$$

$$\overset{\text{Asm. 1}}{\leq} -\frac{K\eta}{2} \|\nabla F(\mathbf{x})\|^2 - \frac{\eta}{2S} \sum_{m=1}^{S} \sum_{k=0}^{K-1} \mathbb{E}\left[\|\nabla F_{\pi_m}(\mathbf{x}_{m,k})\|^2\right] + \frac{L^2\eta}{2S} \sum_{m=1}^{S} \sum_{k=0}^{K-1} \mathbb{E}\left[\|\mathbf{x}_{m,k} - \mathbf{x}\|^2\right]$$

$$(77)$$

For the third term on the right hand side in (76), using Jensen's inequality, we have

$$\frac{L\eta^2}{2S^2} \mathbb{E}\left[\left\|\sum_{m=1}^{S} \sum_{k=0}^{K-1} \mathbf{g}_{\pi_m,k}\right\|^2\right]$$

$$\leq \frac{L\eta^2}{S^2} \mathbb{E}\left[\left\|\sum_{m=1}^{S} \sum_{k=0}^{K-1} \mathbf{g}_{\pi_m,k} - \sum_{m=1}^{S} \sum_{k=0}^{K-1} \nabla F_{\pi_m}(\mathbf{x}_{m,k})\right\|^2\right] + L\eta^2 \mathbb{E}\left[\left\|\sum_{m=1}^{S} \sum_{k=0}^{K-1} \nabla F_{\pi_m}(\mathbf{x}_{m,k})\right\|^2\right]$$

$$\leq \frac{LK\eta^2\sigma^2}{S} + \frac{LK\eta^2}{S} \sum_{m=1}^{S} \sum_{k=0}^{K-1} \mathbb{E}\left[\|\nabla F_{\pi_m}(\mathbf{x}_{m,k})\|^2\right], \quad (78)$$

where we use independence and Lemma 1 for the first term (see Eq. (66)) and Jensen's inequality for the second term in the preceding inequality.

Substituting (77) and (78) into (76), we have

$$\mathbb{E}[F(\mathbf{x} + \Delta \mathbf{x}) - F(\mathbf{x})] \leq -\frac{K\eta}{2} \|\nabla F(\mathbf{x})\|^2 + \frac{LK\eta^2\sigma^2}{S} + \frac{L^2\eta}{2S} \sum_{m=1}^{S} \sum_{k=0}^{K-1} \mathbb{E}\left[\|\mathbf{x}_{m,k} - \mathbf{x}\|^2\right]$$

$$- \frac{\eta}{2S}(1 - 2LK\eta) \sum_{m=1}^{S} \sum_{k=0}^{K-1} \mathbb{E}\left[\|\nabla F_{\pi_m}(\mathbf{x}_{m,k})\|^2\right]$$

Since $\eta \leq \frac{1}{6LK}$, the last term on the right hand side in the preceding inequality is negative. Then

$$\mathbb{E}[F(\mathbf{x} + \Delta \mathbf{x}) - F(\mathbf{x})] \leq -\frac{K\eta}{2} \|\nabla F(\mathbf{x})\|^2 + \frac{LK\eta^2\sigma^2}{S} + \frac{L^2\eta}{2S} \sum_{m=1}^{S} \sum_{k=0}^{K-1} \mathbb{E}\left[\|\mathbf{x}_{m,k} - \mathbf{x}\|^2\right]$$

The claim follows after recovering the superscripts and taking unconditional expectation. $\qquad\square$

### E.3.1 Bounding the client drift with Assumption 3a

**Lemma 14.** *Let Assumptions 1, 2, 3b hold. If the learning rate satisfies $\eta \leq \frac{1}{6LK}$, then the client drift is bounded:*

$$\mathcal{E}_r \leq \frac{9}{4}K^2\eta^2\sigma^2 + \frac{3}{2}K^3\eta^2\zeta^2 + \frac{3}{2}K^3\eta^2(\beta^2+1)\mathbb{E}\left[\left\|\nabla F(\mathbf{x}^{(r)})\right\|^2\right] \tag{79}$$

*Proof.* According to Algorithm 2, the model updates of PFL from $\mathbf{x}^{(r)}$ to $\mathbf{x}_{m,k}^{(r)}$ is

$$\mathbf{x}_{m,k}^{(r)} - \mathbf{x}^{(r)} = -\eta\sum_{j=0}^{k-1}\mathbf{g}_{\pi_m,j}^{(r)}$$

In the following, we focus on a single training round, and hence we drop the superscripts $r$ for a while, e.g., writing $\mathbf{x}_{m,k}$ to replace $\mathbf{x}_{m,k}^{(r)}$. Specially, we would like to use $\mathbf{x}$ to replace $\mathbf{x}_{1,0}^{(r)}$. Without otherwise stated, the expectation is conditioned on $\mathbf{x}^{(r)}$.

We use Jensen's inequality to bound the term $\mathbb{E}\left[\|\mathbf{x}_{m,k}-\mathbf{x}\|^2\right] = \eta^2\mathbb{E}\left[\left\|\sum_{j=0}^{k-1}\mathbf{g}_{\pi_m,j}\right\|^2\right]$:

$$\mathbb{E}\left[\|\mathbf{x}_{m,k}-\mathbf{x}\|^2\right] \leq 4\eta^2\mathbb{E}\left[\left\|\sum_{j=0}^{k-1}(\mathbf{g}_{\pi_m,j}-\nabla F_{\pi_m}(\mathbf{x}_{m,j}))\right\|^2\right] + 4\eta^2\mathbb{E}\left[\left\|\sum_{j=0}^{k-1}(\nabla F_{\pi_m}(\mathbf{x}_{m,j})-\nabla F_{\pi_m}(\mathbf{x}))\right\|^2\right]$$

$$+ 4\eta^2\mathbb{E}\left[\left\|\sum_{j=0}^{k-1}(\nabla F_{\pi_m}(\mathbf{x})-\nabla F(\mathbf{x}))\right\|^2\right] + 4\eta^2\mathbb{E}\left[\left\|\sum_{j=0}^{k-1}\nabla F(\mathbf{x})\right\|^2\right]$$

Applying Lemma 1 to the first term and Jensen's inequality to the last three terms on the right hand side in the preceding inequality, respectively, we get

$$\mathbb{E}\left[\|\mathbf{x}_{m,k}-\mathbf{x}\|^2\right] \leq 4\eta^2\sum_{j=0}^{k-1}\mathbb{E}\left[\|\mathbf{g}_{\pi_m,j}-\nabla F_{\pi_m}(\mathbf{x}_{m,j})\|^2\right] + 4k\eta^2\sum_{j=0}^{k-1}\mathbb{E}\left[\|\nabla F_{\pi_m}(\mathbf{x}_{m,j})-\nabla F_{\pi_m}(\mathbf{x})\|^2\right]$$

$$+ 4k^2\eta^2\mathbb{E}\left[\|\nabla F_{\pi_m}(\mathbf{x})-\nabla F(\mathbf{x})\|^2\right] + 4k^2\eta^2\mathbb{E}\left[\|\nabla F(\mathbf{x})\|^2\right]$$

The first term can be bounded by $4k\eta^2\sigma^2$ with Assumption 2. The second term can be bounded by $4L^2k\eta^2\sum_{j=0}^{k-1}\mathbb{E}\left[\|\mathbf{x}_{m,j}-\mathbf{x}\|^2\right]$ with Assumption 1. The third term can be bounded by $4k^2\eta^2\left(\beta^2\|\nabla F(\mathbf{x})\|^2+\zeta^2\right)$ with Assumption 3a. Thus, we have

$$\mathbb{E}\left[\|\mathbf{x}_{m,k}-\mathbf{x}\|^2\right] \leq 4k\eta^2\sigma^2 + 4L^2k\eta^2\sum_{j=0}^{k-1}\mathbb{E}\left[\|\mathbf{x}_{m,j}-\mathbf{x}\|^2\right] + 4k^2\eta^2(\beta^2+1)\|\nabla F(\mathbf{x})\|^2 + 4k^2\eta^2\zeta^2$$

Then returning to $\mathcal{E}_r := \frac{1}{S}\sum_{m=1}^{S}\sum_{k=0}^{K-1}\mathbb{E}\left[\|\mathbf{x}_{m,k}-\mathbf{x}\|^2\right]$, we have

$$\mathcal{E}_r = 4\eta^2\sigma^2\sum_{k=0}^{K-1}k + 4L^2\eta^2\frac{1}{S}\sum_{m=1}^{S}\sum_{k=0}^{K-1}k\sum_{j=0}^{k-1}\mathbb{E}\left[\|\mathbf{x}_{m,j}-\mathbf{x}\|^2\right] + 4\eta^2(\beta^2+1)\|\nabla F(\mathbf{x})\|^2\sum_{k=0}^{K-1}k^2 + 4\eta^2\zeta^2\sum_{k=0}^{K-1}k^2$$

Using the facts that $\sum_{k=1}^{K-1}k = \frac{(K-1)K}{2} \leq \frac{K^2}{2}$ and $\sum_{k=1}^{K-1}k^2 = \frac{(K-1)K(2K-1)}{6} \leq \frac{K^3}{3}$, we can simplify the preceding inequality:

$$\mathcal{E}_r \leq 2K^2\eta^2\sigma^2 + 2L^2K^2\eta^2\frac{1}{S}\sum_{m=1}^{S}\sum_{j=0}^{k-1}\mathbb{E}\left[\|\mathbf{x}_{m,j}-\mathbf{x}\|^2\right] + \frac{4}{3}K^3\eta^2(\beta^2+1)\|\nabla F(\mathbf{x})\|^2 + \frac{4}{3}K^3\eta^2\zeta^2$$

After rearranging the preceding inequality, we get

$$(1 - 2L^2K^2\eta^2)\mathcal{E}_r \leq 2K^2\eta^2\sigma^2 + \frac{4}{3}K^3\eta^2\zeta^2 + \frac{4}{3}K^3\eta^2(\beta^2+1)\left\|\nabla F(\mathbf{x})\right\|^2$$

Finally, using the condition that $\eta \leq \frac{1}{6LK}$, which implies $1 - 2L^2K^2\eta^2 \geq \frac{8}{9}$, we have

$$\mathcal{E}_r \leq \frac{9}{4}K^2\eta^2\sigma^2 + \frac{3}{2}K^3\eta^2\zeta^2 + \frac{3}{2}K^3\eta^2(\beta^2+1)\left\|\nabla F(\mathbf{x})\right\|^2$$

The claim follows after recovering the superscripts and taking unconditional expectations. $\qquad\square$

### E.3.2   Proof of non-convex case of Theorem 2

*Proof of non-convex case of Theorem 2.* Substituting (52) into (48) and using $\eta \leq \frac{1}{6LK(\beta+1)}$, we can simplify the recursion as follows:

$$\mathbb{E}\left[F(\mathbf{x}^{(r+1)}) - F(\mathbf{x}^{(r)})\right] \leq -\frac{1}{3}K\eta\mathbb{E}\left[\left\|\nabla F(\mathbf{x}^{(r)})\right\|^2\right] + \frac{LK\eta^2\sigma^2}{S} + \frac{9}{8}L^2K^2\eta^3\sigma^2 + \frac{3}{4}L^2K^3\eta^3\zeta^2$$

Letting $\tilde{\eta} := K\eta$, subtracting $F^*$ from both sides and then rearranging the terms, we have

$$\mathbb{E}\left[F(\mathbf{x}^{(r+1)}) - F^*\right] \leq \mathbb{E}\left[F(\mathbf{x}^{(r)}) - F^*\right] - \frac{\tilde{\eta}}{3}\mathbb{E}\left[\left\|\nabla F(\mathbf{x}^{(r)})\right\|^2\right] + \frac{L\tilde{\eta}^2\sigma^2}{SK} + \frac{9L^2\tilde{\eta}^3\sigma^2}{8K} + \frac{3}{4}L^2\tilde{\eta}^3\zeta^2$$

Then applying Lemma 8 with $t = r$ ($T = R$), $\gamma = \tilde{\eta}$, $r_t = \mathbb{E}\left[F(\mathbf{x}^{(r)}) - F^*\right]$, $b = \frac{1}{3}$, $s_t = \mathbb{E}\left[\left\|\nabla F(\mathbf{x}^{(r)})\right\|^2\right]$, $w_t = 1$, $c_1 = \frac{L\sigma^2}{SK}$, $c_2 = \frac{9L^2\sigma^2}{8K} + \frac{3}{4}L^2\zeta^2$ and $\frac{1}{d} = \frac{1}{6L(\beta+1)}$ ($\tilde{\eta} = K\eta \leq \frac{1}{6L(\beta+1)}$), we have

$$\min_{0\leq r\leq R}\mathbb{E}\left[\left\|\nabla F(\mathbf{x}^{(r)})\right\|^2\right] \leq \frac{3\left(F(\mathbf{x}^0) - F^*\right)}{\tilde{\eta}R} + \frac{3L\tilde{\eta}\sigma^2}{SK} + \frac{27L^2\tilde{\eta}^2\sigma^2}{8K} + \frac{9}{4}L^2\tilde{\eta}^2\zeta^2 \tag{80}$$

where we use $\min_{0\leq r\leq R}\mathbb{E}\left[\left\|\nabla F(\mathbf{x}^{(r)})\right\|^2\right] \leq \frac{1}{R+1}\sum_{r=0}^{R}\mathbb{E}\left[\left\|\nabla F(\mathbf{x}^{(r)})\right\|^2\right]$. Then, tuning the learning rate carefully, we get

$$\min_{0\leq r\leq R}\mathbb{E}\left[\left\|\nabla F(\mathbf{x}^{(r)})\right\|^2\right] = \mathcal{O}\left(\frac{\left(L\sigma^2 A\right)^{1/2}}{\sqrt{SKR}} + \frac{\left(L^2\sigma^2 A^2\right)^{1/3}}{K^{1/3}R^{2/3}} + \frac{\left(L^2\zeta^2 A^2\right)^{1/3}}{R^{2/3}} + \frac{L\beta A}{R}\right) \tag{81}$$

where $A := F(\mathbf{x}^0) - F^*$. Eq. (80) and Eq. (81) are the upper bounds with partial client participation. In particular, when $S = M$, we get the claim of the non-convex case of Theorem 2 and Corollary 2. $\qquad\square$

# F   Simulations on quadratic functions

Nine groups of simulated experiments with various degrees of heterogeneity are provided in Table 6 as a extension of the experiment in Subsection 4.1. Figure 7 plots the results of PFL and SFL with various combinations of $\delta$ and $\zeta_*$.

Table 6: Settings of simulated experiments. Each setting has two local objectives (i.e., $M = 2$) and shares the same global objective. Choosing large value of $\zeta_*$ and $\delta$ means higher heterogeneity. The definitions of $\zeta_*$ and $\delta$ can be found in Subsection 4.1.

| Settings | $\zeta_* = 1$ | $\zeta_* = 10$ | $\zeta_* = 100$ |
|---|---|---|---|
| $\delta = 0$ | $\begin{cases} F_1(x) = \frac{1}{2}x^2 + x \\ F_2(x) = \frac{1}{2}x^2 - x \end{cases}$ | $\begin{cases} F_1(x) = \frac{1}{2}x^2 + 10x \\ F_2(x) = \frac{1}{2}x^2 - 10x \end{cases}$ | $\begin{cases} F_1(x) = \frac{1}{2}x^2 + 100x \\ F_2(x) = \frac{1}{2}x^2 - 100x \end{cases}$ |
| $\delta = \frac{1}{3}$ | $\begin{cases} F_1(x) = \frac{2}{3}x^2 + x \\ F_2(x) = \frac{1}{3}x^2 - x \end{cases}$ | $\begin{cases} F_1(x) = \frac{2}{3}x^2 + 10x \\ F_2(x) = \frac{1}{3}x^2 - 10x \end{cases}$ | $\begin{cases} F_1(x) = \frac{2}{3}x^2 + 100x \\ F_2(x) = \frac{1}{3}x^2 - 100x \end{cases}$ |
| $\delta = 1$ | $\begin{cases} F_1(x) = x^2 + x \\ F_2(x) = -x \end{cases}$ | $\begin{cases} F_1(x) = x^2 + 10x \\ F_2(x) = -10x \end{cases}$ | $\begin{cases} F_1(x) = x^2 + 100x \\ F_2(x) = -100x \end{cases}$ |

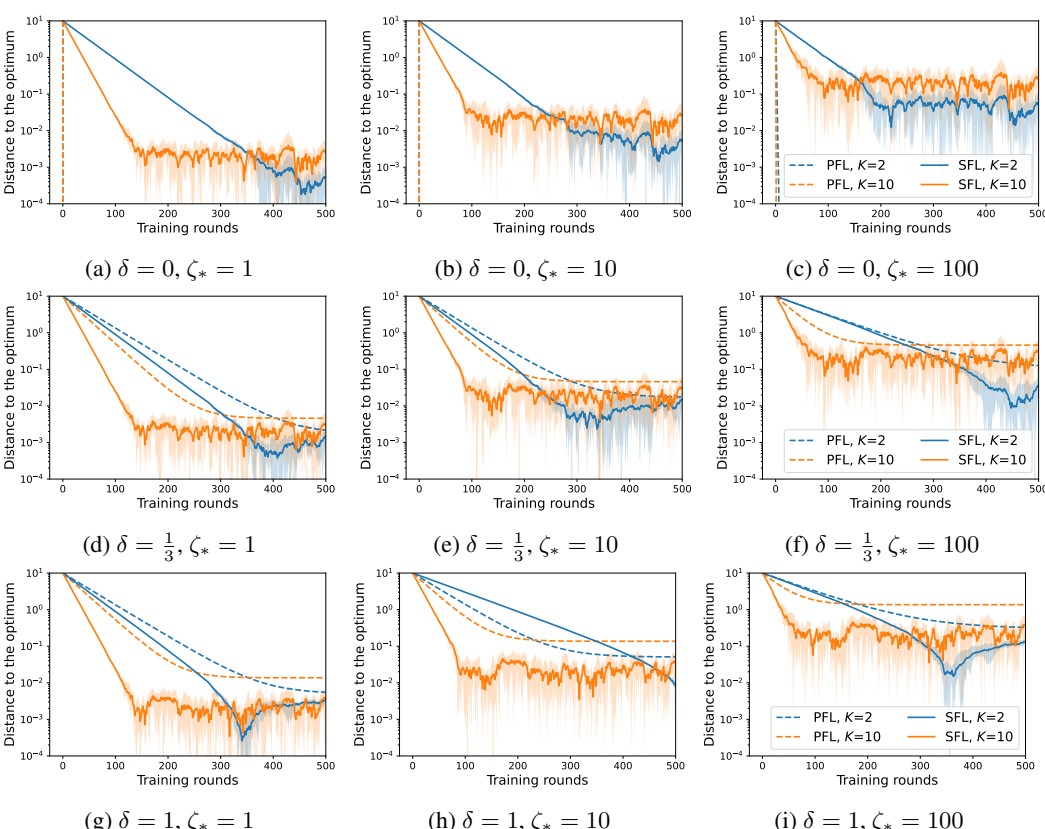

Figure 7: Simulations on quadratic functions. The best learning rates are chosen from [0.003, 0.006, 0.01, 0.03, 0.06, 0.1, 0.3, 0.6] with grid search. We run each experiments for 5 random seeds. Shaded areas show the min-max values.

# G More experimental details

This section serves as a supplement and enhancement to Section 4. Our code is partly from Gao et al. (2021); Zeng et al. (2021); Jhunjhunwala et al. (2023) (more references are included in the code), and it is available at https://github.com/liyipeng00/convergence.

## G.1 Extended Dirichlet partition

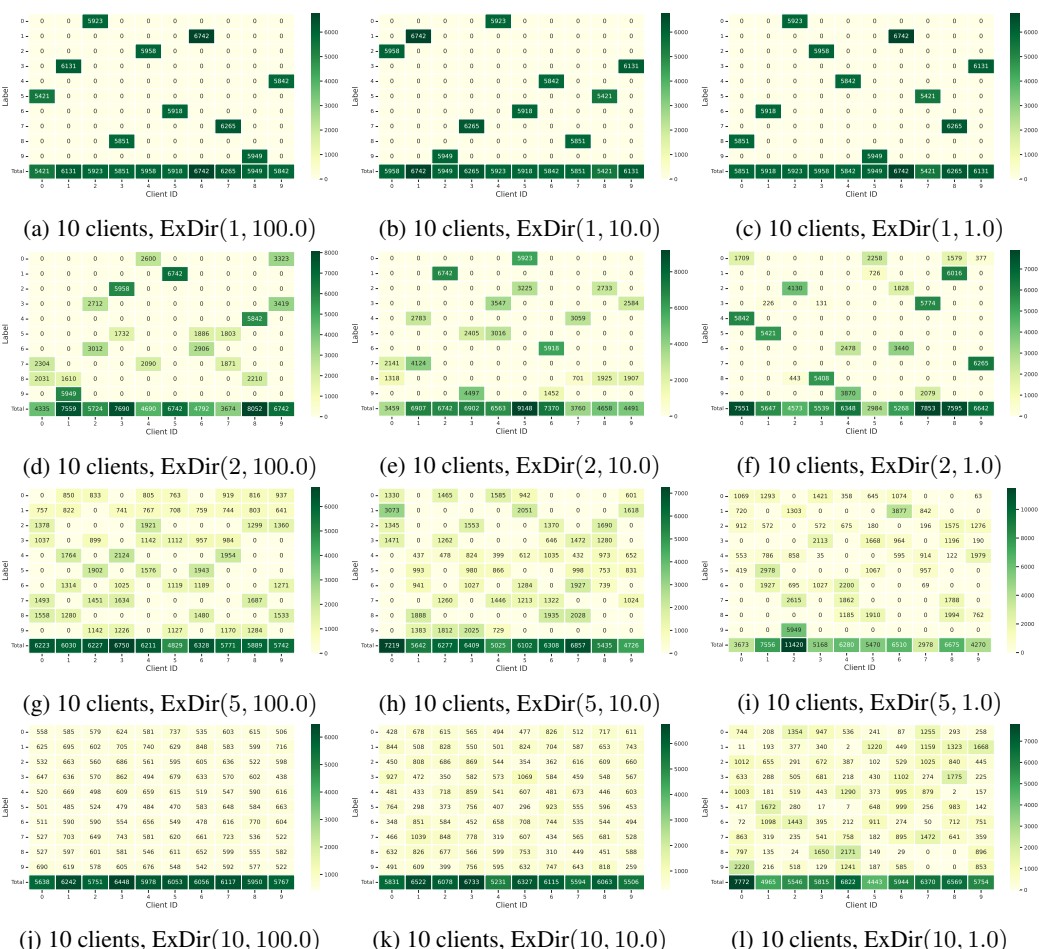

(a) 10 clients, ExDir(1, 100.0)  (b) 10 clients, ExDir(1, 10.0)  (c) 10 clients, ExDir(1, 1.0)

(d) 10 clients, ExDir(2, 100.0)  (e) 10 clients, ExDir(2, 10.0)  (f) 10 clients, ExDir(2, 1.0)

(g) 10 clients, ExDir(5, 100.0)  (h) 10 clients, ExDir(5, 10.0)  (i) 10 clients, ExDir(5, 1.0)

(j) 10 clients, ExDir(10, 100.0)  (k) 10 clients, ExDir(10, 10.0)  (l) 10 clients, ExDir(10, 1.0)

Figure 8: Visualization of partitioning results on MNIST by Extended Dirichlet strategy. The $x$-axis indicates client IDs and the $y$-axis indicates labels. The value in each cell is the number of data samples of a label belonging to that client. For the first row, there are only one possible results in the case where each client owns one label with 10 clients and 10 labels in total, so these three partitions are the same. For the second, third and forth rows, data heterogeneity increases from left to right.

**Baseline.** There are two common partition strategies to simulate the heterogeneous settings in the FL literature. According to Li et al. (2022), they can be summarized as follows:

a) Quantity-based class imbalance: Under this strategy, each client is allocated data samples from a fixed number of classes. The initial implementation comes from McMahan et al. (2017), and has extended by Li et al. (2022) recently. Specifically, Li et al. first randomly assign $C$ different classes to each client. Then, the samples of each class are divided randomly and equally into the clients which owns the class.

b) Distribution-based class imbalance: Under this strategy, each client is allocated a proportion of the data samples of each class according to Dirichlet distribution. The initial implementation, to the best of our knowledge, comes from Yurochkin et al. (2019). For each class $c$, Yurochkin et al. draw $p_c \sim \text{Dir}(\alpha q)$ and allocate a $p_{c,m}$ proportion of the data samples of class $k$ to client $m$. Here $q$ is the prior distribution, which is set to $1$.

**Extended Dirichlet strategy.** This is to generate arbitrarily heterogeneous data across clients by combining the two strategies above. The difference is to add a step of allocating classes (labels) to determine the number of classes per client (denoted by $C$) before allocating samples via Dirichlet distribution (with concentrate parameter $\alpha$). Thus, the extended strategy can be denoted by $\text{ExDir}(C, \alpha)$. The implementation is as follows (one partitioning example is shown in Figure 8):

- Allocating classes: We randomly allocate $C$ different classes to each client. After assigning the classes, we can obtain the prior distribution $\boldsymbol{q}_c$ for each class $c$ (see Figure 9).

- Allocating samples: For each class $c$, we draw $\boldsymbol{p}_c \sim \text{Dir}(\alpha\boldsymbol{q}_c)$ and then allocate a $\boldsymbol{p}_{c,m}$ proportion of the samples of class $c$ to client $m$. For example, $\boldsymbol{q}_c = [1, 1, 0, 0, \ldots,]$ means that the samples of class $c$ are only allocated to the first 2 clients.

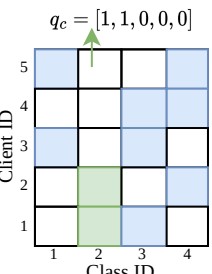

This strategy have two levels, the first level to allocate classes and the second level to allocate samples. We note that Reddi et al. (2021) use a two-level partition strategy to partition the CIFAR-100 dataset. They draw a multinomial distribution from the Dirichlet prior at the root ($\text{Dir}(\alpha)$) and a multinomial distribution from the Dirichlet prior at each coarse label ($\text{Dir}(\beta)$).

Figure 9: Allocating 2 classes (4 classes in total) to 5 clients.

## G.2 Gradient clipping.

Two partitions, the extreme setting $C = 1$ (i.e., $\text{ExDir}(1, 10.0)$) and the moderate settings $C = 2$ (i.e., $\text{ExDir}(2, 10.0)$) are used in the main body. For both settings, we use the gradient clipping to improve the stability of the algorithms as done in previous works Acar et al. (2021); Jhunjhunwala et al. (2023). Further, we note that the gradient clipping is critical for PFL and SFL to prevent divergence in the learning process on heterogeneous data, especially in the extreme setting. Empirically, we find that the fitting "max norm" of SFL is larger than PFL. Thus we trained VGG-9 on CIFAR-10 with PFL and SFL for various values of the max norm of gradient clipping to select the fitting value, and had some empirical observations for gradient clipping in FL. The experimental results are given in Table 7, Table 8 and Figure 10. The empirical observations are summarized as follows:

1) The fitting max norm of SFL is larger than PFL. When the max norm is set to be 20 for both algorithms, gradient clipping helps improve the performance of PFL, while it may even hurt that of SFL (e.g., see the 12-th row in Table 8).

2) The fitting max norm for the case with high data heterogeneity is larger than that with low data heterogeneity. This is suitable for both algorithms. (e.g., see the 12, 24-th rows in Table 8)

3) The fitting max norm for the case with more local update steps is larger than that with less local update steps. This is suitable for both algorithms. (e.g., see the 4, 8, 12-th rows in Table 8)

4) Gradient clipping with smaller values of the max norm exhibits a preference for a larger learning rate. This means that using gradient clipping makes the best learning rate larger. This phenomenon is more obvious when choosing smaller values of the max norm (see Table 7).

5) The fitting max norm is additionally affected by the model architecture, model size, and so on.

Finally, taking into account the experimental results and convenience, we set the max norm of gradient clipping to 10 for PFL and 50 for SFL for all settings in this paper.

Table 7: Test accuracies when using gradient clipping with various values of the max norm for VGG-9 on CIFAR-10. Other settings without being stated explicitly are identical to that in the main body. The results are computed over the last 40 training rounds (with 1000 training rounds in total). The highest test accuracy among different learning rates is marked in cyan for both algorithms.

| Setting | PFL | $10^{-2.0}$ | $10^{-1.5}$ | $10^{-1.0}$ | $10^{-0.5}$ | SFL | $10^{-2.5}$ | $10^{-2.0}$ | $10^{-1.5}$ | $10^{-1.0}$ |
|---|---|---|---|---|---|---|---|---|---|---|
| VGG-9, $C=1$, $K=5$, w/o clip | $\infty$ | 25.43 | 30.95 | 30.63 | 10.00 | $\infty$ | 43.93 | 53.79 | 57.69 | 10.00 |
| VGG-9, $C=1$, $K=5$, w/ clip | 20 | 21.92 | 31.04 | 32.41 | 24.99 | 100 | 43.93 | 53.79 | 57.69 | 10.00 |
| VGG-9, $C=1$, $K=5$, w/ clip | 10 | 12.51 | 25.67 | 34.89 | 28.77 | 50 | 43.79 | 53.73 | 58.63 | 10.00 |
| VGG-9, $C=1$, $K=5$, w/ clip | 5 | 10.54 | 16.72 | 27.01 | 35.11 | 20 | 43.12 | 53.17 | 57.96 | 10.00 |
| VGG-9, $C=1$, $K=20$, w/o clip | $\infty$ | 24.23 | 27.53 | 26.91 | 10.00 | $\infty$ | 35.63 | 10.00 | 10.00 | 10.00 |
| VGG-9, $C=1$, $K=20$, w/ clip | 20 | 19.44 | 27.60 | 26.41 | 15.00 | 100 | 35.63 | 10.00 | 10.00 | 10.00 |
| VGG-9, $C=1$, $K=20$, w/ clip | 10 | 11.51 | 22.81 | 28.79 | 21.10 | 50 | 34.55 | 27.11 | 10.00 | 10.00 |
| VGG-9, $C=1$, $K=20$, w/ clip | 5 | 10.39 | 14.56 | 22.48 | 27.26 | 20 | 30.49 | 10.00 | 10.00 | 10.00 |
| VGG-9, $C=1$, $K=50$, w/o clip | $\infty$ | 22.44 | 23.70 | 20.97 | 10.00 | $\infty$ | 25.11 | 10.00 | 10.00 | 10.00 |
| VGG-9, $C=1$, $K=50$, w/ clip | 20 | 17.82 | 23.80 | 20.72 | 10.00 | 100 | 25.11 | 10.00 | 10.00 | 10.00 |
| VGG-9, $C=1$, $K=50$, w/ clip | 10 | 10.14 | 20.58 | 21.95 | 10.00 | 50 | 23.41 | 10.00 | 10.00 | 10.00 |
| VGG-9, $C=1$, $K=50$, w/ clip | 5 | 10.31 | 10.44 | 18.25 | 18.02 | 20 | 18.27 | 10.00 | 10.00 | 10.00 |
| VGG-9, $C=2$, $K=5$, w/o clip | $\infty$ | 41.36 | 51.34 | 55.22 | 10.00 | $\infty$ | 58.33 | 69.14 | 71.58 | 10.00 |
| VGG-9, $C=2$, $K=5$, w/ clip | 20 | 41.46 | 51.30 | 56.54 | 47.47 | 100 | 58.33 | 69.14 | 71.58 | 10.00 |
| VGG-9, $C=2$, $K=5$, w/ clip | 10 | 38.50 | 50.99 | 57.09 | 53.46 | 50 | 58.35 | 68.75 | 71.24 | 10.00 |
| VGG-9, $C=2$, $K=5$, w/ clip | 5 | 26.07 | 46.49 | 58.26 | 56.71 | 20 | 57.81 | 69.17 | 70.94 | 10.00 |
| VGG-9, $C=2$, $K=20$, w/o clip | $\infty$ | 55.64 | 64.04 | 10.00 | 10.00 | $\infty$ | 60.56 | 67.70 | 64.94 | 10.00 |
| VGG-9, $C=2$, $K=20$, w/ clip | 20 | 55.70 | 64.14 | 66.57 | 10.00 | 100 | 59.48 | 68.11 | 64.94 | 10.00 |
| VGG-9, $C=2$, $K=20$, w/ clip | 10 | 54.70 | 63.97 | 68.07 | 61.92 | 50 | 60.82 | 67.93 | 66.50 | 10.00 |
| VGG-9, $C=2$, $K=20$, w/ clip | 5 | 47.32 | 61.61 | 67.11 | 63.56 | 20 | 58.36 | 67.65 | 67.74 | 10.00 |
| VGG-9, $C=2$, $K=50$, w/o clip | $\infty$ | 63.93 | 67.85 | 10.00 | 10.00 | $\infty$ | 61.93 | 68.05 | 61.77 | 10.00 |
| VGG-9, $C=2$, $K=50$, w/ clip | 20 | 63.94 | 68.26 | 67.15 | 59.46 | 100 | 62.36 | 67.01 | 62.84 | 10.00 |
| VGG-9, $C=2$, $K=50$, w/ clip | 10 | 62.72 | 67.57 | 69.11 | 64.21 | 50 | 62.27 | 67.52 | 62.59 | 10.00 |
| VGG-9, $C=2$, $K=50$, w/ clip | 5 | 58.52 | 65.34 | 66.77 | 64.75 | 20 | 59.80 | 68.72 | 64.26 | 38.22 |

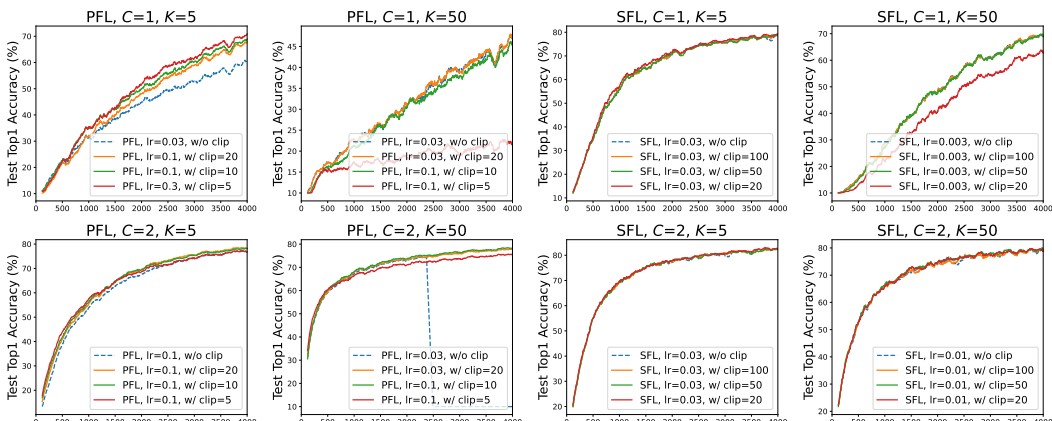

Figure 10: The corresponding training curves of Table 8 (VGG-9 on CIFAR-10).

Table 8: The best learning rate (selected in Table 7) and its corresponding test accuracies in the short run (1000 training rounds) and in the long run (4000 training rounds). The results are computed over the last 40 training rounds in the short run (the 5-th, 9-th columns) and 100 training rounds in the long run (the 6-th, 10-th columns). That the algorithms diverge when without gradient clipping makes the result with $^\dagger$. The results that deviate from the vanilla case (w/o gradient clipping) considerably (more than 2%) are marked in magenta and teal.

| | Setting | PFL | Lr | Acc. | Acc. | SFL | Lr | Acc. | Acc. |
|---|---|---|---|---|---|---|---|---|---|
| | | | | | {CIFAR-10, $C = 1$} | | | | |
| 1 | VGG-9, $K = 5$, w/o clip | $\infty$ | $10^{-1.5}$ | 30.95 | 60.75 | $\infty$ | $10^{-1.5}$ | 57.69 | 78.56 |
| 2 | VGG-9, $K = 5$, w/ clip | 20 | $10^{-1.0}$ | 32.41 | 67.97 (+7.2) | 100 | $10^{-1.5}$ | 57.69 | 78.75 |
| 3 | VGG-9, $K = 5$, w/ clip | 10 | $10^{-1.0}$ | 34.89 (+3.9) | 69.10 (+8.4) | 50 | $10^{-1.5}$ | 58.63 | 78.56 |
| 4 | VGG-9, $K = 5$, w/ clip | 5 | $10^{-0.5}$ | 35.11 (+4.2) | 71.07 (+10.3) | 20 | $10^{-1.5}$ | 57.96 | 79.06 |
| 5 | VGG-9, $K = 20$, w/o clip | $\infty$ | $10^{-1.5}$ | 27.53 | 56.89 | $\infty$ | $10^{-2.5}$ | 35.63 | 72.90 |
| 6 | VGG-9, $K = 20$, w/ clip | 20 | $10^{-1.5}$ | 27.60 | 57.01 | 100 | $10^{-2.5}$ | 35.63 | 73.06 |
| 7 | VGG-9, $K = 20$, w/ clip | 10 | $10^{-1.0}$ | 28.79 | 64.11 (+7.2) | 50 | $10^{-2.5}$ | 34.55 | 73.16 |
| 8 | VGG-9, $K = 20$, w/ clip | 5 | $10^{-0.5}$ | 27.26 | 62.31 (+5.4) | 20 | $10^{-2.5}$ | 30.49 (-5.1) | 69.66 (-3.2) |
| 9 | VGG-9, $K = 50$, w/o clip | $\infty$ | $10^{-1.5}$ | 23.70 | 48.29 | $\infty$ | $10^{-2.5}$ | 25.11 | 69.10 |
| 10 | VGG-9, $K = 50$, w/ clip | 20 | $10^{-1.5}$ | 23.80 | 47.64 | 100 | $10^{-2.5}$ | 25.11 | 69.01 |
| 11 | VGG-9, $K = 50$, w/ clip | 10 | $10^{-1.0}$ | 21.95 | 46.21 (-2.1) | 50 | $10^{-2.5}$ | 23.41 | 68.71 |
| 12 | VGG-9, $K = 50$, w/ clip | 5 | $10^{-1.0}$ | 18.25 (-5.5) | 22.58 (-25.7) | 20 | $10^{-2.5}$ | 18.27 (-6.8) | 62.70 (-6.4) |
| | | | | | {CIFAR-10, $C = 2$} | | | | |
| 13 | VGG-9, $K = 5$, w/o clip | $\infty$ | $10^{-1.0}$ | 55.22 | 76.98 | $\infty$ | $10^{-1.5}$ | 71.58 | 82.09 |
| 14 | VGG-9, $K = 5$, w/ clip | 20 | $10^{-1.0}$ | 56.54 | 78.28 | 100 | $10^{-1.5}$ | 71.58 | 82.17 |
| 15 | VGG-9, $K = 5$, w/ clip | 10 | $10^{-1.0}$ | 57.09 | 78.18 | 50 | $10^{-1.5}$ | 71.24 | 82.18 |
| 16 | VGG-9, $K = 5$, w/ clip | 5 | $10^{-1.0}$ | 58.26 (+3.0) | 76.69 | 20 | $10^{-1.5}$ | 70.94 | 82.48 |
| 17 | VGG-9, $K = 20$, w/o clip | $\infty$ | $10^{-1.5}$ | 64.04 | 77.21 | $\infty$ | $10^{-2.0}$ | 67.70 | 81.31 |
| 18 | VGG-9, $K = 20$, w/ clip | 20 | $10^{-1.0}$ | 66.57 (+2.5) | 78.87 | 100 | $10^{-2.0}$ | 68.11 | 82.08 |
| 19 | VGG-9, $K = 20$, w/ clip | 10 | $10^{-1.0}$ | 68.07 (+4.0) | 78.85 | 50 | $10^{-2.0}$ | 67.93 | 81.50 |
| 20 | VGG-9, $K = 20$, w/ clip | 5 | $10^{-1.0}$ | 67.11 (+3.1) | 76.66 | 20 | $10^{-1.5}$ | 67.74 | 77.59 (-3.7) |
| 21 | VGG-9, $K = 50$, w/o clip | $\infty$ | $10^{-1.5}$ | 67.85 | $10.00^\dagger$ | $\infty$ | $10^{-2.0}$ | 68.05 | 79.30 |
| 22 | VGG-9, $K = 50$, w/ clip | 20 | $10^{-1.5}$ | 68.26 | 77.83 | 100 | $10^{-2.0}$ | 67.01 | 78.88 |
| 23 | VGG-9, $K = 50$, w/ clip | 10 | $10^{-1.0}$ | 69.11 | 78.13 | 50 | $10^{-2.0}$ | 67.52 | 79.42 |
| 24 | VGG-9, $K = 50$, w/ clip | 5 | $10^{-1.0}$ | 66.77 | 75.42 | 20 | $10^{-2.0}$ | 68.72 | 78.51 |

## G.3 Grid search

We use the grid search to find the best learning rate on one random seed "1234". Since we have observed that the best learning rate of PFL is smaller than SFL empirically, the grid for PFL is $\{10^{-2.0}, 10^{-1.5}, 10^{-1.0}, 10^{-0.5}\}$ ({0.01, 0.03, 0.1, 0.3} in fact) and the grid for SFL is $\{10^{-2.5}, 10^{-2.0}, 10^{-1.5}, 10^{-1.0}\}$ ({0.003, 0.01, 0.03, 0.1} in fact). We use these grids for all tasks in this paper, including MNIST and FMNIST in the next subsection.

One practical method used in Jhunjhunwala et al. (2023) to find the best learning rate is running the algorithms for a fewer training rounds and then comparing the short-run results by some metrics (e.g., training accuracy) when the computation resources are restrictive and the task is complex (e.g., CIFAR-10). However, we should pay attention to whether the chosen learning rates are appropriate in the specific scenarios, as the best learning rate in the short run may not be the best in the long run. One alternative method is using the short-run results to find some alternatives (coarse-grained search) and then using the long-run results to find the best one (fine-grained search).

In this paper, for CIFAR-10 and CINIC-10, we run the algorithms for 1000 training rounds to find the candidate learning rates (with a less then 3% difference to the best result in test accuracy), called as coarse-grained search; and then run the algorithms for 4000 training rounds with the candidate learning rates to find the best learning rate (with the highest test accuracy), called as fine-grained search. The max norm of gradient clipping is set to 10 for PFL and 50 for SFL for all settings (see subsection G.2). Other hyperparameters are identical to that in the main body.

The results of the coarse-grained search are collected in Figure 12, Table 10. Taking the setting training VGG-9 on CIFAR-10 as an example. We first find the candidate learning rates, whose short-run test accuracies are only 3% or less below the best accuracy. The candidate learning rates are summarized in Table 9. The training curves are in Figure 11. We then find the best learning rate, whose long-run test accuracy is the highest among the candidate learning rates. The final best learning rates are in Table 9. For fine-grained search of other settings, please refer to the code.

Table 9: Best learning rates found by the fine-grained search. The candidate learning rates are collected in the cell and the correspond long-run test accuracies are in the parentheses. According to the long-run test accuracies, we keep the best learning rate and cross out the others.

| Settings | PFL | SFL |
|---|---|---|
| CIFAR-10, VGG-9, $C = 1$, $K = 5$ | 0.1 | 0.03 |
| CIFAR-10, VGG-9, $C = 1$, $K = 20$ | 0.1 | 0.003 |
| CIFAR-10, VGG-9, $C = 1$, $K = 50$ | ~~0.03 (28.72)~~, 0.1 (46.21) | 0.003 |
| CIFAR-10, VGG-9, $C = 2$, $K = 5$ | 0.1 | ~~0.01 (82.05)~~, 0.03 (82.18) |
| CIFAR-10, VGG-9, $C = 2$, $K = 20$ | 0.1 | 0.01 (81.50), ~~0.03 (78.35)~~ |
| CIFAR-10, VGG-9, $C = 2$, $K = 50$ | ~~0.03 (76.14)~~, 0.1 (78.13) | 0.01 |

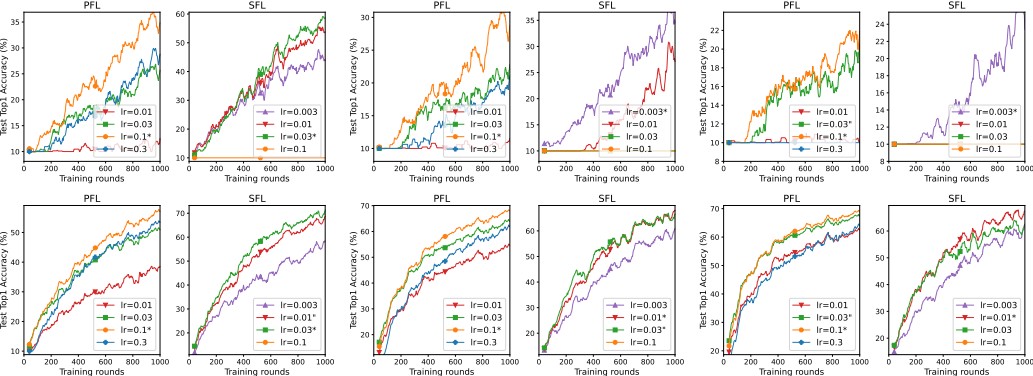

Figure 11: The corresponding training curves of Table 9. We mark the best learning rate in the short run with "*" and other candidate learning rates with """ in the legend. The top row shows the first three settings and the bottom row shows the last three settings in Table 9.

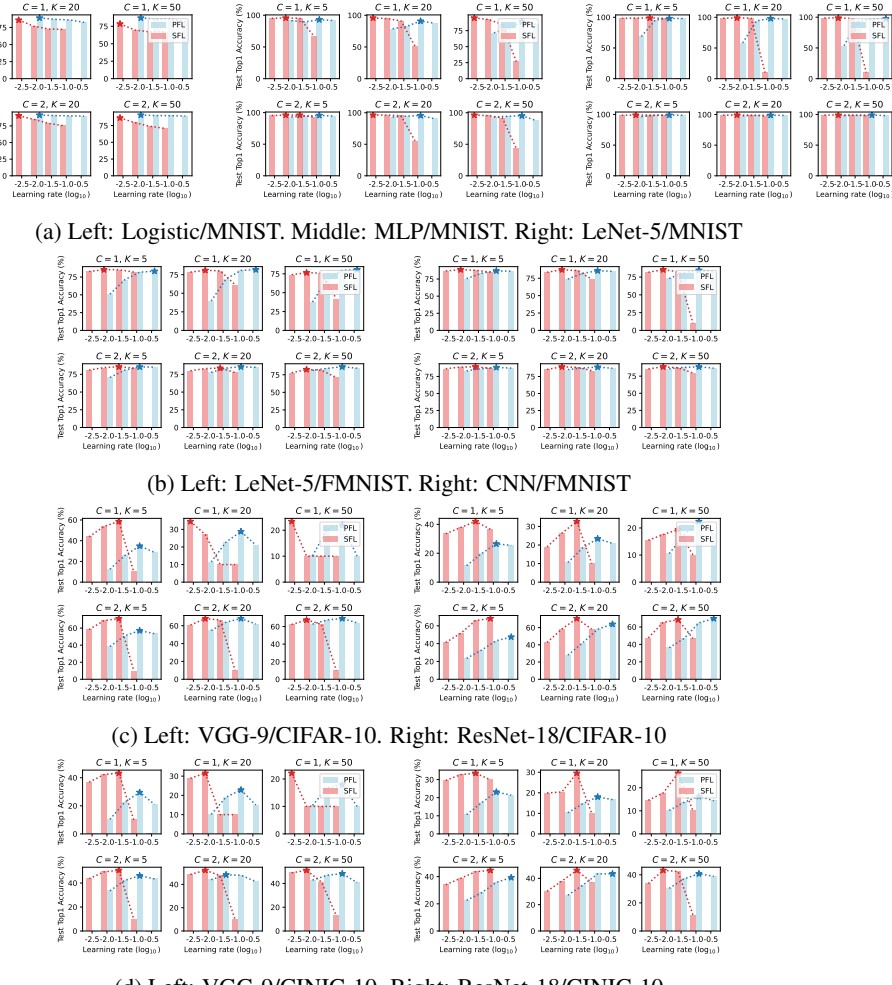

(a) Left: Logistic/MNIST. Middle: MLP/MNIST. Right: LeNet-5/MNIST

(b) Left: LeNet-5/FMNIST. Right: CNN/FMNIST

(c) Left: VGG-9/CIFAR-10. Right: ResNet-18/CIFAR-10

(d) Left: VGG-9/CINIC-10. Right: ResNet-18/CINIC-10

Figure 12: Test accuracies after training for 1000 rounds for various settings. Details are in Table 10.

## G.4 More experimental results

More experimental results are provided in this subsection. These include results on MNIST (LeCun et al., 1998), FMNIST (Xiao et al., 2017) and additional results on CIFAR-10 and CINIC-10.

*Setup on MNIST and FMNIST.* We consider five additional tasks: 1) training Logistic Regression on MNIST , 2) training Multi-Layer Perceptron (MLP) on MNIST, 3) training LeNet-5 (LeCun et al., 1998) on MNIST, 4) training LeNet-5 on FMNIST, 5) training CNN on FMNIST. We partition the training sets of both MNIST and FMNIST into 500 clients by extended Dirichlet strategy $C = 1$ and $C = 2$ (with $\alpha = 10.0$) and spare the test sets for computing test accuracy. We apply gradient clipping with the max norm of 10 for PFL and 50 for SFL. We find the best learning rate by grid search. This is done by running algorithms for 1000 training rounds and choosing the learning rate that achieves the highest test accuracy averaged over the last 40 training rounds. Note that since tasks on MNIST/FMNIST are quite simpler than that on CIFAR-10/CINIC-10, we do not use the coarse, fine-grained search. The results of grid search are given in Table 10. Other hyperparameters without being stated explicitly are identical to that of CIFAR-10/CINIC-10 in the main body.

*Results of MNIST and FMNIST.* The results of these five tasks are in Figures 13, 14 and Table 11. In the tasks MNIST/MLP, MNIST/LeNet-5, FMNIST/CNN, the performance of SFL is better when $C = 1$, which is consistent with the analysis in Subsection 4.2. However, we note that SFL shows worse even when $C = 1$ in simpler tasks MNIST/Logistic and FMNIST/LeNet-5, especially when $K$ is large. This may be because the (objective function) heterogeneity on simple models and datasets is limited even with extreme data distributions (i.e., $C = 1$). Thus, more extensive experiments are still required before drawing conclusions, which is beyond the scope of this paper.

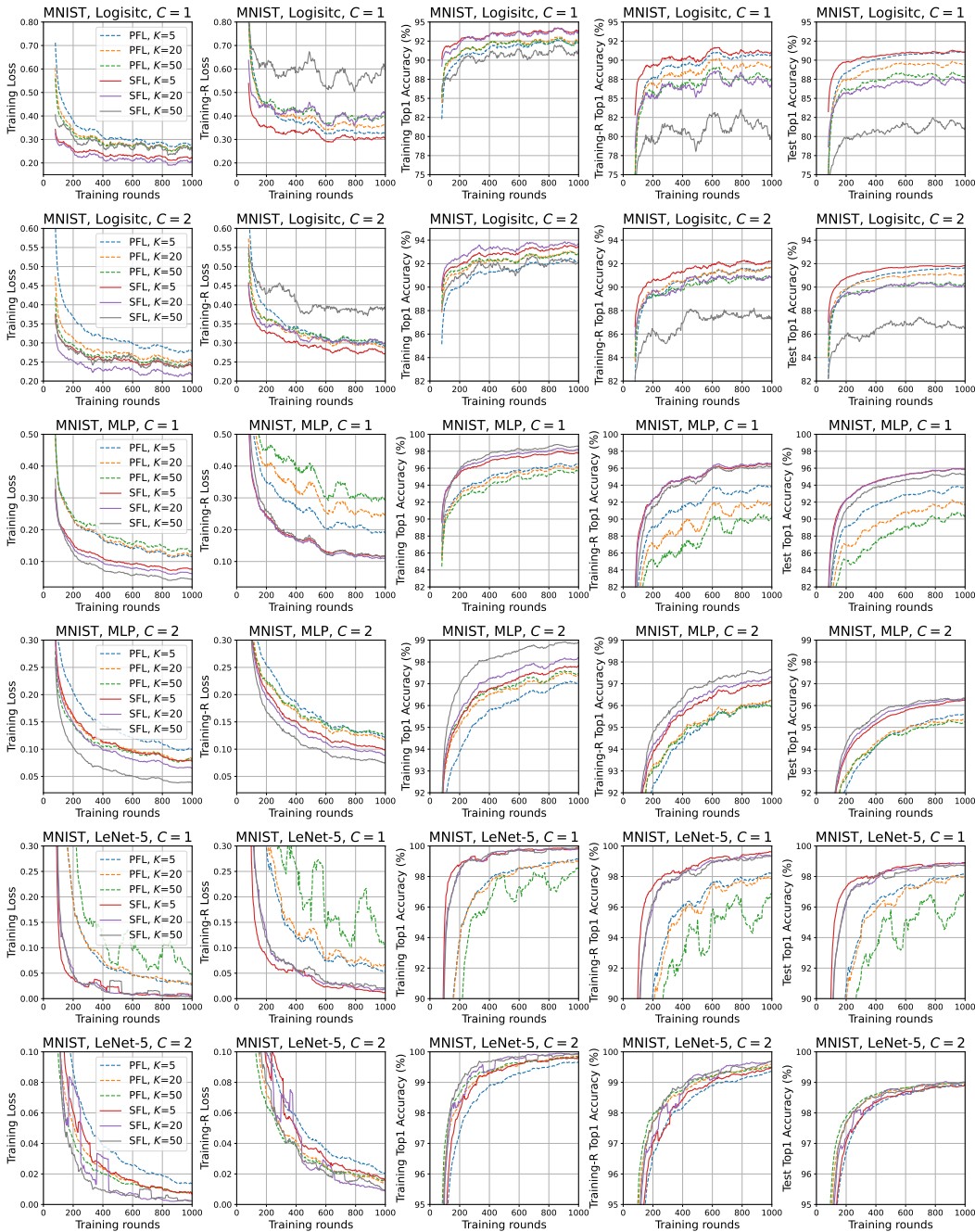

Figure 13: Experimental results on MNIST. For the best viewing experience, we apply moving average over a window length of 8% of the data points. Note that "Traning Loss/Accuracy" are computed over the training data of participating clients, "Training-R Loss/Accuracy" are computed over the training data of random clients and "Test Accuracy" are computed over the original test set.

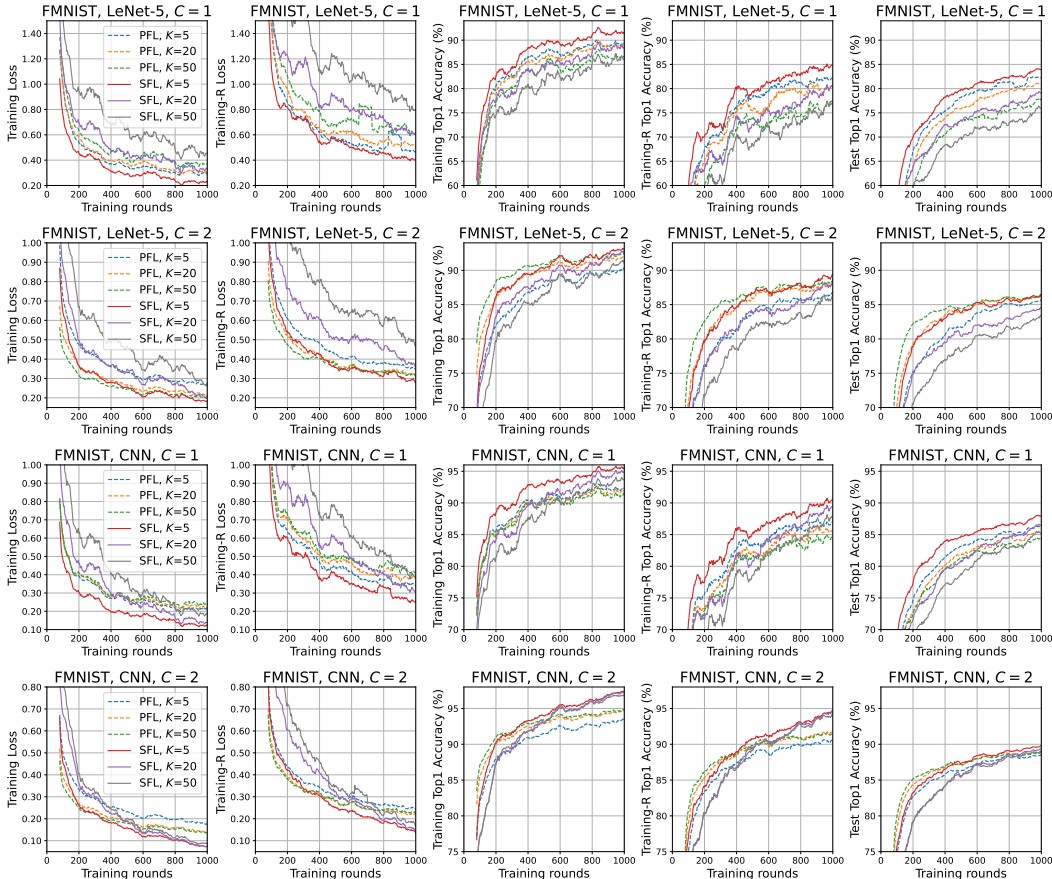

Figure 14: Experimental results on FMNIST. For the best viewing experience, we apply moving average over a window length of 8% of the data points. Note that "Traning Loss/Accuracy" are computed over the training data of participating clients, "Training-R Loss/Accuracy" are computed over the training data of random clients and "Test Accuracy" are computed over the original test set.

Table 10: Test accuracy results of grid searches for various settings. The results are computed over the last 40 training rounds (with 1000 training rounds in total). The highest test accuracy among different learning rates is marked in cyan for both algorithms.

| Setting | PFL | | | | SFL | | | |
|---|---|---|---|---|---|---|---|---|
| | $10^{-2.0}$ | $10^{-1.5}$ | $10^{-1.0}$ | $10^{-0.5}$ | $10^{-2.5}$ | $10^{-2.0}$ | $10^{-1.5}$ | $10^{-1.0}$ |
| MNIST, Logistic, $C=1$, $K=5$ | 90.95 | 89.81 | 87.78 | 84.33 | 90.68 | 87.05 | 81.38 | 78.27 |
| MNIST, Logistic, $C=1$, $K=20$ | 88.98 | 87.00 | 86.06 | 82.49 | 85.87 | 76.88 | 72.82 | 72.14 |
| MNIST, Logistic, $C=1$, $K=50$ | 86.99 | 85.50 | 85.52 | 81.77 | 78.45 | 69.67 | 67.00 | 67.09 |
| MNIST, Logistic, $C=2$, $K=5$ | 91.72 | 91.48 | 90.70 | 89.84 | 91.97 | 90.79 | 86.68 | 82.43 |
| MNIST, Logistic, $C=2$, $K=20$ | 91.31 | 90.54 | 90.10 | 89.66 | 90.29 | 84.86 | 78.94 | 75.18 |
| MNIST, Logistic, $C=2$, $K=50$ | 90.69 | 89.89 | 89.78 | 89.20 | 86.45 | 79.24 | 73.89 | 70.85 |
| MNIST, MLP, $C=1$, $K=5$ | 91.47 | 89.51 | 93.06 | 91.68 | 95.13 | 95.74 | 95.22 | 67.44 |
| MNIST, MLP, $C=1$, $K=20$ | 78.70 | 82.53 | 90.89 | 87.46 | 95.79 | 94.79 | 91.24 | 51.06 |
| MNIST, MLP, $C=1$, $K=50$ | 71.61 | 78.20 | 88.73 | 85.52 | 94.83 | 92.46 | 84.62 | 26.71 |
| MNIST, MLP, $C=2$, $K=5$ | 93.32 | 94.35 | 95.66 | 94.58 | 95.40 | 96.14 | 96.14 | 93.00 |
| MNIST, MLP, $C=2$, $K=20$ | 93.18 | 94.59 | 95.62 | 90.42 | 96.33 | 96.25 | 95.24 | 55.47 |
| MNIST, MLP, $C=2$, $K=50$ | 93.40 | 94.40 | 95.28 | 87.78 | 96.34 | 95.36 | 91.99 | 43.36 |
| MNIST, LeNet-5, $C=1$, $K=5$ | 68.72 | 95.46 | 98.07 | 98.05 | 98.28 | 98.76 | 98.91 | 98.29 |
| MNIST, LeNet-5, $C=1$, $K=20$ | 58.20 | 94.30 | 98.02 | 97.41 | 98.43 | 98.94 | 98.68 | 9.84 |
| MNIST, LeNet-5, $C=1$, $K=50$ | 53.61 | 93.00 | 97.23 | 97.49 | 98.60 | 98.77 | 97.98 | 9.95 |
| MNIST, LeNet-5, $C=2$, $K=5$ | 96.40 | 98.43 | 98.90 | 98.67 | 98.62 | 98.95 | 98.91 | 98.55 |
| MNIST, LeNet-5, $C=2$, $K=20$ | 97.88 | 98.69 | 98.86 | 98.50 | 98.94 | 99.17 | 98.96 | 97.74 |
| MNIST, LeNet-5, $C=2$, $K=50$ | 98.07 | 98.65 | 98.93 | 98.12 | 98.95 | 98.99 | 98.85 | 97.88 |
| FMNIST, LeNet-5, $C=1$, $K=5$ | 50.88 | 71.24 | 81.83 | 83.26 | 82.76 | 85.34 | 85.31 | 82.02 |
| FMNIST, LeNet-5, $C=1$, $K=20$ | 39.64 | 69.26 | 80.73 | 81.74 | 78.44 | 80.84 | 80.24 | 61.50 |
| FMNIST, LeNet-5, $C=1$, $K=50$ | 37.56 | 68.40 | 79.73 | 80.53 | 73.84 | 76.51 | 75.95 | 42.07 |
| FMNIST, LeNet-5, $C=2$, $K=5$ | 70.59 | 81.00 | 85.44 | 85.16 | 80.94 | 84.51 | 85.59 | 83.83 |
| FMNIST, LeNet-5, $C=2$, $K=20$ | 77.85 | 84.00 | 85.70 | 85.00 | 80.09 | 83.20 | 83.88 | 77.65 |
| FMNIST, LeNet-5, $C=2$, $K=50$ | 81.24 | 84.17 | 86.33 | 84.36 | 77.73 | 82.53 | 81.39 | 71.13 |
| FMNIST, CNN, $C=1$, $K=5$ | 75.57 | 83.44 | 86.98 | 86.40 | 86.83 | 88.66 | 87.82 | 85.01 |
| FMNIST, CNN, $C=1$, $K=20$ | 73.84 | 82.17 | 85.86 | 85.11 | 83.62 | 87.37 | 86.14 | 73.85 |
| FMNIST, CNN, $C=1$, $K=50$ | 72.94 | 81.26 | 84.67 | 82.41 | 81.75 | 85.39 | 82.42 | 10.00 |
| FMNIST, CNN, $C=2$, $K=5$ | 83.20 | 86.58 | 88.26 | 87.24 | 86.41 | 88.71 | 89.19 | 87.06 |
| FMNIST, CNN, $C=2$, $K=20$ | 85.12 | 87.46 | 88.80 | 87.08 | 85.39 | 89.24 | 88.48 | 83.65 |
| FMNIST, CNN, $C=2$, $K=50$ | 85.93 | 87.82 | 88.61 | 86.64 | 85.53 | 88.63 | 87.12 | 79.50 |
| CIFAR-10, VGG-9, $C=1$, $K=5$ | 12.51 | 25.67 | 34.89 | 28.77 | 43.79 | 53.73 | 58.63 | 10.00 |
| CIFAR-10, VGG-9, $C=1$, $K=20$ | 11.51 | 22.81 | 28.79 | 21.10 | 34.55 | 27.11 | 10.00 | 10.00 |
| CIFAR-10, VGG-9, $C=1$, $K=50$ | 10.14 | 20.58 | 21.95 | 10.00 | 23.41 | 10.00 | 10.00 | 10.00 |
| CIFAR-10, VGG-9, $C=2$, $K=5$ | 38.50 | 50.99 | 57.09 | 53.46 | 58.35 | 68.75 | 71.24 | 10.00 |
| CIFAR-10, VGG-9, $C=2$, $K=20$ | 54.70 | 63.97 | 68.07 | 61.92 | 60.82 | 67.93 | 66.50 | 10.00 |
| CIFAR-10, VGG-9, $C=2$, $K=50$ | 62.72 | 67.57 | 69.11 | 64.21 | 62.27 | 67.52 | 62.59 | 10.00 |
| CIFAR-10, ResNet-18, $C=1$, $K=5$ | 11.46 | 19.72 | 26.50 | 25.45 | 33.49 | 37.77 | 42.04 | 36.59 |
| CIFAR-10, ResNet-18, $C=1$, $K=20$ | 10.56 | 18.48 | 23.41 | 20.86 | 18.99 | 26.41 | 32.70 | 10.00 |
| CIFAR-10, ResNet-18, $C=1$, $K=50$ | 10.55 | 17.39 | 22.42 | 17.86 | 15.46 | 17.57 | 19.39 | 10.00 |
| CIFAR-10, ResNet-18, $C=2$, $K=5$ | 23.45 | 32.71 | 43.24 | 47.45 | 41.24 | 51.51 | 65.86 | 68.17 |
| CIFAR-10, ResNet-18, $C=2$, $K=20$ | 28.02 | 42.36 | 58.07 | 64.03 | 42.83 | 58.43 | 70.55 | 58.29 |
| CIFAR-10, ResNet-18, $C=2$, $K=50$ | 36.22 | 46.69 | 64.75 | 69.71 | 47.57 | 65.33 | 68.40 | 46.80 |
| CINIC-10, VGG-9, $C=1$, $K=5$ | 10.36 | 22.56 | 29.58 | 21.02 | 36.87 | 42.26 | 43.27 | 10.00 |
| CINIC-10, VGG-9, $C=1$, $K=20$ | 10.13 | 19.09 | 22.83 | 15.00 | 28.88 | 31.54 | 10.00 | 10.00 |
| CINIC-10, VGG-9, $C=1$, $K=50$ | 10.37 | 16.85 | 17.92 | 10.00 | 22.12 | 10.00 | 10.00 | 10.00 |
| CINIC-10, VGG-9, $C=2$, $K=5$ | 33.52 | 42.24 | 46.12 | 43.43 | 43.64 | 49.28 | 50.64 | 10.00 |
| CINIC-10, VGG-9, $C=2$, $K=20$ | 44.04 | 48.26 | 47.80 | 42.40 | 48.54 | 52.10 | 47.55 | 10.00 |
| CINIC-10, VGG-9, $C=2$, $K=50$ | 42.87 | 46.93 | 48.29 | 40.75 | 48.93 | 51.11 | 41.19 | 13.04 |
| CINIC-10, ResNet-18, $C=1$, $K=5$ | 10.64 | 17.16 | 23.15 | 21.37 | 29.52 | 32.70 | 33.55 | 29.92 |
| CINIC-10, ResNet-18, $C=1$, $K=20$ | 10.29 | 14.65 | 18.01 | 16.68 | 19.91 | 20.42 | 29.57 | 10.00 |
| CINIC-10, ResNet-18, $C=1$, $K=50$ | 10.18 | 13.65 | 16.27 | 14.35 | 14.56 | 17.72 | 26.30 | 10.00 |
| CINIC-10, ResNet-18, $C=2$, $K=5$ | 22.59 | 28.03 | 35.78 | 39.35 | 34.06 | 38.73 | 43.57 | 44.67 |
| CINIC-10, ResNet-18, $C=2$, $K=20$ | 27.18 | 34.74 | 43.40 | 43.46 | 30.27 | 37.96 | 46.13 | 36.81 |
| CINIC-10, ResNet-18, $C=2$, $K=50$ | 30.36 | 37.05 | 40.74 | 38.90 | 33.77 | 43.20 | 42.32 | 11.35 |

Table 11: Test accuracy results for various settings. We run PFL and SFL for 1000 training rounds for MNIST and FMNIST and 4000 training rounds for CIFAR-10 and CINIC-10 with 3 different random seeds. The results are computed over the random seeds and the last 40 training rounds for MNIST and FMNIST and the last 100 training rounds for CIFAR-10 and CINIC-10. The better results (with more than 1% advantage for MNIST/FMNIST and 2% advantage for CIFAR-10/CINIC-10) between PFL and SFL in each setting are marked in color.

| Setup | | | $C = 1$ | | | $C = 2$ | | |
|---|---|---|---|---|---|---|---|---|
| Dataset | Model | Method | $K = 5$ | $K = 20$ | $K = 50$ | $K = 5$ | $K = 20$ | $K = 50$ |
| MNIST | Logistic | PFL | $91.10_{\pm0.35}$ | $89.46_{\pm1.20}$ | $87.82_{\pm1.98}$ | $91.69_{\pm0.17}$ | $91.19_{\pm0.47}$ | $90.46_{\pm0.86}$ |
| | | SFL | $91.09_{\pm0.67}$ | $87.11_{\pm2.09}$ | $80.94_{\pm3.70}$ | $91.89_{\pm0.32}$ | $90.16_{\pm1.10}$ | $86.52_{\pm2.41}$ |
| | MLP | PFL | $93.61_{\pm1.42}$ | $91.84_{\pm2.20}$ | $90.27_{\pm3.17}$ | $95.65_{\pm0.38}$ | $95.46_{\pm0.47}$ | $95.34_{\pm0.51}$ |
| | | SFL | $95.91_{\pm0.33}$ | $95.90_{\pm0.44}$ | $95.25_{\pm0.76}$ | $96.26_{\pm0.22}$ | $96.35_{\pm0.22}$ | $96.35_{\pm0.32}$ |
| | LeNet-5 | PFL | $98.21_{\pm0.40}$ | $98.02_{\pm0.63}$ | $97.21_{\pm1.55}$ | $98.94_{\pm0.09}$ | $98.97_{\pm0.10}$ | $98.98_{\pm0.11}$ |
| | | SFL | $98.90_{\pm0.18}$ | $98.87_{\pm0.19}$ | $98.79_{\pm0.19}$ | $98.91_{\pm0.11}$ | $99.07_{\pm0.12}$ | $98.99_{\pm0.10}$ |
| FMNIST | LeNet-5 | PFL | $82.57_{\pm2.03}$ | $81.09_{\pm3.19}$ | $78.22_{\pm4.38}$ | $85.86_{\pm0.87}$ | $86.35_{\pm1.12}$ | $86.58_{\pm0.88}$ |
| | | SFL | $83.97_{\pm2.42}$ | $79.39_{\pm2.59}$ | $76.21_{\pm2.95}$ | $86.52_{\pm1.67}$ | $84.69_{\pm2.26}$ | $83.58_{\pm2.55}$ |
| | CNN | PFL | $86.61_{\pm1.62}$ | $85.40_{\pm2.07}$ | $84.62_{\pm2.18}$ | $88.61_{\pm0.93}$ | $89.16_{\pm0.77}$ | $89.10_{\pm0.89}$ |
| | | SFL | $88.03_{\pm1.28}$ | $86.75_{\pm1.39}$ | $85.44_{\pm1.66}$ | $89.91_{\pm0.96}$ | $89.60_{\pm1.01}$ | $89.05_{\pm1.27}$ |
| CIFAR-10 | VGG-9 | PFL | $67.61_{\pm4.02}$ | $62.00_{\pm4.90}$ | $45.77_{\pm5.91}$ | $78.42_{\pm1.47}$ | $78.88_{\pm1.35}$ | $78.01_{\pm1.50}$ |
| | | SFL | $78.43_{\pm2.46}$ | $72.61_{\pm3.27}$ | $68.86_{\pm4.19}$ | $82.56_{\pm1.68}$ | $82.18_{\pm1.97}$ | $79.67_{\pm2.30}$ |
| | ResNet-18 | PFL | $52.12_{\pm6.09}$ | $44.58_{\pm4.79}$ | $34.29_{\pm4.99}$ | $80.27_{\pm1.52}$ | $82.27_{\pm1.55}$ | $79.88_{\pm2.18}$ |
| | | SFL | $83.44_{\pm1.83}$ | $76.97_{\pm4.82}$ | $68.91_{\pm4.29}$ | $87.16_{\pm1.34}$ | $84.90_{\pm3.53}$ | $79.38_{\pm4.49}$ |
| CINIC-10 | VGG-9 | PFL | $52.61_{\pm3.19}$ | $45.98_{\pm4.29}$ | $34.08_{\pm4.77}$ | $55.84_{\pm0.55}$ | $53.41_{\pm0.62}$ | $52.04_{\pm0.79}$ |
| | | SFL | $59.11_{\pm0.74}$ | $58.71_{\pm0.98}$ | $56.67_{\pm1.18}$ | $60.82_{\pm0.61}$ | $59.78_{\pm0.79}$ | $56.87_{\pm1.42}$ |
| | ResNet-18 | PFL | $41.12_{\pm4.28}$ | $33.19_{\pm4.73}$ | $24.71_{\pm4.89}$ | $57.70_{\pm1.04}$ | $55.59_{\pm1.32}$ | $46.99_{\pm1.73}$ |
| | | SFL | $60.36_{\pm1.37}$ | $51.84_{\pm2.15}$ | $44.95_{\pm2.97}$ | $64.17_{\pm1.06}$ | $58.05_{\pm2.54}$ | $56.28_{\pm2.32}$ |

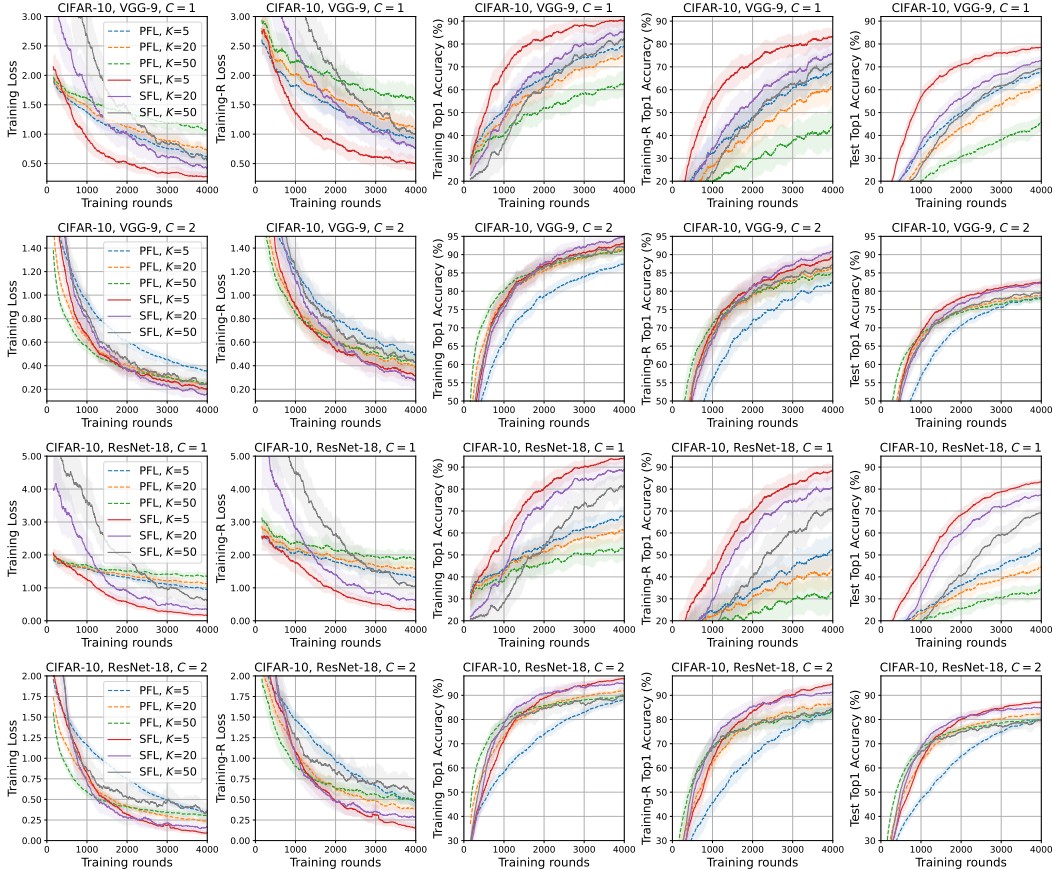

Figure 15: Experimental results on CIFAR-10. For the best viewing experience, we apply moving average over a window length of 4% of the data points. Note that "Traning Loss/Accuracy" are computed over the training data of participating clients, "Training-R Loss/Accuracy" are computed over the training data of random clients and "Test Accuracy" are computed over the original test set. The shaded areas show the standard deviation with 3 random seeds.

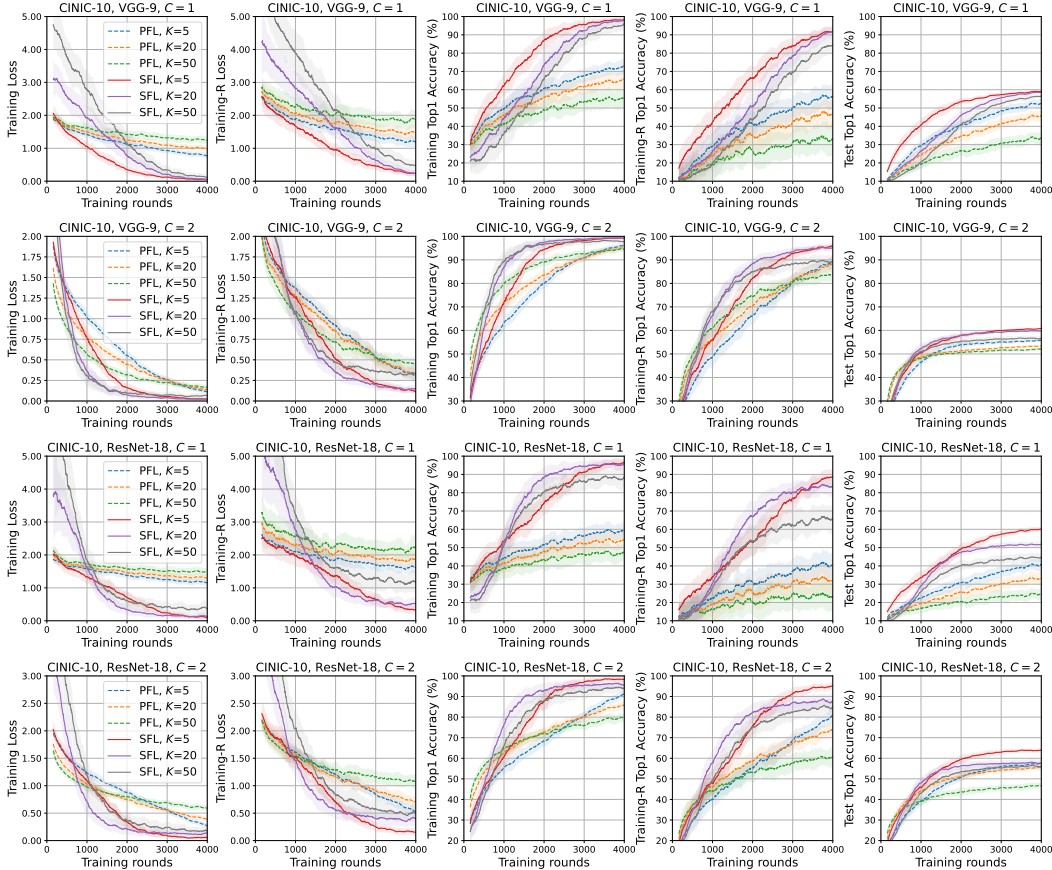

Figure 16: Experimental results on CINIC-10. For the best viewing experience, we apply moving average over a window length of 4% of the data points. Note that "Traning Loss/Accuracy" are computed over the training data of participating clients, "Training-R Loss/Accuracy" are computed over the training data of random clients and "Test Accuracy" are computed over the original test set. The shaded areas show the standard deviation with 3 random seeds.

