# OpenReview forum: "Convergence Analysis of Sequential Federated Learning on Heterogeneous Data"
_NeurIPS.cc/2023/Conference — NeurIPS 2023 poster_

### Official Review · Reviewer_K7U9 · 2023-07-04

**Soundness:** 3 good
**Presentation:** 3 good
**Contribution:** 2 fair
**Rating:** 6
**Confidence:** 3

**Summary:**

This paper presents an intriguing exploration of the counterintuitive comparison results between Sequential SL (SSL) and Federated Averaging (FedAvg) in the context of heterogeneous data. The authors attribute the surprising performance of SSL over FedAvg in worst-case scenarios to the assumption of data heterogeneity (Assumption 3), which is derived from the convergence theory of SGD. The paper provides a compelling argument and empirical evidence for the counterintuitive performance of SSL over FedAvg in certain scenarios. It offers valuable insights into the role of data heterogeneity in the performance of these algorithms and highlights the need for careful consideration of data heterogeneity in the design and analysis of federated learning algorithms. However, the paper could benefit from a more detailed discussion of the implications of these findings for real-world applications. Further exploration of the conditions under which SSL outperforms FedAvg, as well as potential strategies for mitigating the limitations of FedAvg in extremely heterogeneous settings, would be valuable.

**Strengths:**

1. The paper provides a comprehensive theoretical analysis of the counterintuitive performance of Sequantial SL (SSL) over Federated Averaging (FedAvg) in certain scenarios. The author's exploration of the role of data heterogeneity and its impact on these algorithms is particularly insightful.

2. The authors validate their theoretical findings with empirical experiments, which strengthens the credibility of their claims. The use of one-dimensional quadratic functions for simulation and the consideration of practical scenarios in resource-constrained edge devices add depth to their research.

**Weaknesses:**

1. The paper primarily focuses on the performance of Sequential SL(SSL) and Federated Averaging (FedAvg) in the context of data heterogeneity. While this focus is important, it might limit the generalizability of the findings. The performance of these algorithms could be influenced by other factors that are not considered in this paper.

2. While the paper does consider practical scenarios in resource-constrained edge devices, it could benefit from a more detailed discussion of the implications of its findings for real-world applications. Understanding how these findings could be applied in practice would add significant value to the paper.

3. The paper establishes that SSL outperforms FedAvg in extremely heterogeneous settings, but it doesn't dive deeply into why this is the case or how it could be mitigated. Further exploration of these areas would enhance the paper's contribution to the field.

**Questions:**

While the paper provides a detailed convergence analysis for the newly proposed SSL algorithm, my primary concern lies in the experimental evaluation. The authors have chosen to use relatively simple and dated datasets and models, such as MNIST and CIFAR10 with LeNet and VGG-11. These models, being almost a decade old, may not accurately reflect the complexities and challenges of current practical workloads.

To strengthen the paper, I recommend that the authors extend their experimental evaluation to include larger datasets and models. This would not only provide a more rigorous test of the SSL algorithm but also enhance the relevance and applicability of the research. If such larger experimental results can be provided, I would be inclined to increase my score by one point, as it would significantly improve the paper's contribution to the field.

**Limitations:**

Yes.

---

> ### Author Rebuttal · Authors · 2023-08-09
>
> This part contains 4 responses, denoted by R22-R25.
>
> ##### **R22: (Q1) Extend the experimental evaluation to include larger datasets and models.**
>
> As suggested, we have added ResNets (He et al., 2016) [Ref 7] for larger models and CINIC-10 (Darlow et al., 2020) [Ref 8] for larger datasets. The test accuracy results are given in [Tables 5, 6], and the corresponding training curve figures are given in the pdf in the global response.
>
> + *Experiments of ResNet-18 on CIFAR-10 and CINIC-10*. CINIC-10 consists of images from both CIFAR and ImageNet, which is larger than CIFAR-10. There are 9000 images in the training set of CINIC-10, which is partitioned into 1000 clients by Extended Dirichlet Strategy. We use ResNet-18 in (Jhunjhunwala et al., 2023) [Ref 9]. For the other settings, we keep the same as the paper. Besides, we also add the experiments with ResNet-18 on CIFAR-10. The results are in the [Table 5].
>
>   [Table 5: Test accuracies (%) of FedAvg and SSL with ResNet-18 on CIFAR10 and CINIC-10. We run both algorithms for 4000 training rounds and report the averaged test accuracies over the last 100 training rounds and 3 random seeds.]
>
>   | Dataset  | Method | ExDir(1, 10.0)  | ExDir(1, 10.0)  | ExDir(1, 10.0)  | ExDir(2, 10.0)  | ExDir(2, 10.0)  | ExDir(2, 10.0)  |
>   | -------- | ------ | --------------- | --------------- | --------------- | --------------- | --------------- | --------------- |
>   |          |        | $K=5$           | $K=20$          | $K=50$          | $K=5$           | $K=20$          | $K=50$          |
>   | CIFAR-10 | FedAvg | $53.15\pm 4.49$ | $43.79\pm 4.64$ | $33.53\pm 4.91$ | $80.08\pm 1.75$ | $82.40\pm 1.43$ | $79.98\pm 1.90$ |
>   | CIFAR-10 | SSL    | $82.84\pm 2.07$ | $76.93\pm 3.64$ | $68.90\pm 4.59$ | $87.33\pm 1.27$ | $85.21\pm 2.57$ | $78.54\pm 4.76$ |
>   | CINIC-10 | FedAvg | $40.44\pm 4.40$ | $33.63\pm 4.60$ | $24.59\pm 4.22$ | $58.61\pm 2.09$ | $56.00\pm 1.41$ | $47.66\pm 1.74$ |
>   | CINIC-10 | SSL    | $60.42\pm 1.81$ | $53.20\pm 1.78$ | $45.81\pm 3.02$ | $64.70\pm 1.36$ | $58.59\pm 2.15$ | $57.58\pm 1.90$ |
>
> + *Experiments for ResNets with different depths and widths*. To study the effect of the model depth and width, we also conduct experiments on ResNets with different depths and widths based on ResNet-20. ResNet-20 used here is the standard model for CIFAR-10 in (He et al., 2016), which is thinner than ResNet-18 used in Jhunjhunwala et al. (2023). From [Table 6], we have the following observations: (1) Both deeper and wider models can improve the performance of FedAvg and SSL. (2) FedAvg suffers from a much heavier degradation (i.e., $\Delta_{\text{Acc}}$) than SSL when data heterogeneity increases to the extreme scenario (from ExDir(2, 10.0) to ExDir(1, 10.0)), especially on large models.
>
>   [Table 6: Test accuracies (%) of FedAvg and SSL for ResNets with different depths and widths on CIFAR-10. We set $K = 5$ and run both algorithms for 4000 rounds and report the averaged test accuracies over the last 100 training rounds and 3 random seeds. WRN-20-$k$ ($k$ is the width) is the abbreviation of Wide ResNet (Zagoruyko et al, 2016) [Ref 10]. We use $\Delta_{Acc}$ to denote the difference of test accuracy between ExDir(2, 10.0) and ExDir(1, 10.0).]
>
>   | Partition      | Method | ResNet-20 | ResNet-32 | ResNet-44 | ResNet-56 | WRN-20-2 | WRN-20-4 | WRN-20-6 |
>   | -------------- | ------ | --------: | --------: | --------: | --------: | -------: | -------: | -------: |
>   | ExDir(1, 10.0) | FedAvg |   $49.16$ |   $50.31$ |   $49.74$ |   $50.46$ |  $53.58$ |  $55.13$ |  $54.93$ |
>   | ExDir(1, 10.0) | SSL    |   $69.57$ |   $71.23$ |   $71.87$ |   $70.73$ |  $76.71$ |  $81.05$ |  $82.84$ |
>   | ExDir(2, 10.0) | FedAvg |   $68.36$ |   $70.47$ |   $70.98$ |   $72.58$ |  $75.52$ |  $78.84$ |  $80.76$ |
>   | ExDir(2, 10.0) | SSL    |   $77.73$ |   $79.52$ |   $81.18$ |   $81.63$ |  $83.58$ |  $86.64$ |  $87.86$ |
>   | $\Delta_{Acc}$ | FedAvg |   $19.20$ |   $20.16$ |   $21.24$ |   $22.12$ |  $21.94$ |  $23.71$ |  $25.83$ |
>   | $\Delta_{Acc}$ | SSL    |    $8.16$ |    $8.29$ |    $9.31$ |   $10.90$ |  $12.90$ |   $5.59$ |   $5.02$ |
>
> ---
>
> ##### **R23: (W1) The generalizability might be limited.**
>
> Thanks for the insightful suggestions. The main contribution of this work is the convergence guarantee and other findings are served to support it. For this reason, we fix some settings and primarily study the performance comparison of FedAvg and SSL in the context of data heterogeneity. However, as pointed out, their performance can be influenced by other factors. We leave this for future work.
>
> ---
>
> ##### **R24: (W2) The implications of the paper's findings for real-world applications.**
>
> As suggested, we would add a more detailed discussion about the implications of the findings. Here we provide some implications as 1) Broadening the choices of algorithms, 2) Hyperparameter selection, and 3) Inspiring practical and theoretical advancements.
>
> ---
>
> ##### **R25: (W3) It doesn't dive deeply into why this is the case or how it could be mitigated.**
>
> + *About why this is the case*. In this work, (1) First, we derive the convergence guarantee of SSL, which is better than that of FedAvg. This means that SSL outperforms FedAvg at least in the worst case. (2) Then, we note that the upper bounds of FedAvg in existing work underestimate the performance of FedAvg. This means that FedAvg performs better than the upper bound in many practical settings. (3) For these two reasons, with the experiments, we find that SSL outperforms FedAvg in extremely heterogeneous settings.
>
> + *How it could be mitigated*. We agree that this is still one weakness of this work. At the same time, we think this could be our future work.

---

> > ### Comment · Area_Chair_L68e · 2023-08-19
> > **please acknowledge author response**
> >
> > dear reviewer
> >
> > as the discussion phase is ending very soon: could you please let us know if you read the author response and if the response or the discussions here change your assessment?
> >
> > thanks
> >
> > AC

---

> > ### Comment · Reviewer_K7U9 · 2023-08-19
> >
> > I thank the authors can provide experiments on larger datasets. After reading the comments and rebuttals in other reviewers, I decided to increase my score to 6.

---

### Official Review · Reviewer_98XT · 2023-07-05

**Soundness:** 4 excellent
**Presentation:** 3 good
**Contribution:** 2 fair
**Rating:** 5
**Confidence:** 3

**Summary:**

This paper studies the Sequential SL algorithm in which the at each round, sequentially, each machine performs some local SGD iterations, and then the final iterate is passed on to the next machine.  Such an algorithm may be useful in cross-device, resource constrained settings, where local iterations are expensive.The papers compares the performance of this algorithm to the performance of Federated Averaging, where the local SGD iterations are performed in parallel rather than sequentially. They study a heterogenous data setting, where the distribution of data is different at each machine, under strongly convex, convex, and non-convex settings. While the SL algorithm has larger parallel depth than federated averaging, the authors show that the SSL algorithm outperforms FedAvg in the strongly convex setting. They also show some minor improvements in the convergence of Federated Averaging by reanalyzing this popular algorithm. They provide experiments in a heterogenous environment to show that for high levels of hetergeneity, SSL outperforms FedAvg without contradicting prior experimental evidence.

**Strengths:**

- The authors provide a clean analysis of SSL under standard assumptions in strongly convex, convex, and non-convex settings. They also show how to slightly improve the analysis of FedAvg in a heterogeneous setting, which cleans up some small gaps between bounds in the literature.
- The SSL algorithm they analyze is simple and natural, yet is not immediate to understand with existing analyses.


**Weaknesses:**

*Clarity*:
- The two last paragraphs on page 1 are somewhat confusing. The 2nd to last P seems to span about FedAvg. But then the last paragraph seems to speak about but SSL and about FedAvg (which is called SFLV1, which is confusing). Could the authors streamline these paragraphs to make it clear the two algorithms considered, and also clarify if SFLV1 = FedAvg or how they are different.
- What does it mean that the model is trained partially at the server? I don't see this in Algorithm 1?
- I find the emphasis on "counterintuitive" to be off-putting, because it seems natural that SSL, which is a sequentially deeper algorithm will perform better.
- Could the authors make Assumption 3 clearer by breaking it into assumption 3a and 3b or break its into two assumptions environments.

*Comparison to other theoretical results is confusing or sometimes misleading*
- The discussion in the paragraph at line 154 is confusing. Perhaps a table would be helpful. Further, it is misleading that the authors say that their bounds are tight, because there seams to be a difference in depenence on M (which is quite significant), and anyways, the tightness would only be for the K = 1 case.
- In the comparison to FedAvg (line 189 paragraph), the authors only discuss the strongly convex setting. Could they please explain the other cases too. The following paragraph (line 193) is quite confusing - I have no idea what the authors are saying.
- In the discussion of heterogeneity and in the empirical experiments, the authors claim that SSL is better relative to FedAvg when there is more heterogeneity. But this does not completely make sense, because in the homogenous setting, where there is no heterogenity, SSL would be strictly better than FedAvg. Could the authors please discuss and clarify this.

**Questions:**

See weakness above.
Minor comment: Line 134 vanishes --> vanish

**Limitations:**

Yes

---

> ### Author Rebuttal · Authors · 2023-08-09
>
> This part contains 7 responses, denoted by R15-R21.
>
> ##### **R15: (W1) The two last paragraphs on page 1 are somewhat confusing.**
>
> As suggested, we will streamline these paragraphs in the revision. We would like to clarify that SFLV1 shares the same update rule with FedAvg despite SFLV1 $\neq$ FeAvg. Please refer to R1 (in the global response) for details.
>
> ---
>
> ##### **R16: (W2) What does it mean that the model is trained partially at the server?**
>
> Algorithm 1 is a simplified version of SSL, and the detailed version was deferred to Appendix D. Please refer to R1 for details.
>
> ---
>
> ##### **R17: (W3) It seems not counterintuitive that SSL performs better than FedAvg.**
>
> We would like to clarify that the derived result that SSL can outperform FedAvg is novel.
>
> SSL updates the global model more frequently with less accurate gradients (from every single client). In contrast, FedAvg updates the global model less frequently with more accurate gradients (by aggregating the gradients from multiple clients).
>
> Intuitively, one might say that 1) SSL can outperform FedAvg in the homogeneous setting since the gradients from each client are accurate, 2) while the performance of SSL would be worse than FedAvg in the heterogeneous settings since the aggregated gradients from multiple clients are more accurate than that from one single client. The state-of-the-art work  Gao et al. (2021) [Ref 4] arrives at the same conclusion as the aforementioned intuitive one by empirical experiments.
>
> However, in this work, we show the counterintuitive result that SSL can also outperform FedAvg in the "extremely" heterogeneous settings, and analyze the effect of data heterogeneity on the comparison.
>
> ---
>
> ##### **R18: (W4) Break Assumption 3 into two assumptions.**
>
> As suggested, we will break Assumption 3 into two assumptions in the revision.
>
> ---
>
> ##### **R19: (W5) The discussion in the paragraph at line 154 is confusing**
>
> [Table 3: The bounds of SSL and Random Reshuffling (RR). The bound of SSL when $K=1$ and $\sigma=0$ is shown in the first row; the bound of RR (Mishchenko et al., 2020) [Ref 5] in the second row; and the lower bound of RR (Safran and Shamir, 2020) [Ref 6] in the third row. We omit all the constants (including $\mu$ and $L$) for clarity.]
>
> | Algorithm | SSL                                                          | RR                                                           | RR (lower bound)                                        |
> | --| -- | -- | -- |
> | **Bound** | $\tilde{\mathcal{O}} \left(\frac{\zeta_\ast^2}{MR^2} + D^2 \exp(-R)\right)$ | $\tilde{\mathcal{O}} \left(\frac{\zeta_\ast^2}{MR^2} + D^2 \exp(- {\color{red}M}R)\right)$ | $\Omega\left(\min(1, \frac{\zeta_\ast^2}{MR^2})\right)$ |
>
> + *The tightness for the case $K=1$ for sufficiently large $R$.* The difference between the upper and lower bounds is the negative exponential term (smaller than the first term when $R$ is sufficiently large), i.e., the bound will match the lower bound. We will add the restriction in the revision.
>
> + *Difference on* $M$. As pointed out, RR has advantage on the negative exponential term. The advantage can be neglected for sufficiently large $R$.
>
> + *The tightness for the* $K=1$ *case*. We use the lower bound of RR to validate our tightness. As pointed out, the lower bound designed for SSL is still required, which is currently a theoretical limitation of our work. We will clarify the limitations in the revision.
>
> ---
>
> ##### **R20: (W6) Explain the other cases**
>
> [Table 4: Upper bounds in the general convex and non-convex cases with absolute constants omitted. We use the bounds of Theorem 2 for FedAvg to keep the same assumptions and setups. $D\coloneqq\||x^0-x^\ast\||^2$ and $A\coloneqq F(x^0)-F^\ast$. ]
>
> | Algorithm | General convex case                                          | Non-convex case                                              |
> | -- | -- | -- |
> | FedAvg    | $\displaystyle \frac{\sigma D}{\sqrt{MKR}}+ \frac{(L\sigma^2D^4)^{1/3}}{K^{1/3}R^{2/3}}+ \frac{(L\zeta_\ast^2D^4)^{1/3}}{R^{2/3}} + \frac{LD^2}{R}$ | $\displaystyle \frac{(\sigma^2LA)^{1/2}}{\sqrt{MKR}}+ \frac{(L^2\sigma^2A^2)^{1/3}}{K^{1/3}R^{2/3}}+ \frac{(L^2\zeta^2A^2)^{1/3}}{R^{2/3}} + \frac{LB^2A}{R}$ |
> | SSL       | $\displaystyle\frac{\sigma D}{\sqrt{MKR}}+ \frac{(L\sigma^2D^4)^{1/3}}{{\color{red}M^{1/3}}K^{1/3}R^{2/3}}+ \frac{(L\zeta_\ast^2D^4)^{1/3}}{{\color{red}M^{1/3}}R^{2/3}} + \frac{LD^2}{R}$ | $\displaystyle \frac{(\sigma^2LA)^{1/2}}{\sqrt{MKR}}+ \frac{(L^2\sigma^2A^2)^{1/3}}{{\color{red}M^{1/3}}K^{1/3}R^{2/3}}+ \frac{(L^2\zeta^2A^2)^{1/3}}{{\color{red}M^{1/3}}R^{2/3}} + \frac{LB^2A}{R}$ |
>
> (1) From [Table 4], we see that SSL shows the similar advantage over FedAvg in the general convex and non-convex cases, to the strongly convex case. (2) We will clarify the following paragraph (line 193) with more details.
>
> ---
>
> #### **R21: (W7)  Clarification and discussion for SSL's better performance.**
>
> We would like to clarify that SSL can outperform FedAvg in the homogeneous and "extremely" heterogeneous settings, but may underperform in the "moderately" heterogeneous setting.
>
> + We prove that SSL has better convergence bound than FedAvg (i.e., SSL can outperform FedAvg in the worst case). However, the bound for FedAvg may be loose, since the global aggregation of FedAvg can help improve the gradient accuracy to achieve better performance in practice.
>
> + In the homogeneous setting (the one-client gradient of SSL is already accurate) and "extremely" heterogeneous setting (the gradient aggregation cannot achieve accurate gradient), the gradient aggregation in FedAvg does not significantly contribute to the convergence. Nevertheless, in the "moderately" heterogeneous setting (the gradient aggregation does help obtain the accurate gradient), FedAvg can achieve much better performance than its bound and outperform SSL.

---

> > ### Comment · Reviewer_98XT · 2023-08-19
> >
> > I have read the reply of the authors, thank you!

---

### Official Review · Reviewer_D6Q7 · 2023-07-05

**Soundness:** 3 good
**Presentation:** 3 good
**Contribution:** 3 good
**Rating:** 7
**Confidence:** 4

**Summary:**

The authors provides the convergence analysis of a FL-like approach where clients updates the model sequentially based on local datasets. Results are provided for the strongly convex, convex, and non-convex case.  Interestingly, the analysis shows that in some setting the sequential approach can outperform the usual FL approach where multiple clients update the local model in parallel and a server aggregates the updates.
I do not understand why split learning is mentioned in the title and in the rest of the paper, as there is no model split across clients or across the clients and server as far as I can say.

**Strengths:**

* The convergence analysis is interesting; it shows the authors master a number of existing techniques used to prove convergence results of distributed training procedures
* the result about sequential training  outperforming FedAvg is very interesting and indeed counterintuitive as observed by the authors
* This was the best paper in my batch (6 papers)


**Weaknesses:**

* The reference to split computing, while this paper does not deal with split computing.
* The result about SSL outperforming FedAvg is interesting, but the paper does not provide any intuitive explanation about why this may be the case. Also the bounds suggest that it should always be the case in the strongly convex setting for any value of $\zeta^*$, not only in the highly heterogeneous setting (large $\zeta^*$).
* I do not find the comparison with FedAvg under partial clients' participation fair. The authors should consider that at each round the clients available will sequentially perform  SSL. Instead, they assume that all clients participate, but the reduce the total number of rounds.
* there are some papers about the convergence of sequential optimization which should be cited. I think most of them consider a single gradient update at each node and focus on random walks. Examples:
	* Ram, S., Nedić, A., and Veeravalli, V. (2009). Incre- mental stochastic subgradient algorithms for con- vex optimization. SIAM Journal on Optimization, 20(2):691–717.
	* Johansson, B., Rabi, M., and Johansson, M. (2009). A randomized incremental subgradient method for dis- tributed optimization in networked systems. SIAM Journal on Optimization, 20(3):1157–1170
	* Mao, X., Yuan, K., Hu, Y., Gu, Y., Sayed, A. H., and Yin, W. (2020). Walkman: A Communication- Efficient Random-Walk Algorithm for Decentralized Optimization. IEEE Transactions on Signal Process- ing, 68:2513–2528.
	* Ayache, G. and Rouayheb, S. E. (2020). Private Weighted Random Walk Stochastic Gradient De- scent. Technical report, arXiv:2009.01790.
	* Cyffers, E., Bellet, A., Privacy amplification by decentralization, AISTATS 2022
* There are some sentence that are poorly worded, some examples
	* line 72: "it cannot follow the theory of SGD", rather  "SGD theory does not apply"
	* line 120: "Or further, it only holds at the optimum.", it seems the authors mean. "In alternative, the condition is required only at a global minimizer"
	* line 126 and elsewhere: "making it hold that", simply "and it holds"?
	* line 188, what do the authors mean by "our comparison is persuasive"?
	* line 210, the section is not really about "feasability", it is just providing some numerical "evidence" tha the counterintuitive result can appear even for simple quadratic functions. As no intuitive explanation is provided, I think this section should be merged with the experiments one.
	* line 218, "which is tight to capture the gradient diversities", what the authors mean is that the bound may be pessimistic for fedavg where averaging can limitate the effect of statistical heterogeneity, but less for SSL because of its sequential nature.
* Others:
	* line 67, (Wang et al, 2020) is indicated as an example dealing with unbalanced data, but I would say it rather deals with unbalanced computation.
	* line 67 trival -> trivial
	* line 114 and elsewhere: using assumption (4) to indicate the equation (4) in assumption 3 may be confusing
	* There are some papers which are cited as preprint, but have been published. For example Haibo Yang, Minghong Fang, and Jia Liu. Achieving linear speedup with partial worker participa- tion in non-iid federated learning appeared at ICLR 2021


**Questions:**

* why do the authors refer to split computing? which model is split? Algorithm 1 shows well that all clients train the same model architecture.
* what is denoted by SFLv1? if it denotes simply the algorithm proposed in (Thapa et al., 2020) just say it. Also in the rest of the paper, including in the experiment section, SFL is never evaluated, but he proposed solution is compared with FedAvg.
* the paper suggests in different places that increasing the number of local iterations may have a beneficial effect at first and then be harmful, but in reality the bounds in the paper suggest that it is always beneficial to increase K but with a diminishing return, right? If so, some statements while commenting figure 2 should be modified.


**Limitations:**

Yes.

---

> ### Author Rebuttal · Authors · 2023-08-09
>
> This part contains 8 responses, denoted by R7-R14.
>
> ##### **R7: (W1) Does not deal with split computing.**
>
> Due to space limitation, please refer to R1 (the global response) for details.
>
> ---
>
> ##### **R8: (W2) No intuitive explanation about why this may be the case.**
>
> *The Intuition about why SSL performs better than FedAvg in the extremely heterogeneous settings*. Intuitively, FedAvg updates the global model less frequently with more accurate gradients (by aggregating the gradients from multiple clients). In contrast, SSL updates the global model more frequently with less accurate gradients (from every single client). In the extremely heterogeneous settings where even aggregated gradients from multiple clients in FedAvg may be still inaccurate, the benefits of frequent updates in SSL turn dominant, and thus SSL can outperform FedAvg.
>
> *The reason why SSL may not perform better than FedAvg in the moderately heterogeneous settings, even the upper bound of SSL is better than FedAvg.* First, we agree that the upper bound (or worst-case error) of SSL is better than FedAvg regardless of the value of $\zeta_\ast$, which suggests SSL performs better than FedAvg, at least in the worst case. However, the comparison result may not apply to the "normal" cases. The upper bound of SSL is tight considering its sequential training manner, while the upper bound of FedAvg may be too pessimistic (omitting the function of aggregation) in the "normal" cases (Wang et al., 2022) [Ref 2]. For this reason, the performance of FedAvg can be comparable to or even better than SSL in the moderately heterogeneous settings.
>
> ---
>
> ##### **R9: (W3) The fairness of the comparison with FedAvg under partial clients' participation.**
>
> As pointed out, the comparison result in the paper is based on the traditional training patterns of FedAvg and SSL, which may cause unfairness as SSL does not perform client sampling. As suggested, we provide the theoretical comparison results for FedAvg and SSL with partial participation. The sampling error is caused by sampling partial clients per training round. Sampling without replacement is considered here, and $S$ is the sampling size. We see that the improvement ($1/S$, marked in red) of SSL also exists, similar to the full participation setup.
> $$
> \text{FedAvg:}\quad
> \tilde{\mathcal{O}} \left(\frac{\sigma^2}{\mu SKR} + \underbrace{\frac{\zeta\_\ast^2}{\mu R}\frac{M-S}{S(M-1)}}\_{\text{sampling error}} + \frac{L\sigma^2}{\mu^2 KR^2} + \frac{L\zeta_\ast^2}{\mu^2R^2}+\mu D^2 \exp(-\frac{\mu}{12L}R) \right)
> $$
>
> $$
> \text{SSL:}\quad
> \tilde{\mathcal{O}} \left(\frac{\sigma^2}{\mu SKR} + \underbrace{\frac{\zeta_\ast^2}{\mu R}\frac{M-S}{S(M-1)}}\_{\text{sampling error}}+ \frac{L\sigma^2}{\mu^2 {\color{red}S}KR^2} + \frac{L\zeta_\ast^2}{\mu^2{\color{red}S}R^2}+\mu D^2 \exp(-\frac{\mu}{12L}R) \right)
> $$
>
> Experimental validation is also prepared. For convenience, we use the same experimental setups as our new-added experiments (please refer to R22 (in Reviewer K7U9) for results of FedAvg and detailed experimental setups). The experiments are given in [Table 2]. SSL with partial participation is denoted as SSL-II. It can be seen that SSL-II shows a close performance to SSL, with only a loss of less than 1.5 % in test accuracy in all setups.
>
> [Table 2: Test accuracies (%) of FedAvg and SSL with ResNet-18 on CIFAR-10 and CINIC-10. We run both algorithms for 4000 training rounds and report the averaged test accuracies over the last 100 training rounds and 3 random seeds.]
>
> | Dataset  | Method | ExDir(1, 10.0)  | ExDir(1, 10.0)  | ExDir(1, 10.0)  | ExDir(2, 10.0)  | ExDir(2, 10.0)  | ExDir(2, 10.0)  |
> | -------- | ------ | --------------- | --------------- | --------------- | --------------- | --------------- | --------------- |
> |          |        | $K=5$           | $K=20$          | $K=50$          | $K=5$           | $K=20$          | $K=50$          |
> | CIFAR-10 | SSL    | $82.84\pm 2.07$ | $76.93\pm 3.64$ | $68.90\pm 4.59$ | $87.33\pm 1.27$ | $85.21\pm 2.57$ | $78.54\pm 4.76$ |
> | CIFAR-10 | SSL-II | $82.68\pm 2.33$ | $76.58\pm 4.84$ | $67.73\pm 6.46$ | $87.29\pm 1.35$ | $84.72\pm 3.51$ | $78.84\pm 4.15$ |
> | CINIC-10 | SSL    | $60.42\pm 1.81$ | $53.20\pm 1.78$ | $45.81\pm 3.02$ | $64.70\pm 1.36$ | $58.59\pm 2.15$ | $57.58\pm 1.90$ |
> | CINIC-10 | SSL-II | $60.56\pm 1.49$ | $51.98\pm 2.02$ | $44.75\pm 3.46$ | $64.27\pm 1.12$ | $58.15\pm 1.92$ | $56.45\pm 2.29$ |
>
> ---
>
> ##### **R10: (W4) Some papers about the convergence of sequential optimization should be cited.**
>
> We will cite and discuss the recommended papers in the revision as suggested.
>
> ---
>
> ##### **R11: (W5) There are some sentence that are poorly worded.**
>
> + As suggested, we will improve our presentation of lines 67, 72, 114, 120, 126, 188, 218.
> + We will reorganize this section with Section 5 Experiments for line 210.
> + We will adjust our references.
>
> ---
>
> ##### **R12: (Q1) Why do the authors refer to split computing?**
>
> Due to space limitation, please refer to R1 (in the global response) for details.
>
> ---
>
> ##### **R13: (Q2) What is denoted by SFLv1? SFL is not evaluated.**
>
> SFLV1 (Thapa et al., 2020) [Ref 3] is one typical algorithm in SL, which shares the same update rule with FedAvg (Algorithm 2), and shows a close performance to FedAvg (Gao et al., 2021) [Ref 4]. For this reason, we do not evaluate it. Please refer to R1 for details.
>
> ---
>
> ##### **R14: (Q3) Some statements while commenting figure 2 should be modified.**
>
> As pointed out, increasing the value of $K$ can always cause a diminishing return according to our bound. The phenomenon that increasing the number of local iterations may have a beneficial effect at first and then be harmful is common in the literature, which may be due to overfitting. As suggested, we will improve the statements of Fig. 2.

---

> > ### Comment · Reviewer_D6Q7 · 2023-08-10
> >
> > Thanks for your reply. I appreciate the new fairer comparison between sequential training and FedAvg.
> >
> > I still do not understand why the authors decided to frame their contribution in the split computing framework, as nothing seems really specific to split computing. Ok, the model is split between the client and the server in Algorithm 4, but the model is updated exactly how it would be in the classic federated learning setting. Simply some operations are executed at the client and others at the server. Am I right?

---

> > > ### Author Response · Authors · 2023-08-11
> > >
> > > ##### **R26: Nothing seems really specific to split computing.**
> > >
> > > Yes. As pointed out, the split computing in SSL only partitions the operations to be executed at clients and the server collaboratively. Our convergence results are also applicable to the classic distributed machine learning setting (where each client keeps one full model). To highlight, the main feature of our study is the convergence of sequential training manner (not the typical parallel training manner, i.e., Federated Learning) in distributed machine learning (not limited to the split learning).
> > >
> > > Thanks for the constructive comment. We understand that the comparison between sequential training and parallel training manners in distributed machine learning would be more general and may increase the impact of this paper. We will clarify the applicability of our convergence results in the revision.
> > >
> > > The reasons that we focus on the setting of sequential split learning (SSL) may include:
> > >
> > > + Sequential training manner is popular and widely adopted in SL (i.e., SSL considered in the original paper of SL (Gupta 29 and Raskar, 2018) [Ref 11]). SL is also the recent research hotspot. The convergence analysis of SSL is also of great practical and theoretical value.
> > > + The experimental settings (e.g., the "cross-device" setting and the "extremely heterogeneous" data partitioning) are more natural and practical in SL (designed for resource-constrained edge devices).
> > >
> > > [Ref 11] Gupta, O., & Raskar, R. (2018). Distributed learning of deep neural network over multiple agents. *Journal of Network and Computer Applications*, *116*, 1-8.

---

> > > > ### Comment · Reviewer_D6Q7 · 2023-08-19
> > > >
> > > > Thanks for the answer. Yes, I would really advise the authors to frame the whole paper in terms of sequential learning.  I do not know how much it is possible at this stage (e.g., if it is possible to change the title).
> > > > In any case, I think this is a good paper.

---

> > > > > ### Author Response · Authors · 2023-08-19
> > > > >
> > > > > We would like to express our sincere gratitude for approving the quality of our paper. We will carefully consider your advice in the revision.

---

### Official Review · Reviewer_4Dha · 2023-07-06

**Soundness:** 3 good
**Presentation:** 3 good
**Contribution:** 3 good
**Rating:** 6
**Confidence:** 4

**Summary:**

This paper provides the first theoretical convergence rates for sequential split learning (SSL) in the presence of heterogeneous data. In each round of SSL, a random permutation of the machines is computed. According to this permutation, each machine performs $K$ local gradient steps on the model sequentially. When the data on the clients is non-iid, due to the sequential nature of the steps, the updates are biased gradients of the true objective, which is the average of losses on all machines. This is the main challenge in analyzing SSL.

The authors extend the proof of FedAvg to provide tight convergence rates for SSL on non-convex, convex and strongly-convex loss functions under standard assumptions of smoothness, bounded gradient noise and heterogeneity between machines.


They show that SSL has strictly better convergence rates than FedAvg. In particular, the heterogeneity term in the rates decreases by a
factor of $\frac{1}{M}$, where $M$ is the number of machines.


They demonstrate the improvement of SSL over FedAvg empirically for quadratics under high heterogeneity. Additional experiments on CIFAR10 under high heterogeneity support this observation.


**Strengths:**

- **Convergence rates** : Clean convergence results are provided under the same assumptions as FedAvg. For local steps, $K=1$, SSL is equivalent to Random Reshuffling, and the obtained convergence rates match its lower bound.
- **Thorough comparison to FedAvg** : The authors provide thorough comparisons to the best possible rates for FedAvg under full participation and partial participation and demonstrate the improvement due to SSL.

- **Experiments validate theory**: Synthetic experiments show a clear improvement of SSL over FedAvg in high heterogeneity conditions.
- **Detailed Appendix** : There are several additional experiments in the appendix on more quadratic problems, MNIST and FMNIST, under varying hyperparameters. It might be beneficial to move one of these experiments to the main paper.
- **Marginally better rates for FedAvg on strongly convex functions**: The authors obtain marginally better rates for FedAvg in terms of constants by combining two existing proofs.


**Weaknesses:**

- **Analysis of Split learning without any explicit splits** :  In SSL, or any form of split learning (Gupta and Raskar, 2018), a crucial component of the algorithm is the model being split into two parts, one for the server and one for the client. The analysis considers sequential local updates only without any form of splits in the model. This could be clarified in the introduction.
- **Minor improvement in CIFAR10 figures**: This is a minor weakness, but Fig 2 is difficult to read. Using different color schemes for FedAvg and SSL or showing only 2 values of local steps might improve clarity.
- **SSL as a case of FedAvg with partial participation** : Note that if we perform FedAvg with $S=1$ worker per round sampled without replacement for $R$ rounds, then this algorithm is same as running SSL for $R/M$ rounds. Could the analysis of FedAvg with partial participation then recover the same rates as this paper?

**References**
- (Gupta and Raskar, 2018) : Distributed learning of deep neural network over multiple agents. Journal of Network and Computer Applications.


**Questions:**

- A recent paper (Yun et al, 2022) considers Minibatch and Local SGD with Random Reshuffling. It would be interesting to compare its theoretical rates with SSL, which has close connections to Random Reshuffling.
- Does the improvement in the rates for SSL over FedAvg come from a better bound on the client drift?

**References**
- (Yun et al, 2022) : Minibatch vs Local SGD with Shuffling : Tight Convergence Bounds and Beyond. ICLR 2022.


**Limitations:**

Described in Weaknesses

---

> ### Author Rebuttal · Authors · 2023-08-09
>
> This part contains 5 responses, denoted by R2-R6.
>
> ##### **R2: (W1) Analysis of Split learning without any explicit splits.**
>
> As suggested, we would like to clarify that the operation details (e.g., the form of model split) would not affect the convergence analyses of SSL. To this end, we stated the simplified version of SSL in Algorithm 1 in the Introduction and deferred the details to Appendix D. Please refer to R1 (in the global response) for details.
>
> ---
>
> ##### **R3: (W2) Minor improvement in CIFAR10 figures.**
>
> As suggested, we will improve Fig. 2 for better clarity.
>
> ---
>
> ##### **R4: (W3) SSL as a case of FedAvg with partial participation.**
>
> With the existing assumptions and techniques in this paper, SSL would exhibit better convergence bound than FedAvg with partial participation. Let us concentrate on the terms containing $\zeta_\ast^2$ regardless of the terms containing $\sigma^2$ for simplicity, and one simple and intuitive comparison is as follows:
>
> + Following the assumptions in this paper, it is necessary to contain the term $\tilde{\mathcal{O}}(\frac{\zeta_\ast^2}{\mu R})$ ($S=1$) for the final result of FedAvg with partial participation, i.e., the term marked with red in Eq. (12) at line 200 in the paper, which is caused by partial participation variance. In other words, the final result of FedAvg with partial participation will contain at least one term $\tilde{\mathcal{O}}(\frac{\zeta_\ast^2}{\mu R})$. However, the dominant term of SSL is $\tilde{\mathcal{O}}(\frac{L\zeta_\ast^2}{\mu^2MR^2})$ (see Table 1). Then for sufficiently large $R$, the bound of FedAvg with partial participation is worse than our bound of SSL.
> + In summary, SSL would exhibit better convergence bound than FedAvg with partial participation based on the existing assumptions and techniques in this paper. This may require other assumptions or techniques for FedAvg to fill the gap, and is beyond this work.
>
> ---
>
> ##### **R5: (Q1) Compare the theoretical results of Yun et al. (2022) with SSL.**
>
> One main contribution of Yun et al, (2022) [Ref 1] is the convergence guarantees of Minibatch RR and local RR. Considering that local RR with multiple local steps is more relevant to SSL, we focus on the comparison to local RR in the following aspects:
>
> + *Algorithms*. Local RR is one variant of local SGD, where clients perform Random Reshuffling locally (in parallel) instead of SGD. Note that local SGD and FedAvg can be considered equivalent in the context of this work. For this reason, local RR is different from SSL in the algorithm (mechanism) aspect.
>
> + *Assumptions*. (1) Yun et al, (2022) considered the non-convex case with the $\mu$-PL condition, which can be viewed as a nonconvex generalization of strongly convex functions; while we consider three cases of strongly convex, convex and non-convex. (2) Yun et al, (2022) considered a different assumption to bound the heterogeneity among local objectives:
>   $$
>   \frac{1}{M}\sum_{m=1}^M\||\nabla F_m (x)\|| \leq \tau + \rho \||\nabla F(x)\||
>   $$
>   According to their analyses in Appendix A, this assumption is weaker than Eq. (4), but stronger than Eq. (5) of our Assumption 3.
>
> + *Upper bounds*. Since $\mu$-PL condition is one weaker yet close assumption to $\mu$-strongly convex, we compare their bound with ours in the strongly convex case. Besides, due to differences in assumptions, we need to translate their bounds into our settings. The translated upper bounds of their Theorem 2 and our bound of SSL are summarized in [Table 1] below.
>
>   [Table 1: Upper bounds of SSL and local RR. Noting that their bounds hold only for the large epoch regime, i.e., $R$ is sufficiently large, the negative exponential term $\mu D^2 \exp(-\frac{\mu R}{L})$ of our bound can be omitted. The number of instances of each client is denoted as $N$. The difference between the two bounds is marked in red.]
>
>   | Algorithm | Bound                                                        |
>   | --------- | ------------------------------------------------------------ |
>   | local RR  | $\displaystyle \tilde{\mathcal{O}} \left({\color{red}\frac{L^2}{\mu^2}\cdot \frac{N}{KR}}\cdot \frac{\sigma^2}{\mu MKR} + {\color{red}\frac{L}{\mu}\cdot M} \cdot \frac{L\sigma^2}{\mu^2 MKR^2} + {\color{red}\frac{L}{\mu}\cdot M} \cdot \frac{L\zeta_\ast^2}{\mu^2MR^2}\right)$ |
>   | SSL       | $\displaystyle \tilde{\mathcal{O}} \left(\frac{\sigma^2}{\mu MKR} + \frac{L\sigma^2}{\mu^2 MKR^2} + \frac{L\zeta_\ast^2}{\mu^2MR^2}\right)$ |
>
>   We can observe that the first term of local RR is better than the corresponding one of SSL when $\frac{KR}{N} \geq \frac{L^2}{\mu^2}$ yet the last two terms are strictly worse than their corresponding terms of SSL. At last, we would like to clarify that this comparison may not be reasonable considering that Yun et al (2022)'s are high-probability bounds while ours are in-expectation bounds, so more research is still required for a comprehensive comparison.
>
> ---
>
> ##### **R6: (Q2) Does the improvement of SSL come from a better bound on the client drift?**
>
> Yes. Similar to the bound of FedAvg, we can divide the terms into three components, global stochastic term, client drift term and optimization term, where the client drift term is caused by "client drift" and the other two terms appear in the standard convergence bound of SGD (see the 1-st row of Table 1).
>
> As we can see, the advantage of SSL shows in the client drift term, which benefits from its sequential and shuffling-based training manner.
> $$
> \tilde{\mathcal{O}} \left(\underbrace{\frac{\sigma^2}{\mu MKR}}\_{\text{global stochastic term}} + \underbrace{\frac{L\sigma^2}{\mu^2 MKR^2} + \frac{L\zeta\_\ast^2}{\mu^2MR^2}}\_{\text{client drift term}}+\underbrace{\mu D^2 \exp(-\frac{\mu}{12L}R)}\_{\text{optimization term}} \right)
> $$

---

> > ### Comment · Reviewer_4Dha · 2023-08-17
> > **Additional Comments**
> >
> > Thanks for the rebuttal. The comparison to (Yun et al 2022) seems relevant and I would suggest including them in future versions of the paper. Some additional comments --
> > - **SSL as a special case of FedAvg with partial participation**: I made an error when comparing FedAvg with partial participation to Algorithm 1 and apologize for this.  Algorithm 1 is FedAvg with client selection similar to "randomized reshuffling" and cannot be equivalent to any version of FedAvg with partial client participation.
> > - **Comparison to Randomized Reshuffling**: Is there a reason why the optimization term of Algorithm 1 is worse than Randomized Reshuffling by a factor of $M$?
> > - **Algorithm 1 v/s SSL**: I would suggest the authors should use SSL only as motivation as Algorithm 1 is basically Sequential FL or FedAvg with "randomized reshuffling" client selection.

---

> > > ### Author Response · Authors · 2023-08-18
> > >
> > > Thanks for the careful proofreading. The responses to the additional comments are provided below.
> > >
> > > **R27: More detailed comparison to Random Reshuffling**
> > >
> > > The reason is that Theorem 1 in Mishchenko et al. (2020) [Ref 5] for the bound of Random Reshuffling (RR) applied one meticulously designed technique (i.e., introducing their Definition 2, Shuffling variance), which yet can not be applied to the convergence analyses of Algorithm 1 (with multiple local steps). The details (combined with our proof) are as follows:
> > >
> > > + The difference in the optimization term arises from the constraint of the learning rate. For Algorithm 1, the constraint (upper constraint) of the strongly convex case in Theorem 1 is $\eta \leq \mathcal{O}(\frac{1}{{\color{red}M}L})$ when $K=1$. In contrast, for RR, the constraint of Mishchenko et al. (2020)'s Theorem 1 is $\eta\leq \mathcal{O}(\frac{1}{L})$. As a larger value of $\eta$ leads to better convergence rate in the optimization term (Please refer to our Lemma 7, tuning the learning rate, in Appendix B.2), the difference of the learning rate constraints causes the worse bound (by a factor of $M$) of Algorithm 1 than RR.
> > > + The tougher constraint of the learning rate in Theorem 1 (Algorithm 1) arises from bounding the client drift error (Please refer to our Lemma 6, Bounding the client drift, in Appendix B.2). Mishchenko et al. (2020) introduced the shuffling variance (see their Definition 2) to bypass bounding the client drift error (they called it as "deviation"). Such technique leads to the relaxed learning rate constraint ($\eta \leq \mathcal{O}(\frac{1}{L})$) and better optimization term ($\tilde{\mathcal{O}}\left(D^2 \exp(-\frac{\mu {\color{red}M}R}{L})\right)$) for RR than that (the constraint $\eta \leq \mathcal{O}(\frac{1}{{\color{red}M}L})$ and the optimization term $\tilde{\mathcal{O}}\left(D^2 \exp(-\frac{\mu R}{L})\right)$) for Algorithm 1. However, it is hard to apply this trick to our Theorem 1 due to multiple local steps in Algorithm 1.
> > > + We also want to highlight that, in the general convex and non-convex cases (not the strongly convex case above), our convergence bounds (Theorems 1 and Corollary 1) for Algorithm 1 (when $\sigma=0$ and $K=1$) are identical to the bounds of RR (Theorems 3, 4 in Mishchenko et al. (2020)). This is because the introduced shuffling variance technique is not applicable (and not used) in the general convex and non-convex cases in Mishchenko et al. (2020). We will provide more discussions about these two cases in the revision.
> > >
> > > ---
> > >
> > > **R28: Clarification for the other additional comments**
> > >
> > > + *Discussion about Yun et al, (2022)*. We will include the discussion about Yun et al, (2022) [Ref 1] in the revision as suggested.
> > > + *SSL as a special case of FedAvg with partial participation*. As pointed out, Algorithm 1 is similar to Random Reshuffling (RR) and cannot be equivalent to FedAvg with partial participation in the scope of this paper. We will add more comparison analysis with RR in the revision (as stated in R27 above).
> > > + *Algorithm 1 v/s SSL*. As suggested, we will clarify the difference between Algorithm 1 and SSL, and the applicability of our convergence results in the revision.

---

### Author Rebuttal · Authors · 2023-08-09

##### **R1: There is no explicit explanation of "Split" Learning (or split computing).**

***Clarification for no explicit explanation of Split Learning***. We would like to clarify that, due to the page limitation, we only state the basic model update rule of Sequential Split Learning (SSL) in Algorithm 1 of the Introduction. Nevertheless, Algorithm 1 has carried all the information for the convergence analysis of SSL in this paper. The complete training process (with operation details) of SSL is shown in Appendix D. In particular, Algorithm 3 and 4 (in Appendix D) are the detailed version of Algorithm 1.

***Introduction on the complete training processes of SSL and SFL***. To address the reviewers' concerns, we also briefly introduce the complete training processes of two typical Split Leaning (SL) algorithms. SL is proposed to address the computation bottleneck of resource-constrained devices, where the full model is split into two parts: the *client-side model* (front-part) and the *server-side model* (back-part). There are two typical algorithms in SL, i.e., Sequential Split Learning (SSL) and Split Federated Learning (SFL). In this work, we consider the first version of SFL, known as SFLV1. Complete training processes of SSL and SFLV1 are as follows:

+ SSL: Each client keeps one client-side model and the server keeps one server-side model. Thus, each client and the server can collaborate to complete the *local training* of a full model (on the local data kept in clients). The local training operations of each client include: 1) the client executes the forward pass on the local data, and sends the activations of the cut-layer (called the smashed data) and labels to the server; 2) the server executes the forward pass with the received smashed data and computes the loss; 3) the server executes the backward pass and send the gradients of the smashed data to the client; 4) the client executes the backward pass with the received gradients. After finishing the local training, the client sends the updated parameters of its client-side model to the next client. This process continues until all clients complete their local training.
+ SFLV1: Each client keeps one client-side model and the server keeps (named as *main server*) keeps multiple server-side models, whose quantity is the same as the number of clients. Thus, each client and its corresponding server-side model in the main server can complete the local training of a full model in parallel. The local training operations of any client in SFLV1 are identical to that in SSL. After the local training with the server, clients send the updated parameters to the fed server, one server introduced in SFLV1 to achieve the aggregation of client-side models. The fed server aggregates the received parameters and sends the aggregated (averaged) parameters to the clients. The main server also aggregates the parameters of server-side models it kept and updates them accordingly.

***Relationship between SFLV1 and FedAvg***. According to the complete training process of SFLV1, we can conclude the relationship between SFLV1 and FedAvg as follows:

+ SFLV1 shares the same update rule with FedAvg (Algorithm 2);
+ SFLV1 can be viewed as the practical implementation of FedAvg in the context of Split Learning (i.e., partitioning the full model into the client-side and server-side portions).

---

##### Some notes

+ We use "R" to represent the response, W to represent the weakness, Q to represent the question;
+ We use "[Table]" to represent the table and "[Ref]" to represent the reference appearing in the rebuttal response.

##### References appearing in the rebuttal

[Ref 1] Yun, C., Rajput, S., & Sra, S. (2021, October). Minibatch vs Local SGD with Shuffling: Tight Convergence Bounds and Beyond. In *International Conference on Learning Representations*.

[Ref 2] Wang, J., Das, R., Joshi, G., Kale, S., Xu, Z., & Zhang, T. (2022). On the unreasonable effectiveness of federated averaging with heterogeneous data. *arXiv preprint arXiv:2206.04723*.

[Ref 3] Thapa, C., Arachchige, P. C. M., Camtepe, S., & Sun, L. (2022, June). Splitfed: When federated learning meets split learning. In *Proceedings of the AAAI Conference on Artificial Intelligence* (Vol. 36, No. 8, pp. 8485-8493).

[Ref 4] Gao Y, Kim M, Thapa C, et al. Evaluation and optimization of distributed machine learning techniques for internet of things[J]. IEEE Transactions on Computers, 2021, 71(10): 2538-2552.

[Ref 5] Mishchenko, K., Khaled, A., & Richtárik, P. (2020). Random reshuffling: Simple analysis with vast improvements. *Advances in Neural Information Processing Systems*, *33*, 17309-17320.

[Ref 6] Safran, I., & Shamir, O. (2020, July). How good is SGD with random shuffling?. In *Conference on Learning Theory* (pp. 3250-3284). PMLR.

[Ref 7] He, K., Zhang, X., Ren, S., & Sun, J. (2016). Deep residual learning for image recognition. In *Proceedings of the IEEE conference on computer vision and pattern recognition* (pp. 770-778).

[Ref 8] Darlow, L. N., Crowley, E. J., Antoniou, A., & Storkey, A. J. (2018). Cinic-10 is not imagenet or cifar-10. *arXiv preprint arXiv:1810.03505*.

[Ref 9] Jhunjhunwala, D., Wang, S., & Joshi, G. (2022, September). FedExP: Speeding Up Federated Averaging via Extrapolation. In *The Eleventh International Conference on Learning Representations*.

[Ref 10] Zagoruyko, S., & Komodakis, N. (2016). Wide residual networks. *arXiv preprint arXiv:1605.07146*.

---

### Decision · Program_Chairs · 2023-09-21

**Decision:**

Accept (poster)

**Comment:**

The paper analyses sequential versions of federated learning in the realistic heterogeneous data case. This is a very relevant and timely research topic. Reviewers all agree that the contributions are valuable in this setting.

However it remained unclear what the results have to do with the split learning aspect, a particular kind of model parallelism where the server and clients hold different blocks of layers of the model.
The provided results seem to apply in the more general setting of standard learning as well, but it should be clarified what aspects of the analysis are general, as opposed to specific to the split learning setting.
We think this does not diminish the results of the paper but rather could strengthen and generalize them if explained better (even potentially changing the title).

To improve this, please in the paper clearly formalize the results and implications (and novelty) in both settings:
1) the more specialized case of split learning and
2) the more broadly interesting case of general sequential learning.

We hope the detailed feedback can help to clarify the camera-ready version of the paper.